# PROVABLE HIERARCHY-BASED META-REINFORCEMENT LEARNING

## ABSTRACT

Hierarchical reinforcement learning (HRL) has seen widespread interest as an approach to tractable learning of complex modular behaviors. However, existing work either assume access to expert-constructed hierarchies, or use hierarchy-learning heuristics with no provable guarantees. To address this gap, we analyze HRL in the meta-RL setting, where a learner learns latent hierarchical structure during meta-training for use in a downstream task. We consider a tabular setting where natural hierarchical structure is embedded in the transition dynamics. Analogous to supervised meta-learning theory, we provide "diversity conditions" which, together with a tractable optimism-based algorithm, guarantee sample-efficient recovery of this natural hierarchy. Furthermore, we provide regret bounds on a learner using the recovered hierarchy to solve a meta-test task. Our bounds incorporate common notions in HRL literature such as temporal and state/action abstractions, suggesting that our setting and analysis capture important features of HRL in practice.

## 1 INTRODUCTION

Reinforcement learning (RL) has demonstrated tremendous successes in many domains (Schulman et al., 2015; Vinyals et al., 2019; Schrittwieser et al., 2020), learning near-optimal policies despite limited supervision. Nevertheless, RL remains difficult to apply to problems requiring temporally extended planning and/or exploration (Ecoffet et al., 2021). A promising approach to this problem is hierarchical reinforcement learning (HRL), which has seen continued interest due to its appealing biological basis. In its most basic form, HRL seeks to decompose tasks into a sequence of skills, each of which is easier to learn individually than the full task. By restricting the agent to using learned skills, the search space over policies can be greatly reduced. Furthermore, learned skills can induce simpler state and/or action spaces, simplifying the learning problem. Finally, learned skills with useful semantic behavior can be reused across tasks, enabling transfer learning.

Naturally, a hierarchy-based learner is limited by the quality of skills that are made available and/or learned. Accordingly, many empirical works have proposed algorithms for online skill learning in the context of a single RL task (Nachum et al., 2019a; 2018). These approaches have been experimentally demonstrated to be effective in finding useful and interpretable skills. Other approaches consider the skill learning problem in the context of meta-RL (Frans et al., 2018), or in the reward-free setting (Eysenbach et al., 2018). Nevertheless, the heuristics and algorithms proposed in these empirical works do not provide any provable guarantees on the quality of learned skills.

On the other hand, theoretical analyses have mostly focused on how learners benefits from having access to skills. For example, Fruit & Lazaric (2017) provide a regret bound on learning with skills in the infinite-horizon average reward case. Meanwhile, in the meta-RL setting, Brunskill & Li (2014) considers the problem of finding and using skills in a continual learning setting and provides a sample complexity analysis. However, these analyses either sidestep the question of how the skills are obtained, or do not address the problem in a computationally tractable manner.

In this work, we aim to take a step towards providing provable guarantees for hierarchy learning through tractable algorithms. We focus on the meta-RL setting, in which a learner extracts skills from a set of provided tasks which are then used in a downstream task. We work in the tabular case, assuming the transition dynamics of the given tasks share latent hierarchical structure induced by predetermined clustering and bottlenecks. Our contributions are as follows:

1. **"Diversity conditions" ensuring hierarchy recovery.** We develop natural optimism-based coverage conditions which ensure that bottlenecks embedded in the transition dynamics are detectable by solving provided meta-training tasks.

2. **A tractable hierarchy-learning algorithm.** We provide an algorithm that provably learns the latent hierarchy from interactions, assuming the coverage conditions above. Our method has sample complexity scaling as $O(TKS)$ in the leading term compared to $O(TS^2A)$ for a brute-force method, where $T$ are the number of tasks, $S$ is the number of states, $A$ is the number of actions, and $K \ll SA$ is the number of skills to learn.

3. **Regret bounds on downstream tasks.** We provide regret bounds for learners that apply the extracted hierarchy from meta-training on downstream tasks. Furthermore, we show an exponential regret separation between hierarchy-based and hierarchy-oblivious learners for a family of task distributions, corroborating prevailing intuitions regarding when/why HRL helps. In particular, hierarchy-based learners incur regret bounded by $O(\sqrt{H^2N})$ while hierarchy-oblivious learners incur worst-case regret of at least $O(2^{H/2}\sqrt{H^2N})$.

## 2 NOTATION

We now introduce notation which we will use throughout the paper. We write $[K] := \{1, \ldots, K\}$. Furthermore, we use the standard notations $O, \Theta, \Omega$ to denote orders of growth, and $\tilde{O}, \tilde{\Theta}, \tilde{\Omega}$ to indicate suppressed logarithmic factors. We use $\delta(x)$ to denote the Dirac delta measure on $x$.

We work with finite-horizon Markov decision processes (MDPs), defined as a tuple $\mathcal{M} = (\mathcal{S}, \mathcal{A}, \mathbb{P}, r, H)$, where $\mathcal{S}$ is the set of states, $\mathcal{A}$ is the set of actions, $\mathbb{P} : \mathcal{S} \times \mathcal{A} \times \mathcal{S} \to [0, 1]$ are the transition dynamics, $r : \mathcal{S} \times \mathcal{A} \to [0, 1]$ is the reward function, and $H$ is the horizon. We assume stationary dynamics unless otherwise noted, in which case $\mathbb{P}^{(h)}$ is the dynamics at time step $h$. For constants relating to horizons, we will define $[H] := \{0, \ldots, H-1\}$. Given a policy $\pi : [H] \times \mathcal{S} \to \mathcal{A}$, we define the value functions

$$V_h^\pi(s) := \mathbb{E}\left[\sum_{k=h}^{H-1} r(s_k, a_k) \,\middle|\, s_h = s\right] \quad \text{and} \quad Q_h^\pi(s, a) := \mathbb{E}\left[\sum_{k=h}^{H-1} r(s_k, a_k) \,\middle|\, (s_h, a_h) = (s, a)\right]$$

where $s_{k+1} \sim \mathbb{P}(\cdot \mid s_k, a_k)$ and $a_k = \pi_k(s_k)$. Furthermore, we write $V^*$ and $Q^*$ to denote optimal value functions obtained by maximizing over $\pi$ (and are attained by the optimal policy $\pi^*$). When a learner plays $\pi_1, \ldots, \pi_N$ in $\mathcal{M}$, we define its regret as

$$\text{Regret}_N(\mathcal{M}) := \sum_{t=1}^{N} V_0^*(s_0) - V_0^{\pi_t}(s_0).$$

We use $\ominus$ to denote a terminal state. We let $\tau_\pi$ denote the (random) trajectory generated by $\pi$. For a state $s$ and length-$H$ trajectory $\tau$, we write $s \in \tau_\pi$ if $s_h = s$ for some $h \in [H]$. We define $(s, a) \in \tau_\pi$ similarly. Finally, given an MDP $\mathcal{M}$, $\mathcal{M}(2H)$ denotes a copy of $\mathcal{M}$ with a doubled horizon.

## 3 SETTING

We work in the tabular meta-RL setting. The learner can access $T$ meta-training MDPs $\{(\mathcal{S}, \mathcal{A}, \mathbb{P}_t, r_t, H)\}_{t\in[T]} = \{\mathcal{M}_t\}_{t\in[T]}$. Note that $\mathbb{P}_t$ and $r_t$ both vary across tasks. We set $S := |\mathcal{S}|$ and $A := |\mathcal{A}|$. After interacting with these tasks, the learner is presented with a meta-test MDP $(\mathcal{S}, \mathcal{A}, \mathbb{P}_{\text{Tg}}, r_{\text{Tg}}, H)$, where the learner seeks to minimize its regret. We assume, without loss of generality, that the MDPs have a shared starting state $s_0$. For meta-learning to succeed, the MDPs must have shared structure. We focus on the following notion of shared hierarchical structure:

**Definition 3.1** (Latent Hierarchy)**.** Let $\{Z_c\}$ be a partition of the state space $\mathcal{S}$ into clusters. We associate with each cluster $Z$ a set of entrances $\text{Ent}(Z) \subseteq Z$ and a set of exits $\text{Ext}(Z) \subseteq Z \times \mathcal{A}$. We say that the tasks have a *latent hierarchy* with respect to $(\{Z_c\}, \text{Ent}(\cdot), \text{Ext}(\cdot))$ if for any $Z_c$:

(a) For any $(s, a) \in (Z_c \times \mathcal{A}) \setminus \text{Ext}(Z_c)$, $\mathbb{P}_t(\cdot \mid s, a)$ is constant over $t$ and supported on $Z_c$. ◇

(b) For any $(s, a) \in \text{Ext}(Z_c)$, there exists $t, t'$ with $t \neq t'$ such that $\mathbb{P}_t(\cdot \mid s, a) \neq \mathbb{P}_{t'}(\cdot \mid s, a)$. Furthermore, $\mathbb{P}_t(\cdot \mid s, a)$ is supported on $\bigcup_c \text{Ent}(Z_c)$ for any $t \in [T]$.

Definition 3.1 partitions the shared state space of the MDPs into clusters such that (1) non-exit $(s, a)$ dynamics do not change between the MDPs and (2) exits are bottlenecks between clusters.

**Example 3.1** (Gated Four-Room). Figure 1 illustrates the gated four-room environment along with a sample task. The environment has a latent hierarchy with respect to the rooms outlined by colored gates (which can be open or closed). Entrances are colored aqua, while exits are marked by arrows.

Although we assume a single fixed starting state, we can model task-dependent initial states by appending a dummy state $s_0$. A dummy action $a_0$ then takes the agent to the starting state for the task. Observe that $(s_0, a_0)$ is an exit, and therefore must transition to an entrance.

To see how Definition 3.1 captures intuitive notions of hierarchy in practical settings, we provide an example of a continuous setting roughly fitting into our framework:

**Example 3.2** (The Alchemy benchmark). Alchemy (Wang et al., 2021) is a recently proposed empirical benchmark for meta-RL, where the agent needs to place a stone in a series of potions to obtain some desired appearance, as illustrated in Figure 2. Dipping a stone into a potion traverses an edge (determined by the potion) in a graph where nodes are possible stone appearances. We focus on task distributions that randomize the edges of this graph (i.e., potion positions and feasible stone appearances are fixed). Then, the set of obtainable MDPs has a latent hierarchy where dipping the stone into any of the potions is an exit. Indeed, other than dipping the stone into a potion, all other actions (e.g., moving the stone around the room) have the same dynamics in all tasks.

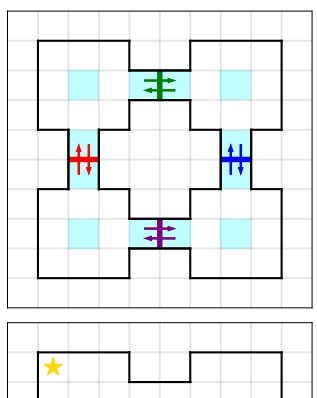

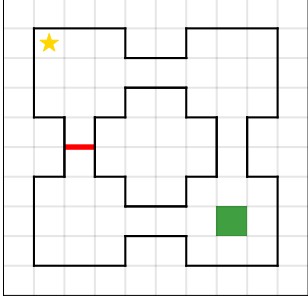

Figure 1: The gated four-room setting, with a task requiring navigation from the green square to the star.

For convenience, we will define several relevant notions. First, for any cluster $Z$, we define its *interior* $Z^\circ := (Z \times \mathcal{A}) \setminus \text{Ext}(Z)$. Furthermore, we let $\text{Ent}(\mathcal{S}) := \bigcup_c \text{Ent}(Z_c)$ denote the set of all entrances and $\text{Ext}(\mathcal{S}) := \bigcup_c \text{Ext}(Z_c)$ the set of all exits. Finally, we define

$$K := |\text{Ext}(\mathcal{S})| \qquad L := |\text{Ent}(\mathcal{S})|$$
$$M := \sup_c |\text{Ext}(Z_c)|$$

so that $K$ and $L$ are the total number of exits and entrances, respectively, while $M$ is the maximal number of exits from any cluster.

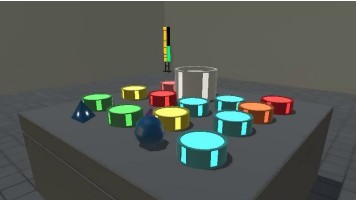

Figure 2: Alchemy. Placing stones in potions moves the agent through a latent graph of object properties.[1]

**Query Model.** We work in the online setting, where the agent interacts with the tasks by playing policies from the initial state $s_0$. During meta-training, we allow the agent to interact with the environments using an unbounded number of timesteps for each trajectory before resetting. We then compute query complexity in terms of the total number of timesteps spent in all tasks in total.

## 4 META-TRAINING ANALYSIS

In this section, we outline an algorithm for uncovering the latent structure that can be used for downstream tasks. Recall that exits are defined by changing dynamics between tasks. To quantify the number of samples needed for exit detection, we define the following quantity:

---

[1]Image from Wang et al. (2021), extracted from a larger figure with no other modifications (License).

**Definition 4.1** ($\beta$-dynamics separation). There exists $\beta > 0$ such that for any $t, t' \in [T]$ and $(s, a) \in \text{Ext}(\mathcal{S})$, $\mathbb{P}_t(\cdot \mid s, a) \neq \mathbb{P}_{t'}(\cdot \mid s, a) \implies \|\mathbb{P}_t(\cdot \mid s, a) - \mathbb{P}_{t'}(\cdot \mid s, a)\|_{\text{TV}} \geq \beta$. ◇

In defining $\beta$ above, we ensure high-probability exit detection with $\tilde{O}(S/\beta^2)$ samples. Note that there is a brute-force approach to learning the underlying structure: one can learn $\mathbb{P}_t(\cdot \mid s, a)$ for all $(s, a)$ and $t \in [T]$, and iterate over $(s, a)$ to check for changing dynamics. This can be done with query complexity $\tilde{O}(TS^2A/\beta^2)$. However, under reasonable "coverage" assumptions outlined in the next section, this query cost can be lowered to $\tilde{O}(TKS/\beta^2)$.

## 4.1 DEFINING A NOTION OF COVERAGE

In supervised meta-learning, "diversity conditions" ensure that the meta-training tasks reveal the underlying latent structure (Tripuraneni et al., 2020; Du et al., 2020). We provide analogous conditions ensuring that $\mathcal{M}_1, \ldots, \mathcal{M}_T$ "cover" the latent hierarchy. Since solving $\max_\pi V^\pi(s_0)$ requires fewer samples than learning $\mathbb{P}$, we expect such conditions to provide sample complexity gains.

**Visitation Probabilities and $\alpha$-Importance.** Minimally, exits should be visited by optimal policies of the meta-training tasks for coverage. We thus define the following notion:

**Definition 4.2.** Fix an MDP $\mathcal{M} = (\mathcal{S}, \mathcal{A}, \mathbb{P}, r, H)$, and let $(s, a) \in \mathcal{S} \times \mathcal{A}$. Construct a modified MDP $\mathcal{M} \setminus (s, a)$, where $(s, a)$ brings the agent to a terminal state with no reward. Then, we say that $(s, a)$ is $\alpha$-important for $\mathcal{M}$ if $V_0^{\mathcal{M} \setminus (s,a), *}(s_0) < V_0^{\mathcal{M}, *}(s_0) - \alpha$. ◇

The $\alpha$-importance condition quantifies the value gap between policies that can use $(s, a)$ and those that cannot. For example, consider the task in Figure 3. Any $\pi$ with $V^\pi(s_0) > 0$ must visit the marked $(s, a)$ pair with some probability. Therefore, $(s, a)$ is $V_0^*(s_0)$-important. This suggests that high $\alpha$-importance implies high visitation probability by near-optimal policies. The following result, proven in Section A, formalizes this connection:

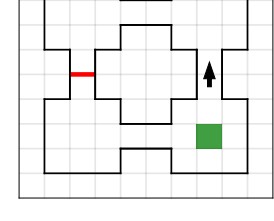

**Lemma 4.1.** If $(s, a)$ is $\alpha$-important for $\mathcal{M}$, then for any policy $\pi$, $V^*(s_0) - V^\pi(s_0) < \varepsilon$ implies that $P((s, a) \in \tau_\pi) > (\alpha - \varepsilon)/H$.

**A Preliminary Coverage Assumption?** Lemma 4.1 suggests a simple coverage condition: for any $(s, a) \in \text{Ext}(\mathcal{S})$, assume that there exists $t, t' \in [T]$ so that $(s, a)$ is $\alpha$-important for $t$ and $t'$, and $\mathbb{P}_t(\cdot \mid s, a) \neq \mathbb{P}_{t'}(\cdot \mid s, a)$. However, as the following example shows, this condition excludes natural settings:

Figure 3: Illustrating $\alpha$-importance. Since the black arrow is the only path to the goal, it is $V_0^*(s_0)$-important.

**Example 4.1.** In Example 3.1, gates are either open or closed. For closed gates, the associated $(s, a)$ pairs are unimportant for any goal. Thus, if $(s, a)$ is $\alpha$-important for some $t, t' \in [T]$ with $\alpha > 0$, then $\mathbb{P}_t(\cdot \mid s, a) = \mathbb{P}_{t'}(\cdot \mid s, a)$, and no change in dynamics can be detected using $\pi_t^*$ and $\pi_{t'}^*$.

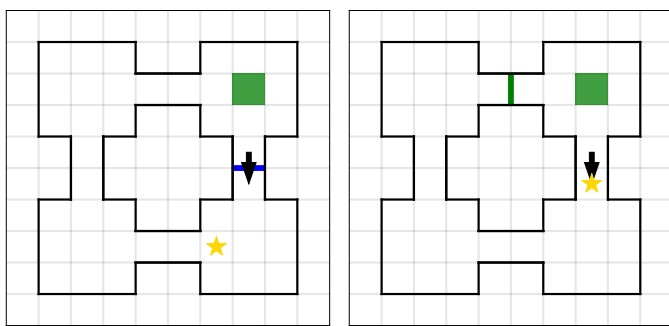

Figure 4: The black arrow is not $\alpha$-significant for one of the tasks, but is nevertheless "covered" by optimistic imagination.

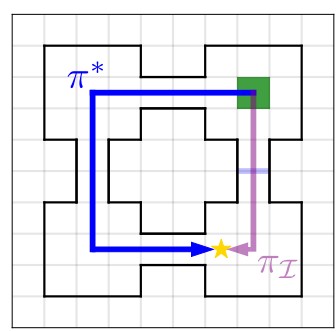

Figure 5: Using optimistic imagination for exit detection.

Figure 4 illustrates how the proposed condition fails in covering an exit marked with a black arrow. While $(s, a)$ is unused by optimal policies in the first task, it is $V_0^*(s_0)$-important for the second. ⌐

**Optimistic Imagination as a Coverage Mechanism.** The proposed assumption fails because there are cases where exits are only $\alpha$-important in certain configurations (e.g., only open corridors are $\alpha$-important in Example 3.1). In such cases, near-optimal policies only ever see one configuration of the dynamics for those exits and thus are insufficient for formalizing exit coverage.

As an alternative, consider the following hypothetical scenario in the context of Figure 4: an agent has solved both tasks, achieving optimal values $V_1^*$ and $V_2^*$. Additionally, in the process of learning the second task, the agent has learned $\mathbb{P}_2(\cdot \mid \Downarrow)$. If the agent then relearns the first task while setting $\hat{\mathbb{P}}_1(\cdot \mid \Downarrow) \leftarrow \mathbb{P}_2(\cdot \mid \Downarrow)$, it would obtain a new value $\hat{V}_1 \gg V_1^*$. Thus, it can reasonably conclude that the black arrow must have been an exit. We illustrate this process in Figure 5, where $\pi_{\mathcal{I}}$ is the optimal policy after "borrowing dynamics." The learner could then run $\pi_{\mathcal{I}}$ for exit detection.

We refer to the counterfactual reasoning about the dynamics used above as *optimistic imagination*. Unlike the preliminary condition, optimistic imagination only requires that an exit be important for one task and induce a value gap in another when borrowing dynamics – a weaker condition in many cases. With the above intuition in mind, we now present the main coverage assumption.

**Assumption 4.1** (($\alpha, \zeta$)-coverage). *Assume $(\mathcal{M}_t)_{t \in [T]}$ have a latent hierarchy with respect to $(\{Z_k\}, \mathrm{Ent}(\cdot), \mathrm{Ext}(\cdot))$. There exists $\alpha, \zeta > 0$ such that for any $\{(s_1, a_1), \ldots, (s_n, a_n)\} \subseteq \mathrm{Ext}(\mathcal{S})$,*

    *(a) For any $i \in [n]$, $(s_i, a_i)$ is $\alpha$-important for some meta-training MDP $\mathcal{M}_i$.*

    *(b) For some $\mathcal{M}_t$ with $t \in [T]$, if we construct a new MDP $\bar{\mathcal{M}}_t = (\mathcal{S}, \mathcal{A}, \bar{\mathbb{P}}, r_t, H)$ via*

$$\bar{\mathbb{P}}(\cdot \mid s, a) = \begin{cases} \mathbb{P}^{\mathcal{M}_i}(\cdot \mid s, a) & (s, a) = (s_i, a_i) \\ \mathbb{P}^{\mathcal{M}_t}(\cdot \mid s, a) & otherwise \end{cases},$$

    *i.e. we replace $(s_i, a_i)$ dynamics with those from $\mathcal{M}_i$, then $V^{\bar{\mathcal{M}}_t, *}(s_0) > V^{\mathcal{M}_t, *}(s_0) + \zeta$.*

Informally, ($\alpha, \zeta$)-coverage ensures that optimistic imagination can borrow dynamics for unknown exits from other tasks to find a better optimal policy.[2] Most reasonable task distributions satisfy ($\alpha, \zeta$)-coverage with enough tasks – we provide a heuristic explanation in the appendix.

## 4.2 ALGORITHM OUTLINE

In this section, we outline the algorithm that we use to detect exits. Our procedure can be naturally divided into three phases: a task-solving phase, a reward-free phase, and an exit detection phase. Throughout, we illustrate our steps in Figure 5, showing how Phases I and II allow the learner to find the imagined policy $\pi_{\mathcal{I}}$. The full details of the algorithm are provided in Section A.1.

### 4.2.1 PHASE I: TASK-SPECIFIC DYNAMICS LEARNING

First, we solve $\mathcal{M}_1, \ldots, \mathcal{M}_T$ with UCBVI. Using UCBVI regret bounds from Azar et al. (2017) together with Lemma 4.1, we can guarantee that all $\alpha$-important exits are sufficiently visited. Thus, during optimistic imagination, the learner would be able to borrow high-quality estimates of exit dynamics from other tasks. For example, a learner that has solved both tasks in Figure 4 can borrow open blue gate dynamics for use in the first task during optimistic imagination, as in Figure 6.

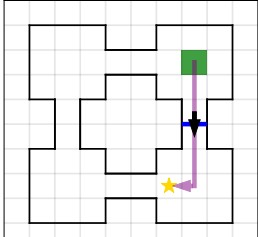

### 4.2.2 PHASE II: REWARD-FREE RL

In order to perform optimistic imagination, the learner also needs to simulate non-borrowed $(s, a)$ dynamics. This is done by fully learning the dynamics of one of the tasks, proving a template $\hat{\mathbb{P}}_0$. Learning $\hat{\mathbb{P}}_0$ is achieved using reward-free RL (Jin et al., 2020).

Figure 6: Phase I contribution to learning $\pi_{\mathcal{I}}$ in Figure 5, marked with an arrow.

---

[2]Incorporating the preliminary condition into Assumption 4.1 can further weaken the required task diversity. However, our algorithm handles this extension trivially, and thus we focus on optimistic imagination.

To understand the necessity of Phase II, note that in Figure 7, near-optimal policies have no coverage over states past the blue gate (as shown by the red region). Therefore, dynamics estimates from Phase I are insufficient for optimistic imagination. On the other hand, by learning the dynamics fully in one of the tasks (and thus learning the green region), the learner can nevertheless simulate the dynamics if the blue gate were open and successfully recover $\pi_{\mathcal{I}}$.

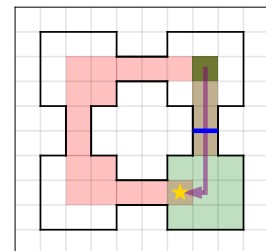

### 4.2.3 PHASE III: EXIT DETECTION

Having completed the previous two phases, the learner can use optimistic imagination to detect exits. In particular, we consider a modified value iteration method where the learner optimistically chooses dynamics estimates from Phases I and II to perform Bellman backups. This implicitly defines an MDP whose optimal value is at least as large as $\bar{\mathcal{M}}_t$ in Assumption 4.1, as $\bar{\mathcal{M}}_t$ would have been feasible for this process. Analogously with $\alpha$-importance, this value gap implies that the corresponding optimal policy $\pi_{\mathcal{I}}$ for this new MDP must visit an $(s, a)$ pair whose dynamics are borrowed. Therefore, by playing $\pi_{\mathcal{I}}$, the learner can determine a new exit.

Figure 7: Phase II contribution. Optimal policy state coverage (red) is insufficient for learning $\pi_{\mathcal{I}}$, which requires the green region.

### 4.3 META-TRAINING GUARANTEE

We now outline our main result for the algorithm in Section 4.2. We first define a "hierarchy oracle" that will be used in downstream tasks:

**Definition 4.3** (Hierarchy oracle). Let $\ominus_S$ and $\ominus_F$ denote successful and failed termination, respectively. Consider any tuple $(x, f, r, \tilde{H})$ such that $x \in \text{Ent}(\mathcal{S})$, $f : \text{Ext}(\mathcal{S}) \to \{\ominus_S, \ominus_F\}$, $r$ is a reward function, and $\tilde{H} \leq H$. Such tuple induces an MDP $\mathcal{M}(x, f, r, \tilde{H}) = (\mathcal{S} \cup \{\ominus_S, \ominus_F\}, \mathcal{A}, \mathbb{P}_f, r, \tilde{H})$ whose starting state is $x$ and whose transition dynamics is given by

$$\mathbb{P}_f(\cdot \mid s, a) = \begin{cases} \delta(f(s, a)) & (s, a) \in \text{Ext}(\mathcal{S}) \\ \delta(s) & s \in \{\ominus_S, \ominus_F\} \\ \mathbb{P}_t(\cdot \mid s, a) & \text{otherwise, for any } t \in [T]. \end{cases}$$

An $\varepsilon$-*suboptimal hierarchy oracle*, when queried with any valid $(x, f, r, \tilde{H})$, returns an $\varepsilon$-suboptimal policy for $\mathcal{M}(x, f, r, \tilde{H})$. ⋄

To motivate Definition 4.3, observe that learning a new MDP with the same latent hierarchy only requires visiting exits (as reward-free learning has coverage over cluster interiors). This, in turn, can be achieved by an agent that has a policy for reaching any exit from every entrance in any cluster. We emphasize that *the learner does not need to know the states in the actual clusters themselves.* Formally, for an entrance $x$ in cluster $Z$ and exit $e \in \text{Ext}(Z)$, we can query the hierarchy oracle with $(x, \mathbb{1}\left[(\cdot, \cdot) = e\right], \mathbb{1}\left[\cdot = \ominus_S\right], H)$ to obtain the desired reaching policy. Disconnecting the clusters ensures that the learner cannot use any other exits, and thus the validity of the policy under any MDP that has the same latent hierarchy.[3]

Our meta-training guarantee ensures that the hierarchy oracle is implementable:

**Theorem 4.1** (Meta-training guarantee, informal). *Under Assumption 4.1 and other assumptions in Section A.2, the data obtained from the algorithm in Section 4.2 allows for:*

*(a) implementing an $\varepsilon$-suboptimal hierarchy oracle, and*

*(b) determining, for every $s \in \text{Ent}(\mathcal{S})$, the reachable exits in the cluster containing $s$,*

*simultaneously with probability at least $1 - p$. Furthermore, this is achieved with query complexity*

$$\tilde{O}\left[T\left(\frac{KL}{\alpha \min(\zeta, \beta)^2} + \frac{KS}{\alpha\zeta^2} + \frac{SA}{\min(\alpha, \zeta)^2} + \frac{KS^2A}{\alpha}\right) + \frac{S^4A}{\min(\varepsilon, \zeta)} + \frac{S^2A}{\min(\varepsilon, \zeta)^2}\right] \text{poly}(H).$$

---

[3]As a side effect, the reachability of $e$ from $x$ can be determined from the predicted value for the prior query.

As a point of comparison, we have the following guarantee on brute-force hierarchy learning:

**Theorem 4.2.** *The brute-force approach outlined in Section A.5, under Assumption 4.1(a), determines the set of exits with high probability and query complexity*

$$\tilde{O}\left[T\left(\frac{S^2A}{\alpha\beta^2} + \frac{SA}{\alpha^2} + \frac{S^4A}{\alpha}\right)\right]\text{poly}(H).$$

When $\alpha$, $\beta$, and $\zeta$ are of the same order, we see that the proposed method incurs a smaller query complexity compared to a brute force learner that has only learned the exits. We provide proofs of both results in Section A, along with all other necessary assumptions and full algorithm details.

## 5 META-TEST ANALYSIS

In this section, we provide regret bounds on learning an MDP $\mathcal{M}_{\text{Tg}}$ using the hierarchy oracle. We first characterize a family of tasks for which one can achieve improved regret bounds. Furthermore, we provide sufficient conditions ensuring that using the hierarchy incurs low suboptimality.

### 5.1 ASSUMPTIONS

In this section, we outline the assumptions that we make to prove a regret bound on the meta-test task. Let $\mathcal{M}_{\text{Tg}} = (\mathcal{S}, \mathcal{A}, \mathbb{P}_{\text{Tg}}, r_{\text{Tg}}, H)$ be the meta-test MDP. We assume that $\mathbb{P}_{\text{Tg}}(\cdot \mid s, a) = \mathbb{P}_t(\cdot \mid s, a)$ for any $(s, a) \notin \text{Ext}(\mathcal{S})$ and $t \in [T]$. Our first assumption restricts our attention to meta-test tasks which are compatible with the hierarchical structure:

**Assumption 5.1** (Task Compatibility). *There exists a cluster $Z^*$ such that $r_{\text{Tg}}$ is supported on $(Z^*)^\circ \cup \text{Ext}(\mathcal{S})$. Furthermore, there exists an optimal policy $\pi^*$ satisfying*

(a) *Conditioned on $s_h \in Z^*$, we have that $(s_{h'}, a_{h'}) \notin \text{Ext}(Z^*)$ for $h' \geq h$ almost surely.*

(b) *The number of exits encountered by $\pi^*$ is bounded by $H_{\text{eff}}$ with probability $\zeta$.*

**Hierarchical compatibility.** Intuitively, Assumption 5.1 suggests that the task can be decomposed into a $(Z^*)$-searching phase and a within-$(Z^*)$ phase. This decomposition captures the goal-conditioned RL setting, which has been studied extensively in recent empirical works (Nachum et al., 2019a; Levy et al., 2018; Nachum et al., 2018). We expect the hierarchy oracle to reduce the complexity of exploration in both phases. Thus, these conditions ensure compatibility with the learned hierarchy.

**Temporal Abstraction.** Since a hierarchical learner makes decisions upon entering a new cluster (to decide which exit to use/whether to stay), hierarchical planning can reduce the planning horizon. Condition (b) quantifies this reduction. Note that in most practical settings, a learner that has access to good sub-skills can achieve low failure probability $\zeta$ even with modest values of $H_{\text{eff}}$.

**Hierarchical Suboptimality.** Restricting the learner to executing hierarchical policies can greatly reduce the search space. While this reduction leads to improved regret bounds, this also incurs approximation error, as the optimal policy may not lie in this restricted class. We refer to this error as *hierarchical suboptimality*. We will show that hierarchical suboptimality is controllable with appropriate conditions on $\mathbb{P}_T$, which require the following notions of reaching times:

**Definition 5.1** (Reaching times). Fix a cluster $Z$, starting and goal states $s, g \in Z$, and planning horizon $\tilde{H} \leq H$. For any policy $\pi$, let $(s_0, s_1, \ldots, s_{\tilde{H}})$ be the states visited by $\pi$ from $s$, where $s_0 = s$. Then, we define

$$T^\pi_{\tilde{H}}(s, g) := \min\left\{h \in \left\{0, \ldots, \tilde{H}\right\} \,\Big|\, s_h = g \text{ and } s_{h'} \in Z \text{ for } h' < h\right\} \cup \{L\}$$

$$T^*_{\tilde{H}}(s, g) := \inf_\pi \mathbb{E}\left[T^\pi_{\tilde{H}}(s, g)\right] \qquad\qquad \diamond$$

$$T^{\min}(s, g) := \inf_\pi \min\left\{h \in \mathbb{N} \mid P(T^\pi(s, g) = h) > 0\right\}.$$

In words, $T_{\tilde{H}}^\pi(s, g)$ is the time $\pi$ takes to reach a state $g$ from $s$ while remaining within the same cluster. By minimizing this quantity in expectation over all policies, we obtain $T_{\tilde{H}}^*(s, g)$. Finally, $T^{\min}(s, g)$ is the minimum time for which it is possible to reach $g$ from $s$.

**Definition 5.2** (Regular and low-variance dynamics). There exists $\alpha, \beta, \gamma > 0$ such that for any cluster $Z$, states $s, g \in Z$, and horizon $\tilde{H} < H$,

(a) $((\alpha, \beta)$-unreliability) For any deterministic policy $\pi$ with $\mathbb{E}[T_{\tilde{H}}^\pi(s, g)] - T_{\tilde{H}}^*(s, g) < \alpha$, $T_{\tilde{H}}^\pi(s, g)$ has a sub-Gaussian upper tail with variance proxy $\beta^2 \mathbb{E}[T_{\tilde{H}}^\pi(s, g)]^2$.

(b) ($\gamma$-goal-reaching suboptimality) $T_{\tilde{H}}^*(s, g) \le (1 + \gamma) T^{\min}(s, g)$. $\qquad\qquad\qquad \diamond$

To understand $(\alpha, \beta)$-unreliability, note that the condition only considers near-optimal deterministic policies. Therefore, (a) quantifies the randomness in $T^\pi(s, g)$ derived from the transition dynamics. On the other hand, $\gamma$-goal-reaching suboptimality quantifies the regularity of the dynamics, measuring whether near-optimal goal-reaching policies nearly achieve the optimal expected reaching time. Deterministic environments satisfies these conditions with $\alpha = \infty$ and $\beta = \gamma = 0$. We provide an extended discussion of why these quantities control hierarchical suboptimality in Section B.5. Note that the guarantees of Definition 5.2 scales with the cluster width, and thus we have the following final assumption:

**Assumption 5.2.** *For any cluster $Z$ and $s, g \in Z$ with $s \ne g$, $T^{\min}(s, g) \le W$. Furthermore, $H_{\text{eff}} W \ll H$.*

Assumption 5.2 limits the length of the subtasks within each cluster. This is consistent with hierarchy-based methods in practice, with skills only being executed for a limited amount of time. This width bound, together with Assumption 5.1, suggests that $\pi^*$ requires $O(H_{\text{eff}} W)$ timesteps with high probability. Therefore, the condition $H_{\text{eff}} W \ll H$ means that the task horizon is much longer than the minimum time required to complete the task, which often holds in practice.

## 5.2 META-TEST REGRET GUARANTEE

We note that Assumption 5.1 implies that the hierarchy oracle induces a "high-level" MDP. We have the following regret bound on applying EULER to this MDP:

**Theorem 5.1.** *We work under Assumptions 5.1 and 5.2. Furthermore, assume that the learner has access to an $\varepsilon$-suboptimal hierarchy oracle as guaranteed by Theorem 4.1, where $\varepsilon < \alpha$. Then, a learner that applies the procedure in Section B.2 to $\mathcal{M}_{\text{Tg}}$ incurs regret*

$$\text{Regret}(N) \lesssim \sqrt{H^2 H_{\text{eff}} W L M N} + N \varepsilon_{\text{subopt}}$$

$$\varepsilon_{\text{subopt}} := (1 + H_{\text{eff}} + \beta \sqrt{H_{\text{eff}}}) \varepsilon + \left[ \gamma H_{\text{eff}} + \beta (1 + \gamma) \sqrt{H_{\text{eff}}} \right] W + \zeta H.$$

*with high probability.*

Observe that the irreducible hierarchical suboptimality $\varepsilon_{\text{subopt}}$ (i.e. when $\varepsilon = 0$) tends to zero as $\gamma, \beta, \zeta \to 0$. In particular, environments with deterministic in-cluster dynamics do not incur hierarchical suboptimality. We prove this regret bound in Section B.

**When does knowing the hierarchy help?** Consider the binary tree environment in Figure 8. All of the leaves take the learner to a state with exits with probability $1/2$, with the exception of a special leaf $\ell^*$ that does so with probability $(1/2) + \varepsilon$. Rewards can only be collected upon performing one of the exits (blue/purple). To achieve low regret, a learner has to quickly identity $\ell^*$ and the correct exit.

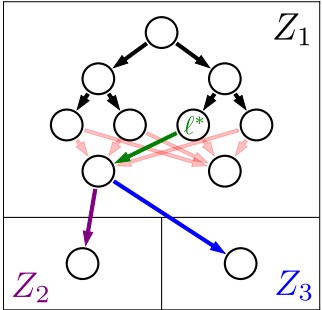

Figure 8: The hard instance in Theorem 5.2.

We consider the set of task distributions indexed by $\ell^*$ that randomize the reward-granting exit. Knowing the hierarchy amounts to knowing $\ell^*$, reducing the exploration problem to determining the correct exit. However, a hierarchy-oblivious learner needs to explore the tree, leading to regret that is exponential in the tree depth. Formally, we have the following result:

**Theorem 5.2.** *There exists a family of task distributions such that any hierarchy-oblivious learner incurs expected regret lower bounded by $\Omega(2^{W/2}\sqrt{H^2 N})$ on at least one task distribution. In contrast, a learner with access to a $0$-suboptimal[4] hierarchy oracle incurs regret bounded by $O(\sqrt{H^2 N})$ with high probability, over any sampled task from any of the task distributions.*

We prove this result in Section B.4.3, using recent results by Domingues et al. (2021) which demonstrate that the set of binary tree subproblems above form a set of minimax instances for any RL algorithm. This separation result suggests that hierarchy-based learners gain in situations where temporally extended exploratory behaviors are needed. This corroborates the experimental findings of Nachum et al. (2019b), which attributes the benefits of hierarchical RL to improved exploration.

## 6 RELATED WORK

Hierarchical reinforcement learning has been studied extensively (Sutton et al., 1999; Parr & Russell, 1998; Dieterich et al., 1998; Vezhnevets et al., 2017). An early approach formalizing the use of hierarchies in RL is the options framework (Sutton et al., 1999), which fixes a finite set of available skills/options. Since then, a large body of work has focused on designing methods for learning and adapting these options during the learning process (McGovern & Barto, 2001; Menache et al., 2002; Şimşek & Barto, 2004; Mann et al., 2014). Of particular note is the work of Frans et al. (2018), which learns a finite set of neural network sub-skills in the meta-RL setting. On the other hand, Laplacian-based option discovery in Machado et al. (2017; 2018) defines options using proto-value functions (Mahadevan, 2005), which captures global features of the state space. In more theoretical directions, Fruit & Lazaric (2017); Brunskill & Li (2014) provide regret and sample complexity bounds, respectively, for learning with options. Additionally, Mann & Mannor (2014) demonstrate that options can improve the convergence rate of approximate value iteration.

More recent empirical work has considered the problem of hierarchy learning beyond the options framework in a wide variety of settings. Nachum et al. (2019a); Levy et al. (2018) provide algorithms for learning hierarchies based on goal-conditioned policies, reducing the learning problem to choosing subgoals. Nachum et al. (2018) considers a more general case when learned representations are used to map states to goals. Other works such as Co-Reyes et al. (2018); Eysenbach et al. (2018); Sharma et al. (2019) provide intrinsic objectives for learning hierarchies without rewards.

Closely related is the work of Wen et al. (2020), which introduces a similar latent structure on the state space, and provides sufficient conditions ensuring sample-efficient and tractable hierarchy-based learning. However, their works hinges on prior knowledge of the latent structure, while a major focus of our work is discovering the structure itself from interactions.

Several heuristics have been proposed in prior work for the detection of useful bottlenecks in MDPs (McGovern & Barto, 2001; Menache et al., 2002; Şimşek & Barto, 2004; Şimşek et al., 2005). In particular, Şimşek & Barto (2004) define the notion of *access states*, states which maximize short-term novelty of future states. Although exits as defined in Definition 3.1 meet this heuristic, we note that cluster interiors may also contain bottlenecks also meet these conditions. In contrast, our algorithm would not detect these bottlenecks; however, this behavior is desirable as such bottlenecks are unimportant for meta-learning.

## 7 CONCLUSION

We have demonstrated that certain natural coverage conditions allow for learning useful hierarchies from tasks. Interesting future directions include analyzing hierarchy-based multi-task RL and extending the ideas in this work to continuous state and/or action spaces. Another interesting direction would be to provide sample-efficient algorithms for learning additional structures that can be imposed on the learned hierarchy, such as cluster equivalences as in Wen et al. (2020).

---

[4]We use a 0-suboptimal hierarchy oracle for the separation result for ease of presentation.

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

# A    META-TRAINING PROOFS

## A.1    ALGORITHM

In this section, we provide the complete algorithm for exit detection with optimistic imagination. For readability, we separate the three phases.

### A.1.1    PHASE I: TASK-SPECIFIC LEARNING

---

**Algorithm 1** Exit Detection, Phase I: Task-Specific Learning

---

**Require:** Tasks $\mathcal{M}_1, \ldots, \mathcal{M}_T$, $N_{\text{UCBVI}}$ UCBVI iterations, $N_{\text{TS}}$ policy samples, threshold $N_{\text{thresh}}^{\text{TS}}$

1: **for all** $t \in [T]$ **do**
2:     $\mathcal{D}^t \leftarrow \varnothing$.
3:     Obtain policy set $\Phi^t \leftarrow \text{UCBVI}(\mathcal{M}_t, N_{\text{UCBVI}})$.
4:     **for all** $n = 1, \ldots, N_{\text{TS}}$ **do**
5:         Sample $\pi \sim \text{Unif}(\Phi^t)$.
6:         Play $\pi$ in $\mathcal{M}_t$, add all $(s, a, s')$ pairs to $\mathcal{D}^t$, get sum of rewards $\hat{V}^{(n)}$.
7:     Form estimated dynamics model $\hat{\mathbb{P}}_t$ from $\mathcal{D}_t$.
8:     Form optimal value estimate $\hat{V}_t \leftarrow \dfrac{1}{N_{\text{TS}}} \displaystyle\sum_{n=1}^{N_{\text{TS}}} \hat{V}^{(n)}$
9:     $N_t(s, a) \leftarrow |\{(x, u, x') \in \mathcal{D}_t \mid x = s, u = a\}|$.
10:     **for all** $(s, a) \in \mathcal{S} \times \mathcal{A}$ **do**
11:         **if** $N_t(s, a) < N_{\text{thresh}}^{\text{TS}}$ **then** $\hat{\mathbb{P}}_t(\cdot \mid s, a) \leftarrow 0$.
12: **return** dynamics estimates $\hat{\mathbb{P}}_t$ and value estimates $\hat{V}_t$ for $t \in [T]$.

---

### A.1.2    PHASE II: LEARNING REFERENCE DYNAMICS

---

**Algorithm 2** Exit Detection, Phase II: Learning Reference Dynamics

---

**Require:** MDP $\mathcal{M}$, $N_{\text{EULER}}^{\text{RF}}$ Euler iterations, $N_{\text{RF}}$ policy samples

1: Set policy class $\Psi \leftarrow \varnothing$ and dataset $\mathcal{D}_{\text{RF}} \leftarrow \varnothing$
2: **for all** $g \in \mathcal{S}$ **do**
3:     Create MDP $\mathcal{M}_g$ from $\mathcal{M}$ with horizon $2H$ and $P(\ominus \mid g, a) = 1$ for any $a$.
4:     $r_g(s, a) \leftarrow \mathbb{1}[s = g]$ for any $(s, a) \in (\mathcal{S} \cup \{\ominus\}) \times \mathcal{A}$.
5:     $\Phi^g \leftarrow \text{EULER}(\mathcal{M}_g, r_g, N_{\text{RF}}^{\text{EULER}})$
6:     $\pi_h(\cdot \mid g) \leftarrow \text{Unif}(\mathcal{A})$ for $\pi \in \Phi^g$, $h \in [H]$.
7:     Add policies in $\Phi^g$ to $\Psi$.
8: **for all** $n = 1, \ldots, N_{\text{RF}}$ **do**
9:     Sample $\pi \sim \text{Unif}(\Psi)$.
10:     Play $\pi$ in $\mathcal{M}$ and obtain trajectory $(s_0, a_0, \ldots, s_{2H})$.
11:     Sample $h \sim \text{Unif}([2H])$ and add $(s_h, a_h, s_{h+1})$ to $\mathcal{D}_{\text{RF}}$.
12: **return** reference dynamics $\hat{\mathbb{P}}_0$ formed from $\mathcal{D}_{\text{RF}}$.

---

### A.1.3 PHASE III: EXIT DETECTION

---

**Algorithm 3** Exit Detection, Phase III: Exit Detection

---

**Require:** $N_{\text{ED}}, N_{\text{thresh}}^{\text{ED}}, N_{\text{EULER}}^{\text{EL}}, N_{\text{EL}}$ policy samples
1: Initialize $\text{ISEXIT}[s, a] \leftarrow \text{FALSE}$ for $(s, a) \in \mathcal{S} \times \mathcal{A}$.
2: **while** True **do**
3:     **for all** $t \in [T]$ **do**
4:         $\hat{\mathbb{P}}_0(\cdot \mid s, a) \leftarrow \hat{\mathbb{P}}_t(\cdot \mid s, a)$ for $(s, a) \in \mathcal{S} \times \mathcal{A}$ with $\text{ISEXIT}[s, a]$.
5:         $\tilde{V}^t, \tilde{Q}^t \leftarrow \text{OPTIMGVI}(\hat{\mathbb{P}}_0, (\hat{\mathbb{P}}_1, \ldots, \hat{\mathbb{P}}_T), r_t, \text{ISEXIT})$
6:         **if** $\tilde{V}_0^t(s_0) - \hat{V}_t > (2/3)\zeta$ **then**
7:             Run greedy policy with respect to $\tilde{Q}$ $N_{\text{ED}}$ times and form estimate $\hat{\mathbb{P}}$ for $(s, a)$ pairs
    visited at least $N_{\text{thresh}}'$ times.
8:             **for all** $(s, a) \in \mathcal{S} \times \mathcal{A}$ with $\hat{\mathbb{P}}(\cdot \mid s, a) \not\equiv 0$ **do**
9:                 **if** $(\exists t \in [T]) \, \hat{\mathbb{P}}_t(\cdot \mid s, a) \not\equiv 0$ and $\left\| \hat{\mathbb{P}}(\cdot \mid s, a) - \hat{\mathbb{P}}_t(\cdot \mid s, a) \right\|_{\text{TV}} > \beta/2$ **then**
10:                     $\text{ISEXIT}[s, a] \leftarrow \text{True}$
11:                     $\hat{\mathbb{P}}_t(\cdot \mid s, a) \leftarrow \text{LEARN-EXIT}(\mathcal{M}_t, (s, a), N_{\text{EULER}}^{\text{EL}}, N_{\text{EL}})$ **for all** $t \in [T]$
12:     **if** no new exits found after passing through $T$ tasks since last found exit **then**
13:         **return** $\text{ISEXIT}$

---

**Algorithm 4** **B**orrowing **O**ptimistically **A**cross **T**asks during **V**alue **I**teration (BOAT-VI)

---

1: **procedure** BOAT-VI(Reference dynamics $\hat{\mathbb{P}}_0$, Estimated dynamics $(\hat{\mathbb{P}}_1, \ldots, \hat{\mathbb{P}}_T)$,
                      Reward function $r$, Table $\text{ISEXIT}[\mathcal{S} \times \mathcal{A}]$)
2:     $\hat{V}_H(s) \leftarrow 0$ for $s \in \mathcal{S}$.
3:     **for all** $h = H - 1, \ldots, 0$ **do**
4:         **for all** $(s, a) \in \mathcal{S} \times \mathcal{A}$ **do**
5:             **if** $\text{ISEXIT}[s, a]$ **then**
6:                 $\hat{Q}_h(s, a) \leftarrow r(s, a) + \hat{\mathbb{P}}_0 \hat{V}_{h+1}(s, a)$
7:             **else**
8:                 $\hat{Q}_h(s, a) \leftarrow r(s, a) + \max_{t=0, \ldots, T} \hat{\mathbb{P}}_t \hat{V}_{h+1}(s, a)$
9:         $\hat{V}_h(s) \leftarrow \max_a \hat{Q}_h(s, a)$ for $s \in \mathcal{S}$.
10:     **return** $\hat{V}, \hat{Q}$

---

**Algorithm 5** Exit-learning subroutine

---

1: **procedure** LEARN-EXIT(MDP $\mathcal{M}$, exit $(s, a)$, $N_{\text{EULER}}^{\text{EL}}$ EULER iterations, $N_{\text{EL}}$ policy samples)
2:     Create MDP $\tilde{\mathcal{M}}$ from $\mathcal{M}$ so $P(\ominus \mid s, a) = 1$.
3:     $\tilde{r}(s', a') \leftarrow \mathbb{1}[(s', a') = (s, a)]$ for any $(s', a') \in (\mathcal{S} \cup \{\ominus\}) \times \mathcal{A}$.
4:     $\Psi \leftarrow \text{EULER}(\tilde{\mathcal{M}}, \tilde{r}, N_{\text{EULER}}^{\text{EL}})$
5:     **for all** $n = 1, \ldots, N_{\text{EL}}$ **do**
6:         Sample $\pi \sim \text{Unif}(\Psi)$.
7:         Play $\pi$ in $\mathcal{M}$ and obtain trajectory $(s_0, a_0, \ldots, s_H)$.
8:     **return** reference dynamics $\hat{\mathbb{P}}(\cdot \mid s, a)$ formed from all trajectory data.

---

## A.2 OTHER ASSUMPTIONS AND RELEVANT DEFINITIONS

The remaining assumptions quantify the reachability of certain states. First, we have the following assumption, which in effect ensures that one can reach most states regardless of exit configuration in the meta-training tasks:

**Assumption A.1** (Non-limiting exit configurations). *Let $\mathcal{M} = (\mathcal{S}, \mathcal{A}, \mathbb{P}, H)$ be any reward-free environment with time-varying dynamics*

$$\mathbb{P}^{(h)}(\cdot \mid s, a) = \mathbb{P}_{t(h,s,a)}(\cdot \mid s, a) \quad \text{for some } t : [H] \times \mathcal{S} \times \mathcal{A} \to [T].$$

*Then, there exists $C > 1$ such that for any $s$ and $t \in [T]$,*

$$\max_{\pi} P_{\mathcal{M}}(s \in \tau_\pi) \le C \max_{\pi} P_{\mathcal{M}_t}(s \in \tau_\pi)$$

Intuitively, the assumption states that the reachability of a state in $\mathcal{M}_t$ would not be significantly improved even under an optimal configuration of the exits. Therefore, running reward-free RL on one of the meta-training tasks is sufficient for learning all non-exit $(s, a)$ pairs.

**Remark A.1.** We note that Assumption A.1 is restrictive in that it requires that every state be roughly reachable in any of the meta-training MDPs. This may not hold in practice, e.g., consider a four-room environment where one of the rooms is blocked off for one of the tasks. However, this can be weakened to requiring that $s$ be reachable in at least one of $N$ arbitrarily chosen meta-training tasks. This would require that the algorithm run Phase II over $N$ meta-training tasks, which results in a benign increase in the query complexity of the algorithm, so long as $N$ is a constant much smaller than $T$. We focus on the $N = 1$ case for ease of presentation. ⌟

To simplify the presentation of the rest of the assumptions, we recall the following definition of $\delta$-significance in Jin et al. (2020):

**Definition A.1.** A state $s$ is $\delta$-*significant* if $\max_\pi P(s \in \tau_\pi) \ge \delta$. Additionally, we say that $(s, a)$ is $\delta$-significant if $s$ is $\delta$-significant. ◇

Note that we have modified the definition to remove the dependence on the timestep $h \in [H]$. This is because the dynamics are stationary, and thus it does not matter when $s$ is visited in a trajectory.

Note that Assumption A.1 implies that a reachable entrance in one task is reachable in all tasks (i.e., is significant in the sense of Definition A.1). Therefore, we can quantify the minimum level of significance:[5]

**Definition A.2** ($\rho$-significant entrances). For any $s \in \text{Ent}(\mathcal{S})$, $s$ is $\rho$-significant for any task. ◇

Finally, we want to quantify the reachability of every exit from any entrance in the same cluster, assuming that it is indeed reachable:

**Definition A.3** (In-cluster exit reachability). Fix any cluster $Z$, entry $s \in \text{Ent}(Z)$, and exit $(g, a) \in \text{Ext}(Z)$. Consider the reward-free environment $\mathcal{M}_t|_Z = (Z, A, \mathbb{P}_t|_Z, H)$, where $\mathbb{P}_t|_Z$ is the restriction of $\mathbb{P}_t$ to $Z \times \mathcal{A}$ and the starting state is $s$. Then, if $g$ has nonzero significance in $\mathcal{M}_t|_Z$ for any $t \in [T]$, then it is $\varepsilon_0$-significant. ◇

The requirement that the assumption hold for any $t \in [T]$ is without loss of generality since non-exit dynamics do not change.

## A.3 VERIFYING EXIT DETECTION

In this section, we demonstrate that the algorithm in Section A.1 can successfully discover $\text{Ext}(\mathcal{S})$ with high probability. Formally, we have the following result:

**Theorem A.1** (Provable exit detection). *Assume we run the algorithm in Section A.1 with the parameter choices given in Table 1. Then, with probability at least $1 - p$, the algorithm returns an array* ISEXIT *satisfying:*

$$\{(s, a) \mid \text{ISEXIT}[s, a]\} = \text{Ext}(\mathcal{S}).$$

To prove this result, we proceed with a phase-by-phase analysis of the algorithm in Section A.1, which we then compile into proof of the desired result.

---

[5]One can weaken this assumption in a way that is compatible with the weakened form of Assumption A.1.

| Parameter | Value |
|---|---|
| $N_{\text{UCBVI}}$ | $\dfrac{H^2 SA}{\min(\alpha, \zeta)^2} \log^2 \dfrac{HSAT}{p}$ |
| $N_{\text{thresh}}^{\text{TS}}$ | $S \max\left(\dfrac{H^4}{\zeta^2}, \dfrac{1}{\beta^2}\right) \log \dfrac{SAHT}{p\alpha \min(\beta, \zeta)}$ |
| $N_{\text{TS}}$ | $S \max\left(\dfrac{H^5}{\alpha \zeta^2}, \dfrac{H}{\alpha \beta^2}\right) \log \dfrac{SAHT}{p\alpha \min(\beta, \zeta)} + \dfrac{H^2}{\min(\alpha, \zeta)^2} \log \dfrac{SAT}{p}$ |
| $N_{\text{EULER}}^{\text{RF}}$ | $\dfrac{H^2 S^4 A}{\min(\rho \min(\varepsilon, \varepsilon_0), \zeta/C)} \log^3 \dfrac{HSA}{p}$ |
| $N_{\text{RF}}$ | $\dfrac{H^5 S^2 A}{\min(\rho \min(\varepsilon, \varepsilon_0)^2, \zeta^2/C)} \log \dfrac{A}{p}$ |
| $N_{\text{thresh}}^{\text{ED}}$ | $\dfrac{S}{\beta^2} \log \dfrac{SAH}{p\zeta\beta}$ |
| $N_{\text{ED}}$ | $\dfrac{HKS}{\zeta \beta^2} \log \dfrac{SAH}{p\zeta\beta} + \dfrac{H^2 K^2}{\zeta^2} \log \dfrac{K}{p}$ |
| $N_{\text{thresh}}^{\text{EL}}$ | $L \max\left(\dfrac{H^4}{\zeta^2}, \dfrac{1}{\beta^2}\right) \log \dfrac{CSAHT}{p\alpha \min(\beta, \zeta)}$ |
| $N_{\text{EULER}}^{\text{EL}}$ | $\dfrac{CH^3 S^2 A}{\alpha} \log^3 \left(\dfrac{HSAT}{p}\right)$ |
| $N_{\text{EL}}$ | $L \max\left(\dfrac{CH^5}{\alpha \zeta^2}, \dfrac{CH}{\alpha \beta^2}\right) \log \dfrac{CSAHT}{p\alpha \min(\beta, \zeta)} + \dfrac{C^2 H^2}{\alpha^2} \log \dfrac{SAT}{p}$ |

Table 1: Table of parameters for the results in Theorem A.1. Since $K \leq SA$ and $L \leq S$, the agent does not need to know $K$ or $L$ in advance, at the expense of a worse sample complexity bound.

### A.3.1   PHASE I ANALYSIS

First, we prove that during Phase I, Algorithm 1 sufficiently visits all relevant exits and that all value estimates are sufficiently close. Formally, we have the following result:

**Proposition A.1.** *Set*

$$N_{\text{thresh}}^{\text{TS}} = \Omega \left[ S \max\left(\frac{H^4}{\zeta^2}, \frac{1}{\beta^2}\right) \log \frac{SAHTN_{\text{TS}}}{p} \right],$$

*and consider the following procedure applied to one of the meta-training tasks $\mathcal{M}_t$:*

1. UCBVI *is run for*

$$N_{\text{UCBVI}} = \Omega \left( \frac{H^2 SA}{\min(\alpha, \zeta)^2} \log^2 \frac{HSAT}{p} \right)$$

*iterations, generating policies $\pi_1^{(t)}, \ldots, \pi_{N_{\text{UCBVI}}}^{(t)}$.*

2. *The learner uniformly samples*

$$N_{\text{TS}} = \Omega \left( \frac{H}{\alpha} N_{\text{thresh}}^{\text{TS}} + \frac{H^2}{\min(\alpha, \zeta)^2} \log \frac{TK}{p} \right)$$

*policies from the previous step, runs each policy in $\mathcal{M}_t$, and obtains a dataset of transitions $\mathcal{D}_t$ and returns $\hat{V}^{(1)}, \dots, \hat{V}^{(N_{\text{TS}})}$.*

*Then, with probability at least $1 - p/3T$,*

(a) *We have the regret bound*

$$V_0^*(s_0) - \frac{1}{N_{\text{UCBVI}}} \sum_{k=1}^{N_{\text{UCBVI}}} V_0^{\pi_k^{(t)}}(s_0) < \frac{\zeta}{6}.$$

(b) *The set of obtained returns satisfy*

$$\left| \frac{1}{N_{\text{TS}}} \sum_{i=1}^{N_{\text{TS}}} \hat{V}^{(i)} - \frac{1}{N} \sum_{k=1}^{N_{\text{UCBVI}}} V_0^{\pi_k^{(t)}}(s_0) \right| < \frac{\zeta}{6}.$$

(c) *If $(s, a)$ is $\alpha$-important for $\mathcal{M}_t$, then $N_t(s, a) \geq N_{\text{thresh}}$.*

(d) *For every $(s, a)$ pair such that $N_t(s, a) \geq N_{\text{thresh}}$,*

$$\sup_{f : \mathcal{S} \to [0, H]} \left| \left[ (\hat{\mathbb{P}}_t - \mathbb{P}_t) f \right] (s, a) \right| < \min \left( \frac{\zeta}{24H}, \frac{\beta H}{2} \right).$$

To prove the above result, we first recall the following regret bound on UCBVI, as proven by Azar et al. (2017):

**Lemma A.1** (UCBVI regret bound)**.** *For sufficiently large $N$, with probability at least $1 - p/6$,*

$$V_0^*(s_0) - \frac{1}{N} \sum_{k=1}^{N} V_0^{\pi_k}(s_0) \lesssim \sqrt{\frac{H^2 S A}{N}} \log \left( \frac{H S A N}{p} \right).$$

As we will see later on, with our choice of $N_{\text{UCBVI}}$, we obtain the desired regret bound in (a). Additionally, by Hoeffding's inequality, the average of $N_{\text{TS}}$ returns concentrates around the desired quantity with high probability, proving (b). Thus, all that remains is ensuring that every $\alpha$-important exit is sufficiently visited, and thus their dynamics are sufficiently well-estimated.

Recall from the main text that the key step is demonstrating that a near-optimal policy for a task must visit its $\alpha$-important states with non-negligible probability:

**Lemma A.2.** *Let $(s, a)$ be $\alpha$-important for $\mathcal{M}$, and let $\pi$ be an $\varepsilon$-suboptimal policy for $\varepsilon < \alpha$. Then,*

$$P((s, a) \in \tau_\pi) > \frac{1}{H}(\alpha - \varepsilon).$$

*Proof.* By $\alpha$-importance,

$$\alpha \leq V_0^{\mathcal{M},*}(s_0) - V_0^{\mathcal{M} \setminus (s,a),*}(s_0) \leq \left[ V_0^{\mathcal{M},*}(s_0) - V_0^{\mathcal{M},\pi}(s_0) \right] + \left[ V_0^{\mathcal{M},\pi}(s_0) - V_0^{\mathcal{M} \setminus (s,a),*}(s_0) \right]$$

$$\leq \left[ V_0^{\mathcal{M},\pi}(s_0) - V_0^{\mathcal{M} \setminus (s,a),*}(s_0) \right] + \varepsilon.$$

Therefore, by applying Lemma A.17 and noting that $\{\Delta \cap \tau_\pi \neq \varnothing\} = \{(s, a) \in \tau_\pi\}$, we obtain the desired result. $\qquad\square$

Through the prior result, we can relate the UCBVI regret bound to the probability that a randomly chosen UCBVI-generated policy visits an $\alpha$-important state:

**Lemma A.3.** *Let $(s, a)$ be $\alpha$-important for $\mathcal{M}$. Assume that* UCBVI, *when run for*

$$N_{\text{UCBVI}} = \Omega \left( \frac{H^2 SA}{\alpha^2} \log^2 \frac{HSA}{p} \right)$$

*iterations, generates policies $\pi_1, \ldots, \pi_N$. If we sample $\pi$ uniformly from these policies and let $\tau$ be the (random) trajectory generated by this randomly selected policy, then*

$$P((s, a) \in \tau) > \frac{\alpha}{2H},$$

*conditioned on the high probability event in Lemma A.1.*

*Proof.* By Lemma A.2, for any fixed $\pi$, we can write

$$P((s, a) \in \tau_\pi) \geq \frac{1}{H}(\alpha - [V_0^*(s_0) - V_0^\pi(s_0)])_+,$$

where $x_+ = x \mathbb{1}[x > 0]$. Then, since $\pi$ is chosen randomly from the policies generated by UCBVI,

$$P((s, a) \in \tau) = \frac{1}{N} \sum_{k=1}^{N} P((s, a) \in \tau_{\pi_k}) \geq \frac{1}{HN} \sum_{k=1}^{N} (\alpha - [V_0^*(s_0) - V_0^{\pi_k}(s_0)])_+$$

$$\geq \frac{1}{H} \left[ \alpha - \frac{1}{N} \sum_{k=1}^{N} V_0^*(s_0) - V_0^{\pi_k}(s_0) \right].$$

Therefore, by applying the regret bound in Lemma A.1 and the choice of $N$, we find that

$$P((s, a) \in \tau) > \frac{\alpha}{2H}. \qquad \square$$

With all of the above intermediate results, we can now prove Proposition A.1.

*Proof of Proposition A.1.* Throughout this proof, we condition on the high-probability event in Lemma A.1, instantiated to occur with probability at least $p/12T$.

(a) By the choice of $N_{\text{UCBVI}}$,

$$N_{\text{UCBVI}} \gtrsim \frac{H^2 SA}{\zeta^2} \log^2 \frac{HSAT}{p},$$

and thus, we obtain the desired bound by plugging this value into the regret bound provided by Lemma A.1.

(b) Note that $(\hat{V}^{(i)})$ are i.i.d., bounded in $[0, H]$, and for any $i \in [N_{\text{TS}}]$,

$$\mathbb{E}\left[\hat{V}^{(i)}\right] = \frac{1}{N} \sum_{k=1}^{N} V_0^{\pi_k^{(t)}}(s_0).$$

Therefore, by applying Hoeffding's inequality, with probability at least $1 - p/12T$,

$$\left| \frac{1}{N_{\text{TS}}} \sum_{i=1}^{N_{\text{TS}}} \hat{V}^{(i)} - \frac{1}{N_{\text{UCBVI}}} \sum_{k=1}^{N_{\text{UCBVI}}} V_0^{\pi_k^{(t)}}(s_0) \right| \lesssim \sqrt{\frac{H^2}{N_{\text{TS}}} \log \frac{T}{p}}$$

The result immediately follows from the fact that $N_{\text{TS}} \gtrsim (H^2/\zeta^2) \log(T/p)$.

(c) The result simply follows from Lemma A.16 instantiated with failure probability $1 - p/12T$, together with the choice of $N_{\text{thresh}}^{\text{TS}}$.

(d) With the choice of $N_{\text{UCBVI}}$, the conclusion of Lemma A.1 can be made to hold with probability at least $1 - p/24T$. Fix an $\alpha$-important exit $(s, a)$ for $\mathcal{M}_t$, so that the probability that $(s, a)$ is visited by the procedure is at least $\alpha/2H$. By Lemma A.15, sampling $N_{\text{TS}}$ trajectories is sufficient to ensure that $N_t(s, a) \geq N_{\text{thresh}}^{\text{TS}}$ with probability at least $1 - p/24TK$. Therefore, by performing a union bound over the set of $\alpha$-important exits (which contains at most $K$ elements), $N_t(s, a) \geq N_{\text{thresh}}^{\text{TS}}$ for any $\alpha$-important exit with probability at least $1 - p/24T$. Thus, overall, this event occurs with probability at least $1 - p/12T$.

Since each part fails with probability at most $p/12T$, the overall failure probability is at most $p/3T$, the desired result. $\qquad \square$

A.3.2 PHASE II ANALYSIS

In this section, we provide guarantees on the dataset $\mathcal{D}_{\mathrm{RF}}$ obtained by performing reward-free RL in Algorithm 2. Formally, we have the following high-probability result:

**Proposition A.2.** *For any $\delta > 0$ and failure probability $p$, if Algorithm 2 is run with parameters*

$$N_{\mathrm{EULER}}^{\mathrm{RF}} = O\left(\frac{H^2 S^2 A}{\delta} \log^3 \frac{HSA}{p}\right)$$

$$N_{\mathrm{RF}} = O\left[\max\left(\frac{C}{\zeta^2}, \frac{1}{\rho \min(\varepsilon, \varepsilon_0)^2}\right) H^5 S^2 A \log \frac{A}{p}\right].$$

*Then, with probability at least $1 - p/3$:*

*(a) The distribution $\mu$ generating each sample in $\mathcal{D}_{\mathrm{RF}}$ satisfies*

$$s \in \mathcal{S} \text{ is } \delta\text{-significant in } \mathcal{M}_1(2H) \implies \max_{a,\pi} \frac{P((s,a) \in \tau_\pi)}{\mu(s,a)} \leq 4SAH.$$

*(b) The estimated dynamics model $\hat{\mathbb{P}}_0$ satisfies*

$$\max_{f:\mathcal{S}\to[0,H]} \max_{\nu:\mathcal{S}\to\mathcal{A}} \mathbb{E}_{(s,a)\sim\mu}\left[\left|\left|\left[(\hat{\mathbb{P}} - \mathbb{P})f\right](s,a)\right|^2 \mathbb{1}\left[a = \nu(s)\right]\right]\right.$$

$$\lesssim \min\left(\frac{\zeta^2}{4 \cdot 24^2 C}, \frac{\rho \min(\varepsilon, \varepsilon_0)^2}{16}\right) \frac{1}{H^3 SA}.$$

The details of the proof of Proposition A.2 follow that of Jin et al. (2020), which we provide here for completeness. First, we adapt the regret bound from Zanette & Brunskill (2019) for any MDP and reward function used in Algorithm 2.

**Lemma A.4.** *For any $g \in \mathcal{S}$, running* EULER *in $\mathcal{M}_g$ for $N$ iterations returns $N$ policies $\pi_1, \ldots, \pi_N$ satisfying the regret bound*

$$V_0^*(s_0) - \frac{1}{N}\sum_{k=1}^N V_0^{\pi_k}(s_0) \lesssim \sqrt{4V_0^*(s_0)\frac{SA}{N}\log\frac{SAHN}{p}} + \frac{S^2AH^2}{N}\log^3\frac{SAHN}{p}$$

*with probability at least $1 - p$.*

*Proof.* Observe that

$$\frac{1}{NH}\sum_{k=1}^N \mathbb{E}_{\pi_k}\left[\left(\sum_{h=1}^{H-1} r(s_h, a_h) - V_0^{\pi_k}(s_0)\right)^2 \Bigg| s_0\right]$$

$$\leq \frac{2}{NH}\sum_{k=1}^N \mathbb{E}_{\pi_k}\left[\left(\sum_{h=1}^{H-1} r(s_h, a_h)\right)^2 + (V_0^{\pi_k}(s_0))^2 \Bigg| s_0\right]$$

$$\leq \frac{2}{NH}\sum_{k=1}^N \mathbb{E}_{\pi_k}\left[\sum_{h=1}^{H-1} r(s_h, a_h) + V_0^{\pi_k}(s_0) \Bigg| s_0\right]$$

$$\leq \frac{4}{H}V_0^*(s_0).$$

Therefore, by applying the regret bounds from Zanette & Brunskill (2019), we obtain the regret bound

$$V_0^*(s_0) - \frac{1}{N}\sum_{k=1}^N V_0^{\pi_k}(s_0) \lesssim \sqrt{4V_0^*(s_0)\frac{SA}{N}\log\frac{SAHN}{p}} + \frac{S^2A^2H^2}{N}\log^3\frac{SAHN}{p}$$

with probability at least $1 - p$. $\qquad\square$

With the regret bound above, we now proceed to prove Proposition A.2.

*Proof of Proposition A.2.* (a) Fix a $\delta$-significant $g \in \mathcal{S}$. Note that for $r_g$, $V_0^\pi(s_0) = P(g \in \tau_\pi)$ for any policy $\pi$. Therefore, via the regret bound from Lemma A.4 and the choice of $N_{\text{EULER}}^{\text{RF}}$, we obtain

$$\max_\pi P(g \in \tau_\pi) - \frac{1}{N_{\text{EULER}}^{\text{RF}}} \sum_{k=1}^{N_{\text{EULER}}^{\text{RF}}} P(g \in \tau_\pi) \leq \frac{1}{2} \max_\pi P(g \in \tau_\pi)$$

$$\implies \max_\pi P(g \in \tau_\pi) \leq \frac{2}{N_{\text{RF}}^{\text{EULER}}} \sum_{\pi \in \Phi_g} P(g \in \tau_\pi)$$

with probability at least $1 - p/2S$. Now, since $\pi(\cdot \mid g) \sim \text{Unif}(\mathcal{A})$, we have that for any $a$,

$$\max_\pi P((g, a) \in \tau_\pi) \leq \frac{2A}{N_{\text{RF}}^{\text{EULER}}} \sum_{\pi \in \Phi_g} P((g, a) \in \tau_\pi).$$

Finally, by applying the same argument above across all $\delta$-significant $g \in \mathcal{S}$, we have that for any $(g, a)$,

$$\max_\pi P((g, a) \in \tau_\pi) \leq \sum_{g \in S} \max_{a, \pi} P((g, a) \in \tau_\pi) \leq 2SA \left[ \frac{1}{SN_{\text{RF}}^{\text{EULER}}} \sum_{\pi \in \Psi} P((g, a) \in \tau_\pi) \right]$$

with probability at least $1 - p/2$. To complete the proof of (a), observe that

$$\frac{1}{SN_{\text{RF}}^{\text{EULER}}} \sum_{\pi \in \Psi} \frac{1}{2H} P((g, a) \in \tau_\pi) \leq \mu(s, a) \implies \max_{s, a, \pi} \frac{P((s, a) \in \tau_\pi)}{\mu(s, a)} \leq 4SAH,$$

since conditioned on $(g, a) \in \tau_\pi$, the probability that $(g, a)$ is sampled is at least $1/2H$.

(b) The result follows by following the same proof of Lemma C.2 in Jin et al. (2020), with failure probability $p/2$. Note that the dynamics are stationary, and thus we do not need to perform a union bound over the time step $h \in [H]$. $\qquad\square$

### A.3.3 PHASE III ANALYSIS

Having analyzed the previous two phases, we now show that Algorithm 3 successfully finds all exits during Phase III. As part of this, we prove the following guarantee:

**Proposition A.3.** *Assume that Algorithm 3 is at Line 3, having just arrived at this step for the first time, or after finding a new exit. Let $E = \{(s, a) \mid \text{ISEXIT}[s, a]\}$. We assume:*

    *(a) $E \subseteq \text{Ext}(\mathcal{S})$.*

    *(b) The high-probability events in Proposition A.1 (for any $t \in [T]$) and Proposition A.2 (for $\delta \leq \zeta/24CH^2S$) both hold, providing estimators $\hat{\mathbb{P}}_0, \hat{\mathbb{P}}_1, \ldots, \hat{\mathbb{P}}_T$.*

    *(c) For every $(s, a) \in E$ and $t \in [T]$, we have access to an estimator $\hat{\mathbb{P}}_t(\cdot \mid s, a)$ for $\mathbb{P}_t(\cdot \mid s, a)$ satisfying*

$$\max_{f: \mathcal{S} \to [0, H]} \left| \left[ (\hat{\mathbb{P}}_t - \mathbb{P}_t) f \right] (s, a) \right| \leq \min \left( \frac{\zeta}{24H}, \frac{\beta H}{2} \right).$$

*Then, if $E = \text{Ext}(\mathcal{S})$, the algorithm terminates after passing through $T$ tasks. Otherwise, if $E \neq \text{Ext}(\mathcal{S})$, the following events hold simultaneously with probability at least $1 - p/3K$:*

    *(a) For one of the next $T$ tasks that the algorithm inspects, there exists at least one $t \in [T]$ such that*

$$\left| \tilde{V}^t - \hat{V}_t \right| > \frac{2}{3} \zeta.$$

*(b) For the task in (a), running Lines 7–11 finds at least one $(s, a) \in \text{Ext}(\mathcal{S}) \setminus E$ (and only $(s, a)$ pairs in this set), and learns an estimator $\hat{\mathbb{P}}_t(\cdot \mid s, a)$ for $\mathbb{P}_t(\cdot \mid s, a)$ satisfying*

$$\max_{f:\mathcal{S}\to[0,H]} \left| \left[ (\hat{\mathbb{P}}_t - \mathbb{P}_t)f \right](s, a) \right| \leq \min\left( \frac{\zeta}{24H}, \frac{\beta H}{2} \right).$$

To prove the above result, we will consider the following special set of MDPs:

**Definition A.4** (Imaginable MDPs). Fix a task $\mathcal{M}_t$. Furthermore, let $E \subseteq \text{Ext}(\mathcal{S})$. For any function $\mathcal{I} : \mathcal{S} \times \mathcal{A} \times [H] \to \{0, \dots, T\}$, we can construct an associated MDP $\mathcal{M}_{\mathcal{I}} = (\mathcal{S}, \mathcal{A}, \mathbb{P}_{\mathcal{I}}, r_t, H)$ via

$$\mathbb{P}_{\mathcal{I}}^{(h)}(\cdot \mid s, a) = \mathbb{P}_{\mathcal{I}(s,a,h)}(\cdot \mid s, a).$$

We define the set of imaginable MDPs to be the set $\mathbb{M}_t(E)$ to be the set of MDPs generated by any $\mathcal{I}$ satisfying

$$\mathcal{I}(s, a, h) \in \begin{cases} \{t\} & (s, a) \in E \\ \{0\} \cup \{k \mid N^k(s, a) \geq N_{\text{thresh}}\} & \text{otherwise} \end{cases}. \qquad \diamond$$

Informally, $\mathbb{M}_t(E)$ is the set of obtainable MDPs by borrowing dynamics for $(s, a)$ pairs that are not known to be exits. This set is of particular interest in our analysis, since BOAT-VI performs a maximization over the MDPs in this set:

**Lemma A.5** (Optimism). *Assume the preconditions of Proposition A.3. Over the course of running Algorithm 4, the algorithm implicitly defines an index function $\mathcal{I} : \mathcal{S} \times \mathcal{A} \times [H] \to \{0, \dots, T\}$. This function $\mathcal{I}$ satisfies $\mathcal{M}_{\mathcal{I}} \in \mathbb{M}_t(E)$, and $\mathcal{M}_{\mathcal{I}}$ is a maximizer of*

$$\max_{\mathcal{M}\in\mathbb{M}_t(E)} \max_{\pi} \hat{V}_0^{\mathcal{M},\pi}(s_0).$$

*Proof.* To see that $\mathcal{M}_{\mathcal{I}} \in \mathbb{M}_t(E)$, note that if $(s, a) \in E$, then $\mathcal{I}_h(s, a) = t$ for any $h \in [H]$. Otherwise, note that although the maximum is over all indices, $\hat{\mathbb{P}}_k(s' \mid s, a) = 0$ for any $s'$ if $N_t(s, a) < N_{\text{thresh}}$. Therefore, since the estimated value function is always positive, the maximum is effectively only over any $k$ with $N_k(s, a) \geq 0$. Thus, $\mathcal{M}_{\mathcal{I}} \in \mathbb{M}_t(E)$.

Now, we prove that $\mathcal{M}_{\mathcal{I}} = \mathcal{M}$ is a maximizer of the estimated value function, which we prove by induction. Let $\mathcal{I}'$ be another index function satisfying $\mathcal{M}' = \mathcal{M}_{\mathcal{I}'} \in \mathbb{M}_t(E)$. Clearly, $\hat{V}_H^{\mathcal{M},*}(s) = 0 \leq \hat{V}_H^{\mathcal{M}',*}(s)$. Then, for any $h \in [H]$ and $(s, a)$,

$$\begin{aligned} \hat{Q}_h^{\mathcal{M},*}(s, a) &= r(s, a) + \hat{\mathbb{P}}_{\mathcal{I}(s,a,h)} \hat{V}_{h+1}^{\mathcal{M},*}(s, a) \\ &\geq r(s, a) + \hat{\mathbb{P}}_{\mathcal{I}'(s,a,h)} \hat{V}_{h+1}^{\mathcal{M},*}(s, a) \geq r(s, a) + \hat{\mathbb{P}}_{\mathcal{I}'(s,a,h)} \hat{V}_{h+1}^{\mathcal{M}',*}(s, a) \\ &= \hat{Q}_h^{\mathcal{M}',*}(s, a), \end{aligned}$$

where the first inequality follows from the definition of $\mathcal{I}$, and the second follows from the inductive hypothesis. Therefore, for any $s$,

$$\hat{V}_h^{\mathcal{M},*}(s) = \max_a \hat{Q}_h^{\mathcal{M},*}(s, a) \geq \max_a \hat{Q}_h^{\mathcal{M}',*}(s, a) = \hat{V}_h^{\mathcal{M}',*}(s)$$

Thus, by induction, $\hat{V}_0^{\mathcal{M},*}(s_0) \geq \hat{V}_0^{\mathcal{M}',*}(s_0)$. Since the argument applies for any $\mathcal{I}'$, we have shown the desired optimality result. $\qquad \square$

Note that $\bar{\mathcal{M}}$ is contained in $\mathbb{M}_t(E)$ via our assumed preconditions, suggesting that the BOAT-VI should find an MDP with a sufficiently over-optimistic value. However, the maximization above makes use of estimated dynamics, and thus we need to prove that every MDP in $\mathbb{M}_t(E)$ is sufficiently well-estimated. We now show that the preconditions of Proposition A.3 are sufficient for estimation. To this end, we recall the performance difference lemma:

**Lemma A.6** (Performance Difference). *Fix two MDPs $\mathcal{M} = (\mathcal{S}, \mathcal{A}, r, \mathbb{P}, H)$ and $\mathcal{M}' = (\mathcal{S}, \mathcal{A}, r, \mathbb{P}', H)$. Then, for any policy $\pi$,*

$$V_0^{\mathcal{M}',\pi}(s_0) - V_0^{\mathcal{M},\pi}(s_0) = \mathbb{E}_{\mathcal{M},\pi}\left[ \sum_{h=0}^{H-1} [(\mathbb{P}_h' - \mathbb{P}_h)\hat{V}_{h+1}](s_h, a_h) \mid s_0 \right].$$

We now present the estimation result:

**Lemma A.7.** *For any $\pi$ and $t \in [T]$, let $V_0^{\mathcal{M},\pi}(s_0)$ be the value of a policy $\pi$ in $\mathcal{M} \in \mathbb{M}_t(E)$, and $\hat{V}_0^{\mathcal{M},\pi}(s_0)$ an estimate using available quantities from the preconditions of Proposition A.3. Then,*

$$\sup_{t \in [T]} \sup_{\substack{\pi \\ \mathcal{M} \in \mathbb{M}_t(E)}} \left| \hat{V}_0^{\mathcal{M},\pi}(s_0) - V_0^{\mathcal{M},\pi}(s_0) \right| < \frac{\zeta}{6}.$$

*Proof.* Fix a $t \in [T]$, $\mathcal{M} \in \mathbb{M}_t(E)$ and policy $\pi$, with associated index function $\mathcal{I}$. Lemma A.6 implies that

$$\left| \hat{V}_0^{\mathcal{M},\pi}(s_0) - V_0^{\mathcal{M},\pi}(s_0) \right| \leq \sum_{h=0}^{H-1} \mathbb{E}_{\mathcal{M},\pi} \left[ \left| \left[ (\hat{\mathbb{P}}^{(h)} - \mathbb{P}^{(h)}) \hat{V}_{h+1}^\pi \right](s_h, a_h) \right| \right]$$

$$\leq \sum_{h=0}^{H-1} \sum_{(s,a)} \left| \left[ (\hat{\mathbb{P}}^{(h)} - \mathbb{P}^{(h)}) \hat{V}_{h+1}^\pi \right](s,a) \right| P_h^{\mathcal{M},\pi}(s,a).$$

We now define the following sets:

$$U_\delta = \{(s,a) \text{ is } \delta\text{-insignificant for } \mathcal{M}_1\}$$
$$B_h = \{(s,a) \mid \mathcal{I}_h(s,a) \neq 0\} \setminus (E \cup U_\delta)$$
$$R_h = \{(s,a) \mid \mathcal{I}_h(s,a) = 0\} \setminus U_\delta.$$

Note that $R_h$ is estimated via reference dynamics from Phase II, while $B_h$ is estimated using task-specific dynamics from Phase I. Then, for a fixed $h$, we can decompose the inner sum above as

$$\sum_{(s,a)} \left| \left[ (\hat{\mathbb{P}}^{(h)} - \mathbb{P}^{(h)}) \hat{V}_{h+1}^\pi \right](s,a) \right| P_h^{\mathcal{M},\pi}(s,a)$$

$$\leq \underbrace{\sum_{(s,a) \in E} \left| \left[ (\hat{\mathbb{P}}^{(h)} - \mathbb{P}^{(h)}) \hat{V}_{h+1}^\pi \right](s,a) \right| P_h^{\mathcal{M},\pi}(s,a)}_{=:(\text{I})}$$

$$+ \underbrace{\sum_{(s,a) \in R_h} \left| \left[ (\hat{\mathbb{P}}^{(h)} - \mathbb{P}^{(h)}) \hat{V}_{h+1}^\pi \right](s,a) \right| P_h^{\mathcal{M},\pi}(s,a)}_{=:(\text{II})}$$

$$+ \underbrace{\sum_{(s,a) \in B_h} \left| \left[ (\hat{\mathbb{P}}^{(h)} - \mathbb{P}^{(h)}) \hat{V}_{h+1}^\pi \right](s,a) \right| P_h^{\mathcal{M},\pi}(s,a)}_{=:(\text{III})}$$

$$+ \underbrace{\sum_{(s,a) \in U_\delta} \left| \left[ (\hat{\mathbb{P}}^{(h)} - \mathbb{P}^{(h)}) \hat{V}_{h+1}^\pi \right](s,a) \right| P_h^{\mathcal{M},\pi}(s,a)}_{=:(\text{IV})}$$

Note the inequality since $E \cap U_\delta$ is not necessarily disjoint. We now bound the four terms above separately.

**Bounding (I): Dynamics Error from Known Exits.** We first bound (I), which we note derives from errors in estimating the dynamics of known exits. Recall that by precondition (c) in Proposition A.3,

$$\sup_{f:\mathcal{S} \to [0,H]} \left| \left[ (\hat{\mathbb{P}}_t - \mathbb{P}_t) f \right](s,a) \right| \leq \frac{\zeta}{24H}.$$

Therefore,

$$(\text{I}) = \sum_{(s,a) \in E} \left| \left[ (\hat{\mathbb{P}}^{(h)} - \mathbb{P}^{(h)}) \hat{V}_{h+1}^\pi \right](s,a) \right| P_h^{\mathcal{M},\pi}(s,a)$$

$$= \sum_{(s,a) \in E} \left| \left[ (\hat{\mathbb{P}}_t - \mathbb{P}_t) \hat{V}_{h+1}^\pi \right](s,a) \right| P_h^{\mathcal{M},\pi}(s,a) \leq \frac{\zeta}{24H} \sum_{(s,a) \in E} P_h^{\mathcal{M},\pi}(s,a)$$

$$\leq \frac{\zeta}{24H}.$$

**Bounding (II): Reference Dynamics Error.** Note that within $R_h$, $\mathbb{P}_h = \mathbb{P}_0$, which we estimate via $\hat{\mathbb{P}}_0$. Therefore, we bound the error resulting from using $\mathcal{D}_{\mathrm{RF}}$ to estimate $\mathbb{P}_0$. This part of the proof follows that of Jin et al. (2020). First, by Cauchy-Schwarz,

$$
(\mathrm{II}) = \sum_{(s,a)\in R_h} \left| \left[ (\hat{\mathbb{P}}^{(h)} - \mathbb{P}^{(h)})\hat{V}_{h+1}^\pi \right](s,a) \right| P_h^{\mathcal{M},\pi}(s,a)
$$

$$
= \sum_{(s,a)\in R_h} \left| \left[ (\hat{\mathbb{P}}_0 - \mathbb{P}_0)\hat{V}_{h+1}^\pi \right](s,a) \right| P_h^{\mathcal{M},\pi}(s,a)
$$

$$
\leq \left[ \sum_{(s,a)\in R_h} \left| \left[ (\hat{\mathbb{P}}_0 - \mathbb{P}_0)\hat{V}_{h+1}^\pi \right](s,a) \right|^2 P_h^{\mathcal{M},\pi}(s,a) \right]^{1/2}.
$$

Observe that $\hat{V}_{h+1}^\pi$ only depends on $\pi$ through timesteps $h+1, \ldots, H-1$. Therefore,

$$
\sum_{(s,a)\in R_h} \left| \left[ (\hat{\mathbb{P}}_0 - \mathbb{P}_0)\hat{V}_{h+1}^\pi \right](s,a) \right|^2 P_h^{\mathcal{M},\pi}(s,a)
$$

$$
\leq \max_{\nu:\mathcal{S}\to\mathcal{A}} \sum_{(s,a)\in R_h} \left| \left[ (\hat{\mathbb{P}}_0 - \mathbb{P}_0)\hat{V}_{h+1}^\pi \right](s,a) \right|^2 P_h^{\mathcal{M},\pi}(s)\mathbb{1}\left[ \nu(s) = a \right].
$$

By applying Assumption A.1,

$$
P_h^{\mathcal{M},\pi}(s) \leq P^{\mathcal{M}}(s \in \tau_\pi) \leq \max_\pi P^{\mathcal{M}}(s \in \tau_\pi) \leq C \max_\pi P^{\mathcal{M}_1}(s \in \tau_\pi)
$$

$$
\leq C \max_\pi P^{\mathcal{M}_1(2H)}(s \in \tau_\pi) \leq 4CHSA\mu(s,a),
$$

where we have applied Assumption 4.1 to move from $\mathcal{M}$ to $\mathcal{M}_1$. Substituting into the earlier expression,

$$
\max_{\nu:\mathcal{S}\to\mathcal{A}} \sum_{(s,a)\in R_h} \left| \left[ (\hat{\mathbb{P}}_0 - \mathbb{P}_0)\hat{V}_{h+1}^\pi \right](s,a) \right|^2 P_h^{\mathcal{M},\pi}(s)\mathbb{1}\left[ \nu(s) = a \right]
$$

$$
\leq 4CHSA \max_{\nu:\mathcal{S}\to\mathcal{A}} \sum_{(s,a)\in R_h} \left| \left[ (\hat{\mathbb{P}}_0 - \mathbb{P}_0)\hat{V}_{h+1}^\pi \right](s,a) \right|^2 \mathbb{1}\left[ a = \nu(s) \right] \mu(s,a)
$$

$$
\leq 4CHSA \max_{\nu:\mathcal{S}\to\mathcal{A}} \sum_{s,a} \left| \left[ (\hat{\mathbb{P}}_0 - \mathbb{P}_0)\hat{V}_{h+1}^\pi \right](s,a) \right|^2 \mathbb{1}\left[ a = \nu(s) \right] \mu(s,a)
$$

$$
= 4CHSA \max_{\nu:\mathcal{S}\to\mathcal{A}} \mathbb{E}_{(s,a)\sim\mu}\left[ \left| \left[ (\hat{\mathbb{P}}_0 - \mathbb{P}_0)\hat{V}_{h+1}^\pi \right](s,a) \right|^2 \mathbb{1}\left[ a = \nu(s) \right] \right].
$$

Thus, by applying the bound on the right-hand side provided by Proposition A.2,

$$
(\mathrm{II}) = \sum_{(s,a)\in R_h} \left| \left[ (\hat{\mathbb{P}}^{(h)} - \mathbb{P}^{(h)})\hat{V}_{h+1}^\pi \right](s,a) \right| P_h^{\mathcal{M},\pi}(s,a) \leq \frac{\zeta}{24H}.
$$

**Bounding (III): Error from Task-Specific Dynamics.** Recall that on $B_h$, $\mathbb{P}_h = \mathbb{P}_k$ for some $k \neq 0$. Thus, (III) is the error resulting from dynamics estimation in Algorithm 1. By following the same argument as that used to bound (I) and applying Proposition A.1, we find that

$$
\sum_{(s,a)\in B_h} \left| \left[ (\hat{\mathbb{P}}^{(h)} - \mathbb{P}^{(h)})\hat{V}_{h+1}^\pi \right](s,a) \right| P_h^{\mathcal{M},\pi}(s,a) \leq \frac{\zeta}{24H}.
$$

**Bounding (IV): Error from $\delta$-Insignificance.** The remaining set of $(s,a)$ pairs are those such that $s$ is $\delta$-insignificant in $\mathcal{M}_1$. Note that

$$
(\mathrm{IV}) = \sum_{(s,a)\in U_\delta} \left| \left[ (\hat{\mathbb{P}}^{(h)} - \mathbb{P}^{(h)})\hat{V}_{h+1}^\pi \right](s,a) \right| P_h^{\mathcal{M},\pi}(s,a) \leq H \sum_{(s,a)\in U_\delta} P_h^{\mathcal{M},\pi}(s,a)
$$

$$
= H \sum_{s\in U_\delta} P_h^{\mathcal{M},\pi}(s).
$$

As a result,

$$P_h^{\mathcal{M},\pi}(s) \le P^{\mathcal{M}}(s \in \tau_\pi) \le \max_\pi P^{\mathcal{M}}(s \in \tau_\pi) \le C \max_\pi P^{\mathcal{M}_1}(s \in \tau_\pi)$$
$$\le C\delta.$$

By setting $\delta = \zeta/24CH^2S$ when performing reward-free RL in Phase II, we thus find that

$$\sum_{(s,a)\in U_\delta} \left| \left[ (\hat{\mathbb{P}}^{(h)} - \mathbb{P}^{(h)})\hat{V}_{h+1}^\pi \right](s,a) \right| P_h^{\mathcal{M},\pi}(s,a) \le \frac{\zeta}{24H}.$$

**Concluding**. By combining the bounds on (I) through (IV) and summing across $h = 0, \ldots, H - 1$, we find that

$$\left| \hat{V}_0^{\mathcal{M},\pi}(s_0) - V_0^{\mathcal{M},\pi}(s_0) \right| \le \frac{\zeta}{6}.$$

Note that this argument simultaneously applies to any such $\mathcal{M}$; therefore, the desired conclusion follows. $\qquad \square$

The prior estimation result, together with Assumption 4.1, suggests that BOAT-VI should find an MDP that sufficiently overestimates the value of the task so long as not all exits have been found. This ensures that the exit-finding routine is triggered. Formally,

**Lemma A.8.** *Assume the preconditions of Proposition A.3, and that $E \ne \text{Ext}(\mathcal{S})$. Additionally, let $t \in [T]$ be the task with a $\zeta$-overoptimistic value when borrowing exits $\text{Ext}(\mathcal{S}) \setminus E$. Finally, let $\tilde{V}^t$ be the value function returned by Algorithm 4 on $\mathcal{M}_t$. Then,*

$$\tilde{V}_0^t(s_0) - \hat{V}_t > \frac{2}{3}\zeta.$$

*Proof.* Throughout this proof, we omit the timestep $0$ and the initial state $s_0$ for brevity. Define $\mathcal{M}^*$ and $\pi^*$ to be the maximizers of

$$\max_{\mathcal{M}\in\mathbb{M}_t(E)} \max_\pi V^{\mathcal{M},\pi}.$$

Furthermore, let $\bar{\mathcal{M}}$ be the imagined MDP guaranteed by Assumption 4.1 on top of $\mathcal{M}_t$, such that $V^{\bar{\mathcal{M}},*} > V^{\mathcal{M}_t,*} + \zeta$. Then, we have that

$$\hat{V}^{\mathcal{M},\pi} - V^{\mathcal{M}_t,*} = \underbrace{(\hat{V}^{\mathcal{M},\pi} - \hat{V}^{\mathcal{M}^*,\pi^*})}_{\ge 0} + \underbrace{(V^{\hat{\mathcal{M}}^*,\pi^*} - V^{\mathcal{M}^*,\pi^*})}_{>-\zeta/6}$$
$$+ \underbrace{(V^{\mathcal{M}^*,\pi^*} - V^{\bar{\mathcal{M}},\bar{\pi}})}_{\ge 0} + \underbrace{(V^{\bar{\mathcal{M}},\bar{\pi}} - V^{\mathcal{M}_t,*})}_{>\zeta}$$
$$> \frac{5}{6}\zeta.$$

Furthermore,

$$V^{\mathcal{M}_t,*} - \hat{V}_t = \underbrace{\left( V^{\mathcal{M}_t,*} - \frac{1}{N_{\text{UCBVI}}} \sum_{k=1}^{N_{\text{UCBVI}}} V^{\mathcal{M}_t,\pi_k^{(t)}} \right)}_{\ge 0} + \underbrace{\left( \frac{1}{N_{\text{UCBVI}}} \sum_{k=1}^{N_{\text{UCBVI}}} V^{\mathcal{M}_t,\pi_k^{(t)}} - \hat{V}_t \right)}_{\ge -\zeta/6},$$

where the first follows from optimality, while the second follows from Proposition A.2. Thus, putting the two inequalities together,

$$\tilde{V}^t - \hat{V}_t > \frac{2}{3}\zeta. \qquad \square$$

While the prior algorithm ensures that at least one of the tasks will trigger the exit condition, the actual task that triggers the condition may not be the same one invoked in the proof above. Nevertheless, we can prove that the trigger condition ensures that the algorithm will find a new exit.

**Lemma A.9.** *Assume the preconditions of Proposition A.3, and that $E \neq \mathrm{Ext}(\mathcal{S})$. Let $t \in [T]$ be a task such that the value estimate $\tilde{V}^t$ returned by Algorithm 4 satisfies $\tilde{V}_0^t(s_0) - \hat{V}_t > (2/3)\zeta$, and let $\pi$ be the optimal policy for $\tilde{V}$. Then, there exists $(s,a) \in \mathrm{Ext}(\mathcal{S}) \setminus E$ such that for some $t' \neq t$,*

(a) $N_{t'}(s,a) \geq N_{\mathrm{thresh}}^{\mathrm{TS}}$ *and* $\mathbb{P}_t(\cdot \mid s, a) \neq \mathbb{P}_{t'}(\cdot \mid s, a)$.

(b) $P^{\mathcal{M}_t}((s,a) \in \tau_\pi) > \zeta/6KH$.

*Proof.* Let $\mathcal{M}$ be the implicit MDP defined by Algorithm 4 in the process of computing $\tilde{V}^t$. We will prove a value gap between $\mathcal{M}$ and $\mathcal{M}_t$, which implies that $\pi$ must visit state-action pairs with imagined dynamics.

Note that

$$
V_0^{\mathcal{M},\pi}(s_0) - V_0^{\mathcal{M}_t,*}(s_0) = \underbrace{\left[V_0^{\mathcal{M},\pi}(s_0) - \tilde{V}_0^t(s_0)\right]}_{\geq -\zeta/6} + \underbrace{\left[\tilde{V}_0^t(s_0) - \hat{V}_t\right]}_{\geq (2/3)\zeta}
$$
$$
+ \underbrace{\left[\hat{V}_t - \frac{1}{N_{\mathrm{UCBVI}}} \sum_{k=1}^{N_{\mathrm{UCBVI}}} V_0^{\mathcal{M}_t, \pi_k^{(t)}}(s_0)\right]}_{\geq -\zeta/6}
$$
$$
+ \underbrace{\left[\frac{1}{N_{\mathrm{UCBVI}}} \sum_{k=1}^{N_{\mathrm{UCBVI}}} V_0^{\mathcal{M}_t, \pi_k^{(t)}}(s_0) - V_0^{\mathcal{M}_t, *}(s_0)\right]}_{\geq -\zeta/6},
$$

where the first inequality comes from Lemma A.7 and the last two inequalities come from Proposition A.1. Thus, $V_0^{\mathcal{M},\pi}(s_0) - V_0^{\mathcal{M}_t,*}(s_0) \geq \zeta/6$.

We now leverage this value gap to show that $\pi$ must use some exit $(s,a)$ whose dynamics in $\mathcal{M}$ have been modified from $\mathcal{M}_t$ with some probability. Formally, define the set $\Delta = \left\{(s,a,h) \mid \mathbb{P}_t(\cdot \mid s,a) \neq \mathbb{P}_h^{\mathcal{M}}(\cdot \mid s,a)\right\}$. By construction, $\Delta \subseteq \mathrm{Ext}(\mathcal{S}) \times [H]$, and for any $(s,a,h) \in \Delta$, there exists $t'$ such that $N_{t'}(s,a) \geq N_{\mathrm{thresh}}^{\mathrm{TS}}$ and $\mathbb{P}_{t'}(\cdot \mid s,a) = \mathbb{P}_h^{\mathcal{M}}(\cdot \mid s,a)$. Furthermore, $\Delta$ must be non-empty, as otherwise, $\mathbb{P}_t = \mathbb{P}_h^{\mathcal{M}}$ for all $h$, and thus $V_0^{\mathcal{M},\pi}(s_0) \leq V_0^{\mathcal{M}_t,*}(s_0)$, a contradiction. Therefore, by applying Lemma A.17, we find that

$$
\frac{\zeta}{6H} < P^{\mathcal{M}_t}(\tau_\pi \cap \Delta \neq \varnothing) \leq \sum_{\{(s,a) \mid (s,a,h) \in \Delta\}} P^{\mathcal{M}_t}((s,a) \in \tau_\pi),
$$

which implies the desired result, as $\{(s,a) \mid (s,a,h) \in \Delta\}$ has at most $K$ elements. $\qquad\square$

Because of Lemma A.9, we simply need to run $\pi$ enough times and threshold at the number of samples needed to reliably determine which $(s,a)$ pairs have an $O(\beta)$ change in TV distance between tasks.

**Lemma A.10.** *We work in the setting of Lemma A.9. Set*

$$
N_{\mathrm{thresh}}^{\mathrm{ED}} = \Omega\left(\frac{S}{\beta^2} \log \frac{SAHN_{\mathrm{ED}}}{p}\right) \quad \text{and} \quad N_{\mathrm{ED}} = \Omega\left(\frac{HK}{\zeta} N_{\mathrm{thresh}}^{\mathrm{ED}} + \frac{H^2 K^2}{\zeta^2} \log \frac{K}{p}\right).
$$

*Then, if we execute $\pi$ within $\mathcal{M}_t$ $N_{\mathrm{ED}}$ and let $N(s,a)$ be the number of times that $(s,a)$ is played in this process, then with probability at least $1 - p/6K$, the following hold:*

(a) *For the $(s,a)$ pair and task $t'$ in Lemma A.9, $N(s,a) \geq N_{\mathrm{thresh}}^{\mathrm{ED}}$ and*

$$
\left\|\hat{\mathbb{P}}(\cdot \mid s,a) - \hat{\mathbb{P}}_{t'}(\cdot \mid s,a)\right\|_{\mathrm{TV}} > \frac{\beta}{2}.
$$

(b) *For any $(s,a) \notin \mathrm{Ext}(\mathcal{S})$ with $N(s,a) \geq N_{\mathrm{thresh}}^{\mathrm{ED}}$ and $t'$ with $N_{t'}(s,a) \geq 0$,*

$$
\left\|\hat{\mathbb{P}}_t(\cdot \mid s,a) - \hat{\mathbb{P}}_{t'}(\cdot \mid s,a)\right\|_{\mathrm{TV}} \leq \frac{\beta}{2}.
$$

*Proof.* By the choice of $N_{\text{ED}}$ and the lower bound $P^{\mathcal{M}_t}((s,a) \in \tau_\pi) > \zeta/6KH$ from Lemma A.9, we guarantee that $N(s,a) \geq N_{\text{thresh}}^{\text{ED}}$ with probability at least $1 - p/12K$. Furthermore, due to the choice of $N_{\text{thresh}}^{\text{ED}}$, with probability at least $1 - p/12K$, we have that for any $(s,a)$ with $N(s,a) \geq N_{\text{thresh}}^{\text{ED}}$,

$$\left\| \hat{\mathbb{P}}(\cdot \mid s,a) - \mathbb{P}_t(\cdot \mid s,a) \right\|_{\text{TV}} < \frac{\beta}{4},$$

by applying Lemma A.16. We condition on these two events simultaneously for the rest of the proof, which occurs with probability at least $1 - p/6K$.

We now prove each part separately. For brevity, we omit $(s,a)$ wherever it is understood.

(a) By applying the triangle inequality,

$$\left\| \mathbb{P}_t - \mathbb{P}_{t'} \right\|_{\text{TV}} \leq \left\| \mathbb{P}_t - \hat{\mathbb{P}} \right\|_{\text{TV}} + \left\| \hat{\mathbb{P}} - \hat{\mathbb{P}}_{t'} \right\|_{\text{TV}} + \left\| \hat{\mathbb{P}}_{t'} - \mathbb{P}_{t'} \right\|_{\text{TV}} \leq \left\| \hat{\mathbb{P}} - \hat{\mathbb{P}}_{t'} \right\|_{\text{TV}} + \frac{\beta}{2}.$$

Therefore, by lower bounding the left-hand side using $\beta$-dynamics separation in Definition 4.1, we find that

$$\left\| \hat{\mathbb{P}}(\cdot \mid s,a) - \hat{\mathbb{P}}_{t'}(\cdot \mid s,a) \right\|_{\text{TV}} > \frac{\beta}{2}.$$

(b) The triangle inequality implies that

$$\left\| \hat{\mathbb{P}} - \hat{\mathbb{P}}_{t'} \right\|_{\text{TV}} \leq \underbrace{\left\| \hat{\mathbb{P}} - \mathbb{P}_t \right\|_{\text{TV}}}_{\leq \beta/4} + \underbrace{\left\| \mathbb{P}_t - \mathbb{P}_{t'} \right\|_{\text{TV}}}_{=0} + \underbrace{\left\| \mathbb{P}_{t'} - \hat{\mathbb{P}}_{t'} \right\|_{\text{TV}}}_{\leq \beta/4} \leq \frac{\beta}{2},$$

where the bound on the first term is provided by Proposition A.1. $\qquad\square$

The prior result demonstrates that if the exit-finding condition is detected at any point by the algorithm, then the algorithm finds a previously undiscovered exit in $\text{Ext}(\mathcal{S})$. At this point, all that remains is to ensure that the algorithm sufficiently learns the dynamics of the newly-found exit in all of the meta-training tasks.

**Lemma A.11.** *Fix an* $(s,a) \in \text{Ext}(\mathcal{S})$, *which was found via exit detection, and let*

$$N_{\text{thresh}}^{\text{EL}} = \Omega \left[ L \max \left( \frac{H^4}{\zeta^2}, \frac{1}{\beta^2} \right) \log \frac{SAHN_{\text{EL}}T}{p} \right]$$

*Assume we run the exit-learning subroutine with*

$$N_{\text{EULER}}^{\text{EL}} = \Omega \left[ \frac{CS^2AH^3}{\alpha} \log^3 \left( \frac{SAHT}{p} \right) \right] \quad \text{and} \quad N_{\text{EL}} = \Omega \left( \frac{CH}{\alpha} N_{\text{thresh}}^{\text{EL}} + \frac{C^2H^2}{\alpha^2} \log \frac{SAT}{p} \right).$$

*in each of the tasks. Then, with probability at least* $1 - p/6K$,

$$\max_{f:\mathcal{S}\to[0,H]} \left| \left[ (\hat{\mathbb{P}}_t - \mathbb{P}_t)f \right](s,a) \right| \leq \min \left( \frac{\zeta}{24H}, \frac{\beta H}{2} \right)$$

*for every* $t \in [T]$.

*Proof.* Fix a task $t \in [T]$. Note that Assumption 4.1 implies that $(s,a)$ is $(\alpha/H)$-significant for some task. Then, $(s,a)$ must be $(\alpha/CH)$-significant for all of the other tasks by Assumption A.1.

By applying Lemma A.4, the set of policies found by the exit-learning subroutine for every task $t \in [T]$ satisfies

$$\max_{\pi} P((s,a) \in \tau_\pi) - \frac{1}{N_{\text{EULER}}^{\text{EL}}} \sum_{k=1}^{N_{\text{EULER}}^{\text{EL}}} P((s,a) \in \tau_{\pi_k})$$

$$\lesssim \sqrt{\max_{\pi} P((s,a) \in \tau_\pi) \frac{SA}{N_{\text{EULER}}^{\text{EL}}} \log \frac{SAHTN_{\text{EULER}}^{\text{EL}}}{p}} + \frac{S^2AH^2}{N_{\text{EULER}}^{\text{EL}}} \log^3 \frac{SAHTN_{\text{EULER}}^{\text{EL}}}{p}$$

with probability at least $1 - p/18TK$. By setting

$$N_{\text{EULER}}^{\text{EL}} \gtrsim \frac{CS^2AH^3}{\alpha} \log^3 \left( \frac{HSAT}{p} \right),$$

we thus find that for any $t \in [T]$,

$$\frac{\alpha}{2CH} \leq \frac{1}{2} \max_{\pi} P((s,a) \in \tau_\pi) \leq \frac{1}{N_{\text{EULER}}^{\text{EL}}} \sum_{\pi \in \Phi_t(s,a)} P((s,a) \in \tau_\pi).$$

Note that the right-hand side is exactly the probability that the trajectory of a randomly chosen policy in $\Phi_t(s,a)$ contains $(s,a)$ in the trajectory. Therefore, by applying Lemma A.15, playing

$$N_{\text{EL}} = \Omega \left( \frac{CH^2 N_{\text{thresh}}^{\text{EL}}}{\alpha} + \frac{C^2 H^4}{\alpha^2} \log \frac{SAT}{p} \right)$$

is sufficient to guarantee that we obtain at least $N_{\text{thresh}}^{\text{EL}}$ samples from $\mathbb{P}_t(\cdot \mid s,a)$ with probability at least $1 - p/18TK$. Since $(s,a) \in \text{Ext}(\mathcal{S})$, $\mathbb{P}_t(\cdot \mid s,a)$ (and by extension, $\hat{\mathbb{P}}_t(\cdot \mid s,a)$) is supported on $\text{Ent}(\mathcal{S})$. Therefore, we can modify the proof in Lemma A.16 so that with probability at least $1 - p/18TK$ we get the bound

$$\max_{f:\mathcal{S}\to[0,H]} \left| \left[ (\hat{\mathbb{P}}_t - \mathbb{P}_t)f \right](s,a) \right| \leq \max_{f:\text{Ent}(\mathcal{S})\to[0,H]} \left| \left[ (\hat{\mathbb{P}}_t - \mathbb{P}_t)f \right](s,a) \right|$$

$$\leq \min \left( \frac{\zeta}{24H}, \frac{\beta H}{2} \right)$$

with $N_{\text{thresh}}^{\text{EL}}$ depending linearly on $L$ instead of $S$.

Note that by performing a union-bound, all events occur with probability at least $1 - p/6TK$. Performing a second union-bound over all of the available tasks results in the desired failure probability. □

At this point, we have effectively proven the second half of our Phase III guarantee. All that remains is to prove that if $\{(s,a) \mid \text{IsExit}[s,a]\} = \text{Ext}(\mathcal{S})$, then the algorithm terminates without triggering the exit-finding condition.

**Lemma A.12.** *Assume that $E = \{(s,a) \mid \text{IsExit}[s,a]\} = \text{Ext}(\mathcal{S})$. Then, under the preconditions of Proposition A.3, every task satisfies*

$$\left| \tilde{V}^t(s_0) - \hat{V}_t \right| \leq \frac{2}{3}\zeta.$$

*Proof.* Fix a task $t \in [T]$. Once $E = \text{Ext}(\mathcal{S})$, then $\mathbb{M}_t(E) = \{\mathcal{M}_t\}$, since the only $(s,a)$-dynamics that can be substituted from other tasks are those of non-exits, which do not change between tasks. Therefore, by Lemma A.5, $\tilde{V}^t(s_0) = \hat{V}_0^{\mathcal{M},*}(s_0)$. Finally, by applying Lemma A.7, we thus find that

$$\tilde{V}^t(s_0) - \hat{V}_t = \left[ \hat{V}_0^{\mathcal{M},*}(s_0) - V_0^{\mathcal{M},*}(s_0) \right] + \left[ V_0^{\mathcal{M},*}(s_0) - \hat{V}_t \right] < \frac{\zeta}{3}.$$

The desired result follows since the argument holds for any task $t$. □

### A.3.4  PROOF OF THEOREM A.1

In this section, we compile the guarantees provided by each of the three phases into a proof of Theorem A.1.

*Proof of Theorem A.1.* We condition on the following high-probability events:

(a) Proposition A.1 guarantees for all $\mathcal{M}_t$ with $t \in [T]$.

(b) Proposition A.2.

Via a union-bound, this holds with probability at least $1 - (2/3)p$.

To prove the theorem, we provide an induction-based analysis of Phase III. In particular, we will show that while $E = \{(s, a) \mid \text{IsEXIT}[s, a]\} \subsetneq \text{Ext}(\mathcal{S})$, Phase III will add at least one state-action pair to $E$ that belongs to $\text{Ext}(\mathcal{S}) \setminus E$.

Formally, let $F_k$ denote the internal state of the algorithm after it has added $k$ state-action pairs. Note that with $k = 0$, $\{(s, a) \mid \text{IsEXIT}[s, a]\}$ in $F_k$ is empty. Thus, $F_k$ satisfies the preconditions of Proposition A.3, which in turn implies that the algorithm adds a new state-action pair in $\text{Ext}(\mathcal{S})$ and sufficiently learns its dynamics for all tasks with probability at least $1 - p/3K$. In short, the internal state of the algorithm at time $F_1$ also satisfies the preconditions of Proposition A.3 with probability at least $1 - p/3K$. More generally, Proposition A.3 ensures that if $F_k$ satisfies the preconditions of Proposition A.3, then so does $F_{k+1}$. Therefore, with probability at least $1 - p/3$, $F_K$ satisfies the preconditions of Proposition A.3, which necessarily implies that $\{(s, a) \mid \text{IsEXIT}[s, a]\} = \text{Ext}(\mathcal{S})$ in $F_K$, and thus the algorithm exits as desired. By performing a union bound, all this occurs with probability at least $1 - p$. $\qquad\square$

## A.4 PROVING THE META-TRAINING GUARANTEE

Having demonstrated that $\text{Ext}(\mathcal{S})$ can be successfully recovered by interacting with the environment, we now show that the data can also be used to determine exit reachability and implement the hierarchy oracle.

We formally state our main result here:

**Theorem A.2.** *Assume that $\mathcal{M}_1, \ldots, \mathcal{M}_T$ have a latent hierarchy with respect to $(\{Z_c\}, \text{Ent}(\cdot), \text{Ext}(\cdot))$, and assume that these tasks satisfy the $(\alpha, \zeta)$-coverage condition in Assumption 4.1. Furthermore, we assume the additional assumptions in Section A.2. Then, by running the algorithm in Section A.1 with the parameters in Table 1, with probability at least $1 - p$, the collected data can be used to implement the following:*

*(a) An $\varepsilon$-suboptimal hierarchy oracle.*

*(b) A function $\text{AvExt}(s) : \text{Ent}(\mathcal{S}) \to \mathcal{P}(\text{Ext}(\mathcal{S}))$ such that, given $s \in \text{Ent}(Z_s)$, returns $\text{Ext}(Z_s)$.*

*The algorithm achieves both of these with query complexity*

$$\tilde{O}\left[\frac{S^4 A}{\min(\rho \min(\varepsilon, \varepsilon_0), \zeta/C)} + \frac{S^2 A}{\min(\rho \min(\varepsilon, \varepsilon_0)^2, \zeta^2/C)}\right.$$
$$\left. + T\left(\frac{SA}{\min(\alpha, \zeta)^2} + \frac{KS}{\zeta\beta^2} + \frac{K^2}{\zeta^2} + \frac{CKS^2 A}{\alpha} + \frac{CKL}{\alpha \max(\zeta, \beta)^2}\right)\right] \text{poly}(H).$$

### A.4.1 IMPLEMENTING THE HIERARCHY ORACLE

We first show that we can implement the hierarchy oracle in this section. In particular, we have the following result:

**Proposition A.4.** *Let $\mathcal{M}$ be the MDP corresponding to the index $(x, f, r, \tilde{H})$ as described in Definition 4.3. Then, given $(x, f, r, \tilde{H})$, we can form the following estimator for $\mathbb{P}_f$:*

$$\hat{\mathbb{P}}_f(\cdot \mid s, a) = \begin{cases} \delta(f(s, a)) & (s, a) \in \text{Ext}(\mathcal{S}) \\ \delta(s) & s = \ominus_S \text{ or } s = \ominus_F \\ \hat{\mathbb{P}}_0(\cdot \mid s, a) & \text{otherwise} \end{cases},$$

*where $\hat{\mathbb{P}}_0$ is the estimator obtained from Phase II in Section A.1. Assuming that the high-probability event in Proposition A.2 holds for $\delta \leq \rho\varepsilon/2SH^2$, value iteration using $\hat{\mathbb{P}}_f$ returns a policy $\pi$ such that $V_0^{\mathcal{M},*}(x) - V_0^{\mathcal{M},\pi}(x_0) \leq \varepsilon$.*

Throughout the rest of this section, we fix the tuple $(x, f, r, \tilde{H})$ and the corresponding MDP $\mathcal{M}$. Furthermore, we write $Z$ for the cluster containing $x$.

To prove Proposition A.4, we will show that $\mathcal{M}$ can be sufficiently simulated so that the value of any policy can be reasonably estimated. Given this simulation result, we can then show that value iteration finds the desired policy. This simulation result depends on the following intermediate result, which provides insight as to why Phase II data is sufficient:

**Lemma A.13.** *For any $s^* \in Z$,*

$$\left[ \max_\pi P^{\mathcal{M}_1}(x \in \tau_\pi) \right] \left[ \max_\pi P^{\mathcal{M}}(s^* \in \tau_\pi) \right] \leq \max_\pi P^{\mathcal{M}_1(2H)}(s^* \in \tau_\pi)$$

*Proof.* First, we note that there exists an MDP such that $P^{\mathcal{M}_1(2H)}(s^* \in \tau_\pi)$ is the corresponding value function. In particular, modifying $\mathcal{M}_1(2H)$ so that any action from $s^*$ leads to a terminal state and defining $r(s, a) = \mathbb{1}[s = s^*]$ results in such an MDP.

Now, let $\pi_x$ and $\pi_{s^*}$ be the policies achieving

$$\max_\pi P^{\mathcal{M}_1}(x \in \tau_\pi) \quad \text{and} \quad \max_\pi P^{\mathcal{M}}(s^* \in \tau_\pi),$$

respectively. Consider the concatenation of $\pi_x$ and $\pi_{s^*}$ into a history-dependent policy that runs $\pi_x$ until the agent reaches $s$, and switches to $\pi_{s^*}$ thereafter. This policy reaches $s$ with probability at least

$$\left[ \max_\pi P^{\mathcal{M}_1}(x \in \tau_\pi) \right] \left[ \max_\pi P^{\mathcal{M}}(s^* \in \tau_\pi) \right].$$

within the modified MDP described above. Since the optimal value among all policies is achieved by a history-independent policy, we obtain the desired inequality. $\square$

Informally, the prior result states that if $x$ is reachable within horizon $H$, then any state reachable from $x$ within $Z$ is also reachable in $\mathcal{M}_1$ within a $2H$ horizon. Therefore, performing reward-free RL with horizon $2H$ during Phase II provides coverage over all clusters. Now, we prove the simulation result.

**Lemma A.14.** *Assume that the Phase II guarantee in Proposition A.2 is instantiated for $\delta \leq \rho\varepsilon/4SH^2$. Then, if $V^\pi$ is the value of $\pi$ under $\mathcal{M}$, and $\hat{V}^\pi$ is its corresponding estimate under $\hat{\mathbb{P}}_f$, then*

$$\left| \hat{V}_0^\pi(x) - V_0^\pi(x) \right| \leq \frac{\varepsilon}{2}.$$

*Proof.* The proof follows similarly to that of Lemma A.7. By the performance difference lemma,

$$\left| \hat{V}_0^\pi(s) - V_0^\pi(s) \right| \leq \sum_{h=0}^{\tilde{H}-1} \mathbb{E}_{\mathcal{M},\pi} \left[ \left| \left[ \left( \hat{\mathbb{P}}_f - \mathbb{P}_f \right) \hat{V}_{h+1}^\pi \right] (s_h, a_h) \right| \right]$$

$$\leq \sum_{h=0}^{\tilde{H}-1} \sum_{(s,a)} \left| \left[ \left( \hat{\mathbb{P}}_f - \mathbb{P}_f \right) \hat{V}_{h+1}^\pi \right] (s, a) \right| P_h^\pi(s, a).$$

Observe that if $s \in \mathcal{S} \setminus Z$, $P_h^\pi(s, a) = 0$ for any $\pi$. Furthermore, since the dynamics within $\{\ominus_S, \ominus_F\}$ are known, $(\hat{\mathbb{P}}_f - \mathbb{P}_f)\hat{V}_{h+1}^\pi(s, a) = 0$ for $s \in \{\ominus_S, \ominus_F\}$. Therefore, we can restrict the sum to be over $Z \times \mathcal{A}$.

Now, let $Z_\delta$ denote the set of $\delta$-significant $(s, a)$ pairs in $Z \times \mathcal{A}$ from $x$, for some $\delta$ to be determined. For a fixed $h \in [\tilde{H}]$, we can decompose the inner sum as

$$\sum_{(s,a)} \left| \left[ \left( \hat{\mathbb{P}}_f - \mathbb{P}_f \right) \hat{V}_{h+1}^\pi \right] (s, a) \right| P_h^\pi(s, a)$$

$$\leq \underbrace{\sum_{(s,a) \in Z_\delta} \left| \left[ \left( \hat{\mathbb{P}}_f - \mathbb{P}_f \right) \hat{V}_{h+1}^\pi \right] (s, a) \right| P_h^\pi(s, a)}_{(\text{I})} + \underbrace{\sum_{(s,a) \notin Z_\delta} \left| \left[ \left( \hat{\mathbb{P}}_f - \mathbb{P}_f \right) \hat{V}_{h+1}^\pi \right] (s, a) \right| P_h^\pi(s, a)}_{(\text{II})}.$$

**Bounding (II): Error from $\delta$-Insignificance**   By the definition of $\delta$-significance,

$$(\text{II}) = \sum_{(s,a)\notin Z_\delta} \left| \left[ \left( \hat{\mathbb{P}}_f - \mathbb{P}_f \right) \hat{V}_{h+1}^\pi \right] (s,a) \right| P_h^\pi(s,a) \le H \sum_{s\notin Z_\delta} P_h^\pi(s) \le HS\delta \le \frac{\varepsilon}{4H},$$

where the last inequality follows from setting $\delta = \varepsilon/4SH^2$.

**Bounding (I): Reference Dynamics Error.**   By the Cauchy-Schwarz inequality,

$$(\text{I}) = \sum_{(s,a)\in Z_\delta} \left| \left[ \left( \hat{\mathbb{P}}_f - \mathbb{P}_f \right) \hat{V}_{h+1}^\pi \right] (s,a) \right| P_h^\pi(s,a)$$

$$\le \left[ \sum_{(s,a)\in Z_\delta} \left| \left[ \left( \hat{\mathbb{P}}_f - \mathbb{P}_f \right) \hat{V}_{h+1}^\pi \right] (s,a) \right|^2 P_h^\pi(s,a) \right]^{1/2}.$$

Then,

$$\sum_{(s,a)\in Z_\delta} \left| \left[ \left( \hat{\mathbb{P}}_f - \mathbb{P}_f \right) \hat{V}_{h+1}^\pi \right] (s,a) \right|^2 P_h^\pi(s,a)$$

$$\le \max_{\nu:\mathcal{S}\to\mathcal{A}} \sum_{(s,a)\in Z_\delta} \left| \left[ \left( \hat{\mathbb{P}}_f - \mathbb{P}_f \right) \hat{V}_{h+1}^\pi \right] (s,a) \right|^2 P_h^\pi(s)\mathbb{1}\left[ \nu(s) = a \right].$$

Since $x$ is $\rho$-significant in $\mathcal{M}_1(2H)$ by Definition A.2, Lemma A.13 together with $\delta$-significance in $\mathcal{M}$ implies $\rho\delta$-significance in $\mathcal{M}_1(2H)$. Therefore,

$$P_h^\pi(s) \le \max_\pi P^{\mathcal{M}}(s \in \tau_\pi) \le \frac{1}{\rho} \max_\pi P^{\mathcal{M}_1(2H)}(s \in \tau_\pi) \le \frac{4HSA}{\rho}\mu(s,a),$$

where the last inequality follows by part (a) of the Phase II guarantee in Proposition A.2. Substituting into the prior expression,

$$\max_{\nu:\mathcal{S}\to\mathcal{A}} \sum_{(s,a)\in Z_\delta} \left| \left[ \left( \hat{\mathbb{P}}_f - \mathbb{P}_f \right) \hat{V}_{h+1}^\pi \right] (s,a) \right|^2 P_h^\pi(s)\mathbb{1}\left[ \nu(s) = a \right]$$

$$\le \frac{4HSA}{\rho} \max_{\nu:\mathcal{S}\to\mathcal{A}} \sum_{(s,a)\in Z_\delta} \left| \left[ \left( \hat{\mathbb{P}}_f - \mathbb{P}_f \right) \hat{V}_{h+1}^\pi \right] (s,a) \right|^2 \mathbb{1}\left[ \nu(s) = a \right]\mu(s,a)$$

$$\le \frac{4HSA}{\rho} \max_{\nu:\mathcal{S}\to\mathcal{A}} \mathbb{E}_{(s,a)\sim\mu} \left[ \left| \left[ \left( \hat{\mathbb{P}}_f - \mathbb{P}_f \right) \hat{V}_{h+1}^\pi \right] (s,a) \right|^2 \mathbb{1}\left[ \nu(s) = a \right] \right].$$

Thus by applying part (b) of the Phase II guarantee in Proposition A.2, we have that

$$(\text{I}) \le \sqrt{\frac{4HSA}{\rho} \max_{\nu:\mathcal{S}\to\mathcal{A}} \mathbb{E}_{(s,a)\sim\mu} \left[ \left| \left[ \left( \hat{\mathbb{P}}_f - \mathbb{P}_f \right) \hat{V}_{h+1}^\pi \right] (s,a) \right|^2 \mathbb{1}\left[ \nu(s) = a \right] \right]} \le \frac{\varepsilon}{4H}.$$

**Concluding.**   By combining the bounds on (I) and (II), we obtain the desired result.   $\square$

With this estimation result, we can now prove Proposition A.4.

*Proof of Proposition A.4.* Let $\pi$ be the policy found by value iteration using $\hat{\mathbb{P}}_f$, which achieves the maximal value in the corresponding MDP. Then, by Lemma A.14

$$V_0^*(x) - V_0^\pi(x) \le \underbrace{\left[ V_0^*(s_0) - \hat{V}_0^{\pi^*}(s_0) \right]}_{\le \varepsilon/2} + \underbrace{\left[ \hat{V}_0^{\pi^*}(s_0) - \hat{V}_0^{\hat{\pi}}(s_0) \right]}_{\le 0} + \underbrace{\left[ \hat{V}_0^{\hat{\pi}}(s_0) - V_0^{\hat{\pi}}(s_0) \right]}_{\le \varepsilon/2} \le \varepsilon. \quad \square$$

A.4.2 DETERMINING AVAILABLE EXITS

In this section, we prove that we can determine the set of available exits. We have the following formal result:

**Proposition A.5.** *Assume access to the $\varepsilon$-suboptimal hierarchy oracle from the previous section and that the guarantee in Theorem A.1 holds. Then, we can implement the function $\mathrm{AvExt}(s)$ : $\mathrm{Ent}(\mathcal{S}) \to \mathcal{P}(\mathrm{Ext}(\mathcal{S}))$ which, given $s \in \mathrm{Ent}(Z_s)$, returns the subset of $\mathrm{Ext}(Z_s)$ that is reachable from $s$.*

*Proof.* Fix an input $x \in \mathrm{Ent}(\mathcal{S})$, which we assume belongs to some cluster $Z_x$. It suffices to demonstrate that we can implement $\mathbb{1}\left[e \in Z_x\right]$ for any fixed $e \in \mathrm{Ent}(\mathcal{S})$. Define

$$f_e(s, a) = \begin{cases} \ominus_S & (s, a) = e \\ \ominus_F & \text{otherwise} \end{cases}$$

and $r_e(s, a) = \mathbb{1}\left[(s, a) = e\right]$. By Definition A.3, the MDP $\mathcal{M}$ corresponding to the tuple $(x, f_e, r_e, H)$ has optimal value $V^* = \varepsilon_0 \mathbb{1}\left[e \in Z_x \wedge e \text{ reachable from } x\right]$. Additionally, by Lemma A.14, $|V_0^\pi(s) - \hat{V}_0^\pi(x)| \le \varepsilon/2$. We now proceed by cases. If $e \notin Z_x$ or $e$ is not reachable from $x$, then $V_0^\pi(x) = 0$ for any policy $\pi$, and thus value iteration can only find a policy $\pi$ with $\hat{V}_0^\pi(x) \le \varepsilon_0/3$. Otherwise, for $e \in Z_x$, $V_0^*(x) = \varepsilon_0$, and thus value iteration necessarily must find a $\pi$ with $\hat{V}_0^\pi(x) \ge 2\varepsilon_0/3$. Putting these together, if $\hat{V}$ is the optimal estimated value in $\mathcal{M}$, then

$$\mathbb{1}\left[e \in Z_x\right] = \mathbb{1}\left[\hat{V} \ge \frac{2}{3}\varepsilon_0\right].$$

Note that this is implementable for all $e \in \mathrm{Ext}(\mathcal{S})$ (and returns a subset of $\mathrm{Ext}(Z_x)$ for the query above) since the set of exits are already known. $\square$

A.4.3 FINALIZING THE GUARANTEE: QUERY COMPLEXITY

In this section, we finalize the proof of the meta-training guarantee by computing the query complexity.

*Proof of Theorem A.2.* As demonstrated by Proposition A.4 and Proposition A.5, running the algorithm in Section A.1 with the parameters in Table 1 provides the desired guarantees with probability at least $1 - p$.

To compute the query complexity, observe that we perform the following number of trajectories while executing the algorithm in Section A.1.

$$O\left[T(N_{\mathrm{UCBVI}} + N_{\mathrm{TS}}) + N_{\mathrm{EULER}}^{\mathrm{RF}} + N_{\mathrm{RF}} + KN_{\mathrm{ED}} + TK(N_{\mathrm{EULER}}^{\mathrm{EL}} + N_{\mathrm{EL}})\right].$$

Ignoring terms that do not depend on $T$ or $\varepsilon$, we obtain the claim. $\square$

A.5 BRUTE-FORCE LEARNING OF THE HIEARCHY

**Theorem A.3.** *Assume that Algorithm 6 is run with parameters satisfying*

$$N_{\mathrm{EULER}} = \Omega\left(\frac{CH^3S^2A}{\alpha} \log^3 \frac{SAHT}{p}\right)$$

*and*

$$N_{\mathrm{thresh}} = \Omega\left(\frac{S}{\beta^2} \log \frac{SAHNT}{p}\right) \quad \text{and} \quad N = \Omega\left(\frac{CH}{\alpha}N_{\mathrm{thresh}} + \frac{C^2H^2}{\alpha^2} \log \frac{SAT}{p}\right)$$

*Then, the set returned by the algorithm is exactly $\mathrm{Ext}(\mathcal{S})$ with probability at least $1 - p$. Furthermore, the algorithm achieves this result with query complexity*

$$\tilde{O}\left[T\left(\frac{CS^4A}{\alpha} + \frac{CS^2A}{\alpha\beta^2}\right)\right] \mathrm{poly}(H).$$

---

**Algorithm 6** Brute-force learning of the latent hierarchy.

1: **procedure** LEARNHIERARCHY$((\mathcal{M}_1, \ldots, \mathcal{M}_T), N_{\text{EULER}}$ iterations, $N$ policy samples, threshold $N_{\text{thresh}})$
2:     **for all** $t \in [T], s \in \mathcal{S}$ **do**
3:         Create MDP $\mathcal{M}_t^s$ so $P(\ominus \mid s, a) = 1$ for any $a$.
4:         $\tilde{r}_s(s', a') \leftarrow \mathbb{1}[s' = s]$.
5:         $\Psi_t^s \leftarrow \text{EULER}(\mathcal{M}_t^s, r, N_{\text{EULER}})$
6:     **for all** $t \in [T], s \in \mathcal{S}, a \in \mathcal{A}$ **do**
7:         Modify policies in $\Psi_t^s$ to play $a$ on $s$.
8:         **for all** $n \in [N]$ **do**
9:             Sample $\pi \sim \text{Unif}(\Psi_t^s)$.
10:            Play $\pi$ in $\mathcal{M}_t$, collect sample $(s, a, s_n')$ if $(s, a)$ is encountered
11:         $N_t(s, a) \leftarrow$ number of times $(s, a)$ is encountered above.
12:         $\hat{\mathbb{P}}_t(\cdot \mid s, a) \leftarrow$ estimate of $(s, a)$ dynamics in $t$.
13:     **return** $\left\{ (s, a) \mid (\exists t \neq t') \left\| \hat{\mathbb{P}}_t - \hat{\mathbb{P}}_{t'} \right\|_{\text{TV}} > \beta/2, \min(N_t(s, a), N_{t'}(s, a)) \geq N_{\text{thresh}} \right\}.$

---

*Proof.* For any $(s, a) \in \text{Ext}(\mathcal{S})$, Lemma A.2 implies that $s$ is $\alpha/H$-significant for some task $t \in [T]$. Therefore, $s$ is $\alpha/CH$-significant for any task $t \in [T]$, by Assumption A.1.

Now, by an argument similar to that used in the proof of Lemma A.2, we have that with probability at least $1 - p/3T$, the choice of $N_{\text{EULER}}$ implies

$$\frac{1}{N_{\text{EULER}}} \sum_{\pi \in \Psi_t^s} P^{\mathcal{M}_t}(s \in \tau_\pi) \geq \frac{\alpha}{2CH}$$

for any exit $(s, a) \in \text{Ext}(\mathcal{S})$ and a fixed task $t \in [T]$. Therefore, by a union-bound over the tasks, the same guarantee holds for all tasks simultaneously with probability at least $1 - p/3$.

Now, for any fixed $(\alpha/CH)$-significant $(s, a)$ pair, sampling from $\Psi_t^s$ at least $N$ times guarantees that with probability at least $1 - p/3SAT$, $N_t(s, a) \geq N_{\text{thresh}}$. Therefore, once again performing the necessary union-bound, we obtain the same result uniformly over any $(\alpha/CH)$-significant $(s, a)$ and $t \in [T]$ with probability at least $1 - p/3$.

Finally, for a fixed $(s, a)$ and $t$, the estimator for $\mathbb{P}_t(\cdot \mid s, a)$ satisfies the property that when $N(s, a) > 0$,

$$\left\| \hat{\mathbb{P}}_t(\cdot \mid s, a) - \mathbb{P}_t(\cdot \mid s, a) \right\|_{\text{TV}} \leq \sqrt{\frac{H^2 S}{N_t(s, a)} \log \frac{SAHNT}{p}} + \frac{HS}{N_t(s, a)} \log \frac{SAHNT}{p}$$

with probability at least $1 - p/3SAT$, using an argument similar to that used in Lemma A.16. Again, by a union bound, the same guarantee holds for any $(s, a)$ and $t \in [T]$. In particular, for any $(s, a)$ with $N_t(s, a) \geq N_{\text{thresh}}$,

$$\left\| \hat{\mathbb{P}}_t(\cdot \mid s, a) - \mathbb{P}_t(\cdot \mid s, a) \right\|_{\text{TV}} \leq \frac{\beta}{4}.$$

Therefore, by a similar argument to Lemma A.10, the following are true:

(a) If $(s, a) \in \text{Ext}(\mathcal{S})$, then there exists $t, t'$ for which

$$\left\| \hat{\mathbb{P}}_t(\cdot \mid s, a) - \hat{\mathbb{P}}_{t'}(\cdot \mid s, a) \right\|_{\text{TV}} > \frac{\beta}{2}.$$

(b) If $(s, a) \notin \text{Ext}(\mathcal{S})$, then for any $t \neq t'$ with $N_t(s, a), N_{t'}(s, a) \geq N_{\text{thresh}}$,

$$\left\| \hat{\mathbb{P}}_t(\cdot \mid s, a) - \hat{\mathbb{P}}_t(\cdot \mid s, a) \right\|_{\text{TV}} \leq \frac{\beta}{2},$$

Putting everything together, we see that the set returned by Algorithm 6 is exactly $\text{Ext}(\mathcal{S})$, with probability at least $1 - p$. $\qquad\square$

A.6 TECHNICAL LEMMAS

**Lemma A.15.** *Let $X_1, \ldots, X_M$ be i.i.d.* Ber $(p)$ *random variables. Then, if*

$$M = \Omega\left(\frac{N}{p} + \frac{1}{p^2}\log\frac{1}{\delta}\right),$$

*then with probability at least $1 - \delta$,*

$$\sum_{i=1}^{M} \mathbb{1}\left[X_i = 1\right] \geq N.$$

*Proof.* By applying Hoeffding's inequality,

$$P\left(\sum_{i=1}^{M} \mathbb{1}\left[X_i = 1\right] < N\right) = P\left(\frac{1}{M}\sum_{i=1}^{M} \mathbb{1}\left[X_i = 1\right] - p < \frac{N}{M} - p\right)$$

$$= P\left(\frac{1}{M}\sum_{i=1}^{M} \mathbb{1}\left[X_i = 0\right] - (1-p) > p - \frac{N}{M}\right)$$

$$\leq \exp\left[-2M\left(p - \frac{N}{M}\right)^2\right]$$

Setting the final expression to the failure probability $\delta$ and solving, we obtain the quadratic inequality

$$p^2 M^2 - \left(2Np + \frac{1}{2}\log\frac{1}{\delta}\right)M + N^2 \geq 0.$$

Finally, via solving this inequality for $M$, we find that

$$M \geq \frac{2N}{p} + \frac{1}{2p^2}\log\frac{1}{\delta}$$

is sufficient to guarantee the desired event with failure probability $\delta$, as desired. $\qquad\square$

**Lemma A.16** (Dynamics estimation error bound). *Fix a policy $\pi$, MDP with stationary dynamics $\mathcal{M} = (\mathcal{S}, \mathcal{A}, \mathbb{P}, r, H)$, and $N \in \mathbb{N}$. Assume that $\pi$ is played $N$ times in $\mathcal{M}$, and all transitions are used to form an estimator $\hat{\mathbb{P}}(\cdot \mid s, a)$ using empirical averages. For any $(s, a) \in \mathcal{S} \times \mathcal{A}$, let $N(s, a)$ be the number of times $(s, a)$ is encountered in this process. Then, with probability at least $1 - p$, any $(s, a)$ with $N(s, a) > 0$ satisfies*

$$\sup_{f:\mathcal{S}\to[0,H]}\left|\left[\left(\hat{\mathbb{P}} - \mathbb{P}\right)f\right](s, a)\right| \leq \sqrt{\frac{H^2 S}{N(s, a)}\log\frac{SAHN}{p}} + \frac{HS}{N(s, a)}\log\frac{SAHN}{p}.$$

*Proof.* Assume that the obtained samples are given by $\{(s_k, a_k, s'_k) \mid k \in [HN]\}$, so that $(s_{Hn+r}, a_{Hn+r}, s'_{Hn+r+1})$ is the $r^{\text{th}}$ time step in the $n^{\text{th}}$ execution of $\pi$ in $\mathcal{M}$ for any $0 \leq n \leq N-1$ and $0 \leq r \leq H - 1$.

Fix any $(s, a) \in \mathcal{S} \times \mathcal{A}$, and assume that $(s_{(j)}, a_{(j)}, s'_{(j)})$ is the $j^{\text{th}}$ sample from $\mathbb{P}(\cdot \mid s, a)$. Furthermore, let $m_j(s, a)$ denote the index at which the $j^{\text{th}}$ sample is obtained. We claim that for any $s^* \in \mathcal{S}$ and $0 < M \leq HT$,

$$\left|\frac{1}{M}\sum_{j=1}^{M}\mathbb{1}\left[m_j(s, a) \leq HT\right]\left(\mathbb{1}\left[s'_{(j)} = s^*\right] - \mathbb{P}(s^* \mid s, a)\right)\right|$$

$$\leq \sqrt{\frac{\mathbb{P}(s' \mid s, a)}{M}\log\frac{S}{\delta}} + \frac{1}{M}\log\frac{S}{\delta}.$$

Let $\mathcal{F}_i$ be defined as the $\sigma$-algebra induced by the set of random variables

$$\left\{\left(m_j(a), \mathbb{1}\left[s'_{(j)} = s^*\right]\right) \,\middle|\, j \leq i\right\}.$$

Clearly, $(\mathcal{F}_i)$ is a filtration such that the $j^{\text{th}}$ term in the sum above is measurable with respect to $\mathcal{F}_j$. Furthermore, observe that

$$
\begin{aligned}
&\mathbb{E}\left[\mathbb{1}\left[m_j(s,a) \le HT\right]\left(\mathbb{1}\left[s'_{(j)} = s^*\right] - \mathbb{P}(s^* \mid s, a)\right) \,\Big|\, \mathcal{F}_{j-1}\right] \\
&\quad = \mathbb{E}\left[\mathbb{1}\left[s'_{(j)} = s^*\right] - \mathbb{P}(s^* \mid s, a) \,\Big|\, \mathcal{F}_{j-1}, m_j(s,a) \le HT\right] P(m_j(s,a) \le HT) \\
&\quad = 0.
\end{aligned}
$$

Therefore, the random variables in the sum forms martingale difference sequence. Furthermore, the sequence is bounded in $[-1, 1]$, and satisfies

$$
\begin{aligned}
&\mathrm{Var}\left[\mathbb{1}\left[m_j(s,a) \le HT\right]\left(\mathbb{1}\left[s'_{(j)} = s^*\right] - \mathbb{P}(s^* \mid s, a)\right) \,\Big|\, \mathcal{F}_{j-1}\right] \\
&\quad = \mathbb{E}\left[\mathrm{Var}\left[\mathbb{1}\left[s'_{(j)} = s^*\right] - \mathbb{P}(s^* \mid s, a) \,\Big|\, \mathcal{F}_{j-1}, m_j(s,a) \le HT\right] \,\Big|\, \mathcal{F}_{j-1}\right] \\
&\quad \le \mathbb{P}(s^* \mid s, a).
\end{aligned}
$$

Therefore, by applying Azuma-Bernstein, we have that

$$
\begin{aligned}
&\left|\frac{1}{M}\sum_{j=1}^{M}\mathbb{1}\left[m_j(s,a) \le HT\right]\left(\mathbb{1}\left[s'_{(j)} = s^*\right] - \mathbb{P}(s^* \mid s, a)\right)\right| \\
&\qquad\qquad \le \sqrt{\frac{2\mathbb{P}(s' \mid s, a)}{M}\log\frac{SAHN}{\delta}} + \frac{2}{M}\log\frac{SAHN}{\delta}.
\end{aligned}
$$

with probability at least $1 - p/SAHN$.

By applying a union bound on $(s, a, s^*)$ and $M$, we thus have that with probability at least $1 - p$,

$$
\begin{aligned}
&\left|\frac{1}{M}\sum_{j=1}^{M}\mathbb{1}\left[m_j(s,a) \le HT\right]\left(\mathbb{1}\left[s'_{(j)} = s^*\right] - \mathbb{P}(s^* \mid s, a)\right)\right| \\
&\qquad\qquad \le \sqrt{\frac{2\mathbb{P}(s' \mid s, a)}{M}\log\frac{SAHN}{\delta}} + \frac{2}{M}\log\frac{SAHN}{\delta}
\end{aligned}
$$

holds for any $(s, a, s^*)$ and $M$. Conditioned on this event, we thus have that for any $(s, a)$ with $N(s,a) > 0$,

$$
\begin{aligned}
\left\|\hat{\mathbb{P}}_t(\cdot \mid s, a) - \mathbb{P}_t(\cdot \mid s, a)\right\|_{\mathrm{TV}} &= \frac{1}{2}\sum_{s' \in \mathcal{S}}\left|\hat{\mathbb{P}}_t(s' \mid s, a) - \mathbb{P}_t(s' \mid s, a)\right| \\
&\lesssim \sum_{s' \in \mathcal{S}}\sqrt{\frac{\mathbb{P}(s' \mid s, a)}{N(s,a)}\log\frac{SAHN}{\delta}} + \frac{S}{N(s,a)}\log\frac{SAHN}{\delta} \\
&\lesssim \sqrt{\frac{S}{N(s,a)}\log\frac{SAHN}{\delta}} + \frac{S}{N(s,a)}\log\frac{SAHN}{\delta}.
\end{aligned}
$$

The final result follows simply by noting that

$$
\left|\left[(\hat{\mathbb{P}}_t - \mathbb{P}_t)f\right](s,a)\right| \lesssim \left\|\hat{\mathbb{P}}_t(\cdot \mid s, a) - \mathbb{P}_t(\cdot \mid s, a)\right\|_{\mathrm{TV}}\|f\|_{\infty}. \qquad\qquad \square
$$

**Lemma A.17.** *Fix two MDPs $\mathcal{M} = (\mathcal{S}, \mathcal{A}, \mathbb{P}, r, H)$ and $\mathcal{M}' = (\mathcal{S}, \mathcal{A}, \mathbb{P}', r, H)$. Let $\Delta$ denote the subset of $\mathcal{S} \times \mathcal{A} \times [H]$ for which $\mathbb{P}_h(\cdot \mid s, a) \ne \mathbb{P}'_h(\cdot \mid s, a)$. Then, for any policy $\pi$,*

$$
V_0^{\mathcal{M}',\pi}(s_0) - V_0^{\mathcal{M},*}(s_0) > \rho \implies P_{\mathcal{M}}(\tau_\pi \cap \Delta \ne \varnothing) = P_{\mathcal{M}'}(\tau_\pi \cap \Delta \ne \varnothing) > \frac{\rho}{H}.
$$

*Proof.* Write $q = P_{\mathcal{M}'}(\tau_\pi \cap \Delta \neq \varnothing)$. Note that $V_0^{\mathcal{M}',\pi}(s_0)$ can be decomposed as

$$V_0^{\mathcal{M}',\pi}(s_0) = q\mathbb{E}_{\mathcal{M}'}\left[\sum_{h=0}^{H-1} r_h(s_h, a_h) \,\middle|\, \tau_\pi \cap \Delta \neq \varnothing\right]$$

$$+ (1-q)\mathbb{E}_{\mathcal{M}'}\left[\sum_{h=0}^{H-1} r_h(s_h, a_h) \,\middle|\, \tau_\pi \cap \Delta = \varnothing\right]$$

$$\leq qH + (1-q)\mathbb{E}_{\mathcal{M}'}\left[\sum_{h=0}^{H-1} r_h(s_h, a_h) \,\middle|\, \tau_\pi \cap \Delta = \varnothing\right].$$

Since $\mathbb{P}$ and $\mathbb{P}'$ agree on $(\mathcal{S} \times \mathcal{A} \times [H]) \setminus \Delta$, the dynamics of $\mathcal{M}$ and $\mathcal{M}'$ agree up until $\pi$ performs an action in $\Delta$, and thus

$$P_{\mathcal{M}}(\tau_\pi \cap \Delta \neq \varnothing) = P_{\mathcal{M}'}(\tau_\pi \cap \Delta \neq \varnothing)$$

$$\mathbb{E}_{\mathcal{M}}\left[\sum_{h=0}^{H-1} r_h(s_h, a_h) \,\middle|\, \tau_\pi \cap \Delta = \varnothing\right] = \mathbb{E}_{\mathcal{M}'}\left[\sum_{h=0}^{H-1} r_h(s_h, a_h) \,\middle|\, \tau_\pi \cap \Delta = \varnothing\right]$$

Furthermore,

$$(1-q)\mathbb{E}\left[\sum_{h=0}^{h-1} r_h(s_h, a_h) \,\middle|\, \tau_\pi \cap \delta = \varnothing\right] \leq V_0^{\mathcal{M},\pi}(s_0) \leq V_0^{\mathcal{M},*}(s_0).$$

Putting everything together,

$$V_0^{\mathcal{M}',\pi}(s_0) \leq qH + V_0^{\mathcal{M},*}(s_0) \implies q > \frac{\rho}{H}. \qquad \square$$

## A.7 WHY SHOULD OPTIMISTIC IMAGINATION HOLD?

In this section, we provide a heuristic explanation as for why optimistic imagination can be expected to hold in most scenarios for which Definition 3.1 holds.

Recall from the main text that if an exit $e$ satisfies the preliminary coverage condition, then $e$ can be detected using only Phase I data. Therefore, we can restrict our discussion of Assumption 4.1 to subsets of exits that only have nonzero importance for a task $t$ if $\mathbb{P}_t(\cdot \mid e)$ is some distribution $d_e$. In other words, we can only guarantee that near-optimal policies for the source tasks observe $d_e$. We will provide a heuristic reason as to why we can then expect to find an MDP such that condition (b) of Assumption 4.1 holds for such a subset $S$.

**State visitation measures.** Recall that the value function in an MDP with a stationary reward function is given by

$$V_0^{\mathcal{M},\pi}(s_0) = \sum_{h=0}^{H-1} \mathbb{E}_{\mathcal{M},\pi}\left[r(s_h, a_h) \mid s_0\right].$$

Now, if we assume that the reward is only a function of the state (as in goal-conditioned RL), and view $r$ as a vector in $\mathbb{R}^{|\mathcal{S}|}$, then we can write $V_0^{\mathcal{M},\pi}(s_0) = H\rho_{\mathcal{M},\pi}^\top r$, where

$$\rho_{\mathcal{M},\pi}(s) = \frac{1}{H}\sum_{h=0}^{H-1} P^{\mathcal{M},\pi}(s_h = s \mid s_0).$$

That is, $\rho_{\mathcal{M},\pi}$ is the state visitation measure of $\pi$ within $\mathcal{M}$ with horizon $H$. This geometric viewpoint was explored by Eysenbach et al. (2021) in the context of unsupervised RL in the discounted infinite-horizon setting. We note that for any fixed $\mathcal{M}$, $\{\rho_{\mathcal{M},\pi}\}$ is a convex subset of the probability simplex.

**Constructing the desired reward function.** Let $\mathcal{M}$ be any source task such that the set $T = \left\{ e \in S \mid \mathbb{P}^{\mathcal{M}}(\cdot \mid e) \neq d_e \right\}$ is non-empty. Furthermore, let $\tilde{\mathcal{M}}$ be the corresponding MDP described in Assumption 4.1(b) after borrowing dynamics, so that

$$\tilde{\mathbb{P}}(\cdot \mid s, a) = \begin{cases} \mathbb{P}(\cdot \mid s, a) & (s, a) \notin S \\ d_{(s,a)} & \text{otherwise} \end{cases}$$

is the dynamics of $\tilde{\mathcal{M}}$. Since $\left\| P^{\mathcal{M}}(\cdot \mid e) - d_e \right\|_{TV} > \beta$ for any $e \in S$, heuristically, we expect the following to be true:

**Assumption A.2.** *There exists a $\pi^*$ such that $\rho_{\tilde{\mathcal{M}}, \pi^*} \notin \{\rho_{\mathcal{M}, \pi}\}$.*

Stated less formally, we expect there to be states that $\pi^*$ can reach better in $\tilde{\mathcal{M}}$ compared to any policy in $\mathcal{M}$.

**Example A.1.** In the four-room example, this intuition is captured by the fact that the agent can spend more time in states past open gates compared to if they were closed. ⌐

Under Assumption A.2 above, since $\{\rho_{\mathcal{M}, \pi}\}$ is convex, we can find a reward direction separating $\{\rho_{\mathcal{M}, \pi}\}$ and $\rho_{\tilde{\mathcal{M}}, \pi^*}$, or more formally,

$$\rho_{\tilde{\mathcal{M}}, \pi^*}^\top r > \sup_\pi \rho_{\tilde{\mathcal{M}}, \pi^*}^\top r.$$

Then, the MDP $\mathcal{M}$ with reward function $r$ satisfies the desired condition in Assumption 4.1(b). Thus, under Assumption A.2, the coverage condition can indeed hold with sufficiently diverse source tasks.

# B    META-TEST PROOFS

We now provide an analysis of the regret incurred by a learner using an approximately learned hierarchy at meta-test time. We first show that the hierarchy oracle from the source tasks can provide useful temporally extended behavior. We then show that using these policies results in bounded suboptimality and achieves a better regret bound compared to standard UCB-VI.

Throughout this section, we fix an optimal $\pi^*$ satisfying the conditions of Assumption 5.1. Furthermore, we assume that we have access to a hierarchy oracle that provides $\varepsilon$-suboptimal policies as defined in Definition 4.3.

## B.1    USING THE HIERARCHY ORACLE

In this section, we show that the hierarchy oracle can be used to implement two useful behaviors: (1) reaching exits and (2) behaving optimally within a cluster.

### B.1.1    NEAR-OPTIMAL GOAL REACHING

Assume that the agent is currently at a state $z \in \{s_0\} \cup \mathrm{Ent}(\mathcal{S})$ at time step $h$, and intends to exit the current cluster $Z$ via exit $g = (s^*, a^*) \in \mathrm{Ext}(Z)$. We obtain a policy implementing the high-level intent as follows:

(1) Define the termination for any $(s, a) \in \mathrm{Ext}(\mathcal{S})$ as:

$$f_g(s, a) := \begin{cases} \ominus_S & (s, a) = g \\ \ominus_F & \text{otherwise} \end{cases}$$

(2) Define reward as $r_{\ominus_S}(s, a) := \mathbb{1}\left[s = \ominus_S\right]$

(3) Provide $(z, f_g, r_{\ominus_S}, H - h)$ to the hierarchy oracle and obtain a policy $\pi_{z,g,h}$.

For simplicity, we will write $T_{H-h}^{\mathrm{hier}}(z, g)$ for $T_{H-h}^{\pi_{z,g,h}}(z, g)$ throughout our analysis. The following proposition quantifies the performance of the obtained policy:

**Proposition B.1.** $T^{\mathrm{hier}}$ *satisfies the following inequality:*

$$\mathbb{E}\left[T_{H-h}^{\mathrm{hier}}(z, g)\right] \le T_{H-h}^*(z, g) + \varepsilon.$$

*Proof.* Due to the definition of $\mathbb{P}_{f_g}$ and $r$, observe that for any $\pi$,

$$V_0^\pi(z) = \mathbb{E}\left[\sum_{h=0}^{H-h} r(s_h, a_h) \,\middle|\, s_0 = z\right] = (H - h) - \mathbb{E}\left[T_{H-h}^\pi(z, g)\right].$$

Therefore,

$$(H - h) - T_{H-h}^*(z, g) - \varepsilon \le (H - h) - \mathbb{E}\left[T_{H-h}^{\mathrm{hier}}(z, g)\right]$$
$$\implies \mathbb{E}\left[T_{H-h}^{\mathrm{hier}}(z, g)\right] \le T_{H-h}^*(z, e) + \varepsilon. \qquad \square$$

### B.1.2    NEAR-OPTIMAL WITHIN-CLUSTER BEHAVIOR

Assume that the agent is currently at a state $z \in \{s_0\} \cup \mathrm{Ent}(\mathcal{S})$ at time $h$, and intends to remain in the current cluster $Z$ while maximizing a given reward function $r$. We obtain a policy for this high-level intent as follows:

(1) Define transition dynamics for any $(s, a) \in \mathrm{Ext}(Z)$ as $\mathbb{P}(\cdot \mid s, a) = \delta(\ominus_F)$.

(2) Provide $\mathbb{P}$, r, and planning horizon $H - h$ to the hierarchy oracle, and obtain a policy $\pi$.

### B.2 FORMAL LEARNING PROCEDURE

In this section, we describe the procedure for learning a policy using the oracle-provided policies described in the previous section. Formally, we construct a surrogate MDP whose dynamics are determined by $\mathcal{M}$ and the oracle. We can then apply any tabular learning method to this new MDP (in our case, EULER), obtaining a policy in the surrogate MDP that readily translates into a policy in $\mathcal{M}$.

The components defining the surrogate $\mathcal{M}_{\mathrm{hl}} = (\mathcal{Z}, \mathcal{G}, \mathbb{P}_{\mathrm{hl}}, R_{\mathrm{hl}}, H_{\mathrm{eff}})$ are as follows:

**Meta-state space $\mathcal{Z}$.** We set

$$\mathcal{Z} \coloneqq \left(\mathrm{Ent}(\mathcal{S}) \times \{0, \dots, \bar{H} + 1\}\right) \cup \{\ominus\},$$

where $\bar{H}$ is a high-probability bound on the time to move through $H_{\mathrm{eff}}$ exits (to be determined later). We incorporate the time step into the meta-state to ensure that both the dynamics and reward are computable from the state information (ensuring that $\mathcal{M}_{\mathrm{hl}}$ is indeed an MDP).

**Meta-action space $\mathcal{G}$.** Given a current meta-state $(s, h)$ where $s \in Z$, the available meta-actions $\mathcal{G}$ can be identified with $\mathrm{Ext}(Z) \cup \{\ominus\}$.

---

**Algorithm 7** Performing a Meta-Transition

---

1: **procedure** PERFORMMETATRANSITION($(z, g) \in \mathcal{Z} \times \mathcal{G}$)
    ▷ Executes the desired meta-transition in the original MDP $\mathcal{M}$.
2:    **if** $z = \ominus$ **then**
3:        **return** $\ominus$
4:    **else if** $z = (s, h)$ **then**
5:        **if** $h \leq \bar{H}$ **then**
6:            **if** $s \in Z^*$ or $g = \ominus$ **then**
7:                Execute within-cluster policy from oracle until termination.
8:                **return** $\ominus$
9:            **else**
10:                Execute $\pi_{z,g,h}$ obtained from oracle until $g$ is performed or $h = \bar{H}$.
11:                $s', h' \leftarrow$ current state and time step
12:                **if** $g$ was performed **then**
13:                    **return** $(s', h')$
14:                **else**
15:                    **return** $(s, \bar{H} + 1)$
16:        **else**
17:            **if** $s \in Z^*$ or $g = \ominus$ **then**
18:                **return** $\ominus$
19:            **else**
20:                **return** $(s, h)$

---

**Meta-dynamics $\mathbb{P}_{\mathrm{hl}}$.** Fix $(z, g) \in \mathcal{Z} \times \mathcal{G}$ for some $z \neq \ominus$, so that $z = (s, h)$. We consider the procedure in Algorithm 7 for generating the meta-dynamics. Intuitively, we execute a meta-action $g \neq \ominus$ by running the oracle-provided policy until the learner encounters $g$, or has acted for $\bar{H}$ timesteps in the current episode. On the other hand, if $g = \ominus$, the agent executes the oracle-provided $\varepsilon$-suboptimal policy that remains within the current cluster and acts for $H - h$ timesteps.

Formally, the next state $z'$ is given by

$$z' = \begin{cases} \ominus & s \in Z^* \text{ or } g = \ominus \\ (s', h') & h \leq \bar{H} \\ (s, h) & \text{otherwise} \end{cases},$$

where $s'$ and $h'$ are generated given $T_{H-h}^{\text{hier}}(s,g)$ as

$$h' \mid T_{H-h}^{\text{hier}}(s,g) = \min(h + T_{H-h}^{\text{hier}}(s,g), \bar{H} + 1)$$

$$s' \mid h' \sim \begin{cases} \mathbb{P}(\cdot \mid g) & h' \leq \bar{H} \\ \delta(s) & \text{otherwise} \end{cases}.$$

Note that the learner can only execute meta-actions while $h \leq \bar{H}$. Furthermore, given access to $\mathcal{M}$, one can easily simulate the dynamics of $\mathcal{M}_{\text{hl}}$.

**Meta-reward $R_{\text{hl}}$.** Fix $((s,h),g) \in \mathcal{Z} \times \mathcal{G}$. Recall that the reward function of $\mathcal{M}$ is supported on $\text{Ext}(\mathcal{S}) \cup (Z^*)^\circ$. Thus, this reward function can be lifted onto $\mathcal{M}_{\text{hl}}$. Formally, we define the following reward function:

$$R_{\text{hl}}(z,g) = \begin{cases} W_h(s) & z = (s,h), s \in Z^* \text{ and } h \leq \bar{H} \\ r(g) & z = (s,h), s \notin Z^* \text{ and } h' \leq \bar{H} \\ 0 & \text{otherwise} \end{cases},$$

where $W_h(s)$ is the random sum of rewards obtained by playing a within-cluster policy starting from $s'$ for the rest of the episode. Note that $R_{\text{hl}}$ depends on $\mathbb{P}_{\text{hl}}$ and is thus random. Furthermore, this reward function is consistent with how meta-transitions are performed in Algorithm 7.

**Meta-horizon $H_{\text{eff}}$.** Recall that there exists an optimal policy that encounters at most $H_{\text{eff}}$ exits with high probability. Accordingly, we limit the learner to being able to choose $H_{\text{eff}}$ high-level actions, which recall can be choices of exits.

**Solving $\mathcal{M}_{\text{hl}}$.** To obtain the desired policy, we apply EULER to $\mathcal{M}_{\text{hl}}$. By the construction in Algorithm 7, the policy set returned by EULER easily translates into policies on $\mathcal{M}$. Furthermore, the value of this policy is the same on both MDPs.

### B.3 PROVING THE REGRET BOUND

Having defined the procedure for learning a policy using the hierarchy, we now proceed with the regret analysis. Our analysis proceeds by constructing a policy expressible in $\mathcal{M}_{\text{hl}}$ that achieves near-optimal returns by imitating the high-level decisions made by $\pi^*$. We then use this policy as a comparator policy when applying EULER regret bounds to $\mathcal{M}_{\text{hl}}$.

To formally construct the desired comparator policy, we need to first define the notion of a meta-history, which contains the set of high-level decisions made by any policy:

**Definition B.1.** Fix a policy $\pi$, which given some horizon $L$, generates a (random) trajectory $(s_0, a_0, \ldots, s_L)$. Let $\text{Ext}(\pi)$ be the number of exits performed in the trajectory, i.e.

$$\text{Ext}(\pi) = \sum_{h=0}^{L-1} \mathbb{1}\left[(s_h, a_h) \in \text{Ext}(\mathcal{S})\right].$$

The meta-history $\mathcal{H}_{\text{hl}}(\pi)$ corresponding to this trajectory is the sequence

$$(z_0, g_0, z_1, g_1, \ldots, z_{\text{Ext}(\pi)}) = (s_{i_0}, (s_{j_0}, a_{j_0}), s_{i_1}, (s_{j_1}, a_{j_1}) \ldots, s_{i_{\text{Ext}(\pi)}}),$$

where

$$i_n := \begin{cases} 0 & n = 0 \\ j_{n-1} + 1 & \text{otherwise} \end{cases}$$

$$j_n := \min_{h=i_n,\ldots,L-1} \mathbb{1}\left[(s_h, a_h) \in \text{Ext}(\mathcal{S})\right].$$

Note that $z_i \in \text{Ent}(\mathcal{S})$ and $g_i \in \text{Ext}(\mathcal{S})$ for all $i = 0, \ldots, \text{Ext}(\pi)$. We omit $\pi$ in writing $\mathcal{H}_{\text{hl}}$ when the underlying policy $\pi$ is understood. $\diamond$

Informally, $\mathcal{H}_{\text{hl}}$ tracks all entrances and exits contained in a trajectory generated by $\pi$. We define the length of a meta-history $\mathcal{H}_{\text{hl}}$, denoted as $|\mathcal{H}_{\text{hl}}|$, as the number of exits contained in $\mathcal{H}_{\text{hl}}$.

### B.3.1 POLICY CONSTRUCTION

We now proceed with constructing the desired policy. Intuitively, the comparator imitates the distribution over $\mathcal{H}_{\mathrm{hl}}(\pi^*)$, conditioned on $|\mathcal{H}_{\mathrm{hl}}(\pi^*)| \leq H_{\mathrm{eff}}$. To see why this is sufficient for near-optimality, recall that the reward on $\mathcal{M}_{\mathrm{Tg}}$ is supported on $\mathrm{Ext}(Z^*) \cup (Z^*)^\circ$. Consequently, by imitating the distribution over meta-histories, the policy is expected to obtain roughly the same sum of rewards in expectation from the exits. Therefore, all that remains is to ensure that the learner collects roughly the same sum of rewards from $Z^*$, which is the same as ensuring that this policy does not take too long to reach $Z^*$.

**Construction.** Let $\mathcal{H}$ be the running meta-history, containing $k \leq H_{\mathrm{eff}}$ actions. The optimal policy induces a distribution $q(\cdot \mid \mathcal{H})$ over $\mathcal{A}_{\mathrm{hl}}$ representing the next exit it takes[6]. We then define $\pi$ as

$$\pi(\cdot \mid z, \mathcal{H}) = \begin{cases} q(\cdot \mid \mathcal{H}) & z = (s, h), h < \bar{H} \\ \ominus & \text{otherwise.} \end{cases}$$

Observe that $\pi$ terminates the episode upon reaching $\bar{H}$. Furthermore, this policy is dependent on the meta-history. However, since $\mathcal{M}_{\mathrm{hl}}$ is an MDP, there exists a stationary policy that achieves at least the same value.

### B.3.2 SUBOPTIMALITY ANALYSIS

In this section, we prove that $\pi$ achieves bounded suboptimality. Rather than analyzing $\pi$ directly in $\mathcal{M}_{\mathrm{hl}}$, we construct a new $\tilde{\mathcal{M}}_{\mathrm{hl}}$ and $\tilde{\pi}$ to better track the meta-history. In particular, conditioned on the event that $\pi$ requires more than $\bar{H}$ time steps to execute, then the agent would not be able to imitate the full meta-history generated by $\pi^*$, even after having performed less than $H_{\mathrm{eff}}$ exits.

**Constructing a surrogate for analysis.** We now formalize the construction of the surrogate MDP $\tilde{\mathcal{M}}_{\mathrm{hl}}$ and the policy $\tilde{\pi}$ corresponding to $\pi$ in this MDP. To obtain $\tilde{\mathcal{M}}_{\mathrm{hl}}$, we redefine the dynamics from $\mathcal{M}_{\mathrm{hl}}$ so that $s' \mid h' \sim \mathbb{P}(\cdot \mid g)$ in $\tilde{\mathcal{M}}_{\mathrm{hl}}$. In effect, we allow the policy to continue performing transitions beyond $\bar{H}$, although without any reward. Accordingly, we define $\tilde{\pi}$ as $\tilde{\pi}(\cdot \mid z, \mathcal{H}) = q(\cdot \mid \mathcal{H})$. The following lemma formalizes how $\tilde{\pi}$ and $\tilde{\mathcal{M}}_{\mathrm{hl}}$ have desirable properties for the analysis:

**Lemma B.1** (Surrogate Policy Characterization)**.** *Let $\mu^*$ denote the distribution of $\mathcal{H}_{\mathrm{hl}}(\pi^*) \mid |\mathcal{H}_{\mathrm{hl}}(\pi^*)| \leq H_{\mathrm{eff}}$ in $\mathcal{M}_{\mathrm{Tg}}$, and $\tilde{\mu}$ the distribution of $\mathcal{H}(\tilde{\pi})$ in $\tilde{\mathcal{M}}_{\mathrm{hl}}$. Then, $(1-\zeta)\mu^* \leq \tilde{\mu}$.*

*Proof.* Let $\nu^*$ be the distribution induced by the following procedure:

(1) Sample a meta-history from the distribution $\mathcal{H}_{\mathrm{hl}}(\pi^*) \mid |\mathcal{H}_{\mathrm{hl}}(\pi^*)| > H_{\mathrm{eff}}$.

(2) Truncate the obtained meta-history to length $H_{\mathrm{eff}}$.

It is easy to see from the definition of $\tilde{\pi}$ that $\tilde{\mu} = (1-\zeta)\mu^* + \zeta\nu^*$. The desired result follows. $\square$

Thus, we have indeed shown the desired property that $\tilde{\pi}$ properly tracks the (truncated) meta-history generated by $\pi^*$. To justify performing our analysis on $(\tilde{\mathcal{M}}_{\mathrm{hl}}, \tilde{\pi})$, we have the following result, which shows that any result on the value of the pair above applies to the value of $\pi$ in $\mathcal{M}_{\mathrm{hl}}$.

**Lemma B.2.** *As constructed above, $V_0^{\tilde{\mathcal{M}}_{\mathrm{hl}}, \tilde{\pi}}(s_0) = V_0^{\mathcal{M}_{\mathrm{hl}}, \pi}(s_0)$.*

*Proof.* We write $\mathcal{M} := \mathcal{M}_{\mathrm{hl}}$ and $\mathcal{M}' := \tilde{\mathcal{M}}_{\mathrm{hl}}$. Similarly, we write $\pi' := \tilde{\pi}$. We proceed by proving a chain of equalities.

$(V^{\pi', \mathcal{M}'}(s_0) = V^{\pi, \mathcal{M}'}(s_0))$. We omit $\mathcal{M}'$ in this part of the argument for clarity. By the performance difference lemma, we have that for any $k \in [H_{\mathrm{eff}}]$ and $z \in \mathcal{S}_{\mathrm{hl}}$,

$$V_0^\pi(s_0) - V_0^{\pi'}(s_0) = \sum_{j=0}^{H_{\mathrm{eff}}-1} \mathbb{E}_{z \sim d_j^\pi}\left[A_j^{\pi'}(z, \pi)\right].$$

---

[6]The distribution $q$ can return $\ominus$ if the learner stays in the cluster until episode termination.

Let $\Delta := \left\{ z \in \mathcal{S}_{\mathrm{hl}} \mid z = (s, h), s \notin Z^*, h \geq \bar{H} \right\}$, which is the set on which $\pi$ and $\pi'$ disagree. Observe that for any $\pi$ and $k$, $V_k^\pi(z) = 0$ for any $z \in \Delta$, and thus $A_k^{\pi'}(z, \pi) = 0$ for all such states. For any other $z$, $A_k^{\pi'}(z, \pi)$ is clearly 0. Thus, we obtain the desired result.

$(V^{\pi, \mathcal{M}'}(s_0) = V^{\pi, \mathcal{M}}(s_0))$ We omit $\pi$ in this part of the argument for clarity. Using the simulation lemma,

$$V_0^{\mathcal{M}}(s_0) - V_0^{\mathcal{M}'}(s_0) = \sum_{j=0}^{H_{\mathrm{eff}}-1} \mathbb{E}_{(z,g) \sim d_j^{\mathcal{M}'}} \left[ [(\mathbb{P}_{\mathcal{M}} - \mathbb{P}_{\mathcal{M}'}) V_{j+1}^{\mathcal{M}}](z, g) \right].$$

Observe that the behavior of the two MDPs are identical conditioned on $h' \leq \bar{H}$. On the other hand, conditioned on $h' > \bar{H}$, $\pi$ can no longer receive rewards from either MDP. Therefore, $[(\mathbb{P}_{\mathcal{M}} - \mathbb{P}_{\mathcal{M}'}) V_j^{\mathcal{M}}](z, g) = 0$ for any $j, z, g$ by decomposing the relevant expectations along the two events. We thus obtain the desired result. $\qquad\square$

**Analyzing the surrogate.**  With the results above, we now proceed to analyze the difference in values

$$V_0^{\mathcal{M}_{\mathrm{Tg}}, *}(s_0) - V_0^{\tilde{\mathcal{M}}_{\mathrm{hl}}, \tilde{\pi}}(s_0),$$

which then implies the desired suboptimality result. First, we have the following lemma characterizing the time $\tilde{\pi}$ requires to fully execute a given meta-history in the base MDP $\mathcal{M}$:

**Lemma B.3.** *Fix any $\mathcal{H}_{\mathrm{hl}} = (z_0, g_0, \dots)$ such that $|\mathcal{H}_{\mathrm{hl}}| \leq H_{\mathrm{eff}}$. Furthermore, define the sequence of reaching times*

$$T_0 := 0 \quad \text{and} \quad T_k := T_{k-1} + T_{H-T_{k-1}}^{\mathrm{hier}}(z_{k-1}, u_{k-1}).$$

*We define $T^{\mathrm{hier}}(\mathcal{H}_{\mathrm{hl}})$ to be the time required by the hierarchy to execute $\mathcal{H}_{\mathrm{hl}}$, which is formally given by $T_{|\mathcal{H}_{\mathrm{hl}}|}$ in the sequence above. Then,*

(a) $\mathbb{E}\left[ T^{\mathrm{hier}}(\mathcal{H}_{\mathrm{hl}}) \right] \leq [1 + (1 + \gamma)W + \varepsilon] H_{\mathrm{eff}}$.

(b) *Let $\sigma^2 := \beta^2 [(1 + \gamma)W + \varepsilon]^2 H_{\mathrm{eff}}$. Then, for any $t > 0$,*

$$P\left( T^{\mathrm{hier}}(\mathcal{H}_{\mathrm{hl}}) \geq [1 + (1 + \gamma)W + \varepsilon] H_{\mathrm{eff}} + t \right) \leq e^{-t^2/2\sigma^2}.$$

*Proof.* We prove the two parts separately:

(a) We will prove via induction that $\mathbb{E}[T_k] \leq k[1 + (1 + \gamma)W + \varepsilon]$. For any $k$ and $T_{k-1}$,

$$\mathbb{E}\left[ T_{H-T_{k-1}}^{\mathrm{hier}}(z_{k-1}, g_{k-1}) \,\Big|\, T_{k-1} \right] \leq \mathbb{E}\left[ T_{H-T_{k-1}}^*(z_{k-1}, g_{k-1}) \,\Big|\, T_{k-1} \right] + \varepsilon$$

$$= 1 + \mathbb{E}\left[ T_{H-T_{k-1}}^*(z_{k-1}, s(g_{k-1})) \,\Big|\, T_{k-1} \right] + \varepsilon$$

$$\leq 1 + (1 + \gamma)W + \varepsilon,$$

where the first inequality uses properties of the hierarchy oracle, while the final inequality follows by combining Definition 5.2(b) and Assumption 5.2. Therefore, by linearity and the tower property of expectation,

$$\mathbb{E}[T_k] = \mathbb{E}[T_{k-1}] + \mathbb{E}\left[ T_{H-T_{k-1}}^{\mathrm{hier}}(z_{k-1}, g_{k-1}) \right]$$

$$= \mathbb{E}[T_{k-1}] + \mathbb{E}\left[ \mathbb{E}\left[ T_{H-T_{k-1}}^{\mathrm{hier}}(z_{k-1}, g_{k-1}) \,\Big|\, T_{k-1} \right] \right]$$

$$\leq \mathbb{E}[T_{k-1}] + 1 + (1 + \gamma)W + \varepsilon.$$

The desired result then follows by induction.

(b) Let $B_k := k[1 + (1 + \gamma)W + \varepsilon]$ and $f_k(t) := \mathbb{E}\left[ T_{H-t}^{\mathrm{hier}}(z_k, g_k) \right]$. Note that for any $k$ and $t$, $B_{k-1} + f_{k-1}(t) \leq B_k$, by following the argument in (a). Therefore, for any $\lambda > 0$,

$$\mathbb{E}\left[ \exp\left\{ \lambda (T_k - B_k) \right\} \right]$$

$$= \mathbb{E}\left[ \mathbb{E}\left[ \exp\left\{ \lambda (T_k - B_k) \right\} \mid T_{k-1} \right] \right]$$

$$\leq \mathbb{E}\left[ \mathbb{E}\left[ \exp\left\{ \lambda\left( T_{k-1} + T_{H-T_{k-1}}^{\mathrm{hier}}(z_{k-1}, g_{k-1}) - B_{k-1} - f_{k-1}(T_{k-1}) \right) \right\} \mid T_{k-1} \right] \right],$$

where the last inequality uses the monotonicity of the exponential function. Therefore, by applying the sub-Gaussian condition given in Definition 5.2,

$$\mathbb{E}\left[\exp\left\{\lambda\left(T_k - B_k\right)\right\}\right]$$

$$\leq \mathbb{E}\left[\exp\left\{\lambda\left(T_{k-1} - B_{k-1}\right)\right\}\right.$$

$$\left.\mathbb{E}\left[\exp\left\{\lambda\left(T_{H-T_{k-1}}^{\text{hier}}(z_{k-1}, g_{k-1}) - f_{k-1}(T_{k-1})\right)\right\} \mid T_{k-1}\right]\right]$$

$$\leq \mathbb{E}\left[\exp\left\{\lambda(T_{k-1} - B_{k-1})\right\}\right]\exp\left[\lambda^2 C^2/2\right],$$

where we have used the fact that $T_{H-T_{k-1}}^{\text{hier}}(z_{k-1}, s(g_{k-1}))$ has a sub-Gaussian upper tail with variance proxy

$$C^2 = \beta^2 \mathbb{E}\left[T_{H-T_{k-1}}^{\pi}(z_{k-1}, s(g_{k-1})) \mid T_{k-1}\right]^2$$

$$\leq \beta^2\left[(1+\gamma)W + \varepsilon\right]^2.$$

Note that we have once again used the properties of the hierarchy oracle, and Assumption 5.2. Therefore, by induction, $\mathbb{E}\left[\exp\left\{\lambda\left(T_k - B_k\right)\right\}\right] \leq \mathbb{E}\left[\lambda^2(\sqrt{k}C)^2/2\right]$, from which the desired tail bound follows by making use of Chernoff's inequality. $\qquad\square$

As we have shown that $\tilde{\pi}$ closely tracks the meta-history of $\pi^*$ and have analyzed the distribution of time it takes to execute a given meta-history, we can now analyze its suboptimality:

**Lemma B.4.** *There exists a policy $\pi$ expressible in $\mathcal{M}_{\text{hl}}$ such that*

$$V_0^{\mathcal{M}_{\text{Tg}},*}(s_0) - V_0^{\mathcal{M}_{\text{Tg}},\pi}(s_0) \lesssim (1 + H_{\text{eff}} + \beta\sqrt{H_{\text{eff}}})\varepsilon + \left[\gamma H_{\text{eff}} + \beta(1+\gamma)\sqrt{H_{\text{eff}}}\right]W + \zeta H.$$

*Proof.* Assume that $\pi^*$ generates a (random) meta-history of length $N$ given by $\mathcal{H}_{\text{hl}} = (z_0, g_0, z_1, g_1, \ldots, z_N)$. Furthermore, let $T^*$ denote the (random) time $\pi^*$ takes to reach $z_N$. Then, given $\mathcal{H}_{\text{hl}}$ and $T^*$, observe that we can write

$$V_0^*(s_0) = \mathbb{E}\left[R_{T^*}^*(\mathcal{H}_{\text{hl}})\right], \quad \text{where } R_T^*(\mathcal{H}_{\text{hl}}) := V_T^*(z_N)\mathbb{1}\left[z_N \in Z^*\right] + \sum_{k=0}^{N-1} r(g_k),$$

using the assumptions on the reward function and condition (a) in Assumption 5.1. Subsequently, letting $E$ be the event $\{N \leq H_{\text{eff}}\}$, we can bound the right-hand side as

$$V_0^*(s_0) = \mathbb{E}\left[R_{T^*}^*(\mathcal{H}_{\text{hl}})\right] \leq (1-\zeta)\mathbb{E}\left[R_{T^*}^*(\mathcal{H}_{\text{hl}}) \mid E\right] + \zeta H,$$

where we have used Assumption 5.1 to bound the probability that $N > H_{\text{eff}}$.

Our goal for the rest of this proof is to transform the expectation on the right-hand side into a form that lower bounds $V_0^\pi(s_0)$. To this end, we define

$$R_T^{\text{hier}}(\mathcal{H}_{\text{hl}}) := V_T^{\text{hier}}(z_N)\mathbb{1}\left[z_N \in Z^*\right] + \sum_{k=0}^{N-1} r(g_k),$$

and the sequence of times

$$T_0 = 0 \quad \text{and} \quad T_k := T_{k-1} + T_{H-T_{k-1}}^{\text{hier}}(z_k, g_k).$$

Note that $R^{\text{hier}}$ and $T_N$ are analogous to $R^*$ and $T^*$, respectively. Then, letting $F$ be the event $\{T_N \leq \bar{H}\}$, note that

$$V_0^*(s_0) \leq (1-\zeta)\mathbb{E}\left[R_{T^*}^* \mid E\right] + \zeta H$$

$$= (1-\zeta)\mathbb{E}\left[R_{T^*}^* - R_{T_N}^{\text{hier}} + R_{T_N}^{\text{hier}} \mid E\right] + \zeta H$$

$$\leq \underbrace{\mathbb{E}\left[\left(R_{T^*}^* - R_{T_N}^{\text{hier}}\right)\mathbb{1}\left[F\right] \mid E\right]}_{(\text{I})} + \underbrace{(1-\zeta)\mathbb{E}\left[R_{T_N}^{\text{hier}}\mathbb{1}\left[F\right] \mid E\right]}_{(\text{II})} + \left[\zeta + P\left(F^C \mid E\right)\right]H.$$

We bound (I) and (II) separately.

*Bounding* (I). Let $G$ be the event $E \cap \{z_N \in Z^*\}$. Then, if we define

$$T_{\min} = \sum_{k=0}^{N-1} T_{\min}(z_k, u_k) \le N(W+1)$$

as the minimum time needed to execute $\mathcal{H}_{\mathrm{hl}}$, we then have that

$$
\begin{aligned}
\mathbb{E}\left[\left(R_{T^*}^* - R_{T_N}^{\mathrm{hier}}\right) \mathbb{1}\left[F\right] \mid E\right] &\le \mathbb{E}\left[R_{T^*}^* - R_{T_N}^{\mathrm{hier}} \mid E\right] \\
&\le \mathbb{E}\left[V_{T^*}^*(z_N) - V_{T_N}^{\mathrm{hier}}(z_n) \mid G\right] \\
&\le \mathbb{E}\left[V_{T_{\min}}^*(z_N) - V_{T_N}^{\mathrm{hier}}(z_N) \mid G\right] \\
&\le \int_0^H P\left(V_{T_{\min}}^*(z_N) - V_{T_N}^{\mathrm{hier}}(z_N) > \alpha \mid G\right) \, \mathrm{d}\alpha.
\end{aligned}
$$

Note that the bound on $T_{\min}$ follows from Assumption 5.2. To convert the different in values into a difference of times, observe that if $T_N - T_{\min} \le \alpha - \varepsilon$, then

$$
\begin{aligned}
V_{T_{\min}}^*(z_N) - V_{T_N}^{\mathrm{hier}}(z_N) &= V_{T_{\min}}^*(z_N) - V_{T_N}^*(z_N) + V_{T_N}^*(z_N) - V_{T_N}^{\mathrm{hier}}(z_N) \\
&\le (\bar{T} - T_{\min}) + \varepsilon \\
&\le \alpha.
\end{aligned}
$$

Therefore,

$$
\begin{aligned}
&\int_0^H P\left(V_{T_{\min}}^*(z_N) - V_{T_N}^{\mathrm{hier}}(z_N) > \alpha \mid G\right) \, \mathrm{d}\alpha \\
&\le \int_0^H P(T_N - T_{\min} > \alpha - \varepsilon \mid G) \, \mathrm{d}\alpha \\
&\le \mathbb{E}\left[\int_0^H P(T_N - T_{\min} > \alpha - \varepsilon \mid \mathcal{H}_{\mathrm{hl}}) \, \mathrm{d}\alpha \,\middle|\, G\right] \\
&\le \mathbb{E}\left[\int_0^H P(T_N - [1 + (1+\gamma)W + \varepsilon]H_{\mathrm{eff}} > \alpha - \varepsilon - H_{\mathrm{eff}}(\gamma W + \varepsilon) \mid \mathcal{H}_{\mathrm{hl}}) \, \mathrm{d}\alpha \,\middle|\, G\right] \\
&\le \varepsilon + H_{\mathrm{eff}}(\gamma W + \varepsilon) + \mathbb{E}\left[\int_0^\infty P(T_N - [1 + (1+\gamma)W + \varepsilon]H_{\mathrm{eff}} > \alpha \mid \mathcal{H}_{\mathrm{hl}}) \, \mathrm{d}\alpha \,\middle|\, G\right] \\
&\lesssim \varepsilon + H_{\mathrm{eff}}(\gamma W + \varepsilon) + \beta[(1+\gamma)W + \varepsilon]\sqrt{H_{\mathrm{eff}}},
\end{aligned}
$$

where the final inequality integrates the tail bound provided in Lemma B.3. Overall, we have that by rearranging,

$$(\mathrm{I}) \lesssim (1 + H_{\mathrm{eff}} + \beta\sqrt{H_{\mathrm{eff}}})\varepsilon + \left[\gamma H_{\mathrm{eff}} + \beta(1+\gamma)\sqrt{H_{\mathrm{eff}}}\right] W.$$

*Bounding* (II). By the characterization of $\tilde{\pi}$ in Lemma B.1,

$$(1 - \zeta)\mathbb{E}\left[R_{T_N}^{\mathrm{hier}}(\mathcal{H}_{\mathrm{hl}}(\pi^*))\mathbb{1}\left[F\right] \mid E\right] \le \mathbb{E}\left[R_{T_N}^{\mathrm{hier}}(\mathcal{H}_{\mathrm{hl}}(\tilde{\pi}))\mathbb{1}\left[F\right]\right] \le V_0^{\tilde{\pi}}(s_0),$$

where the final inequality uses the fact that $\bar{R}_{\bar{T}}(\mathcal{H}_{\mathrm{hl}}(\tilde{\pi}))$ is the return of $\tilde{\pi}$ in $\tilde{\mathcal{M}}_{\mathrm{hl}}$, given $F$.

*Concluding.* Putting all of the previous bounds together, we find that

$$
\begin{aligned}
V_0^*(s_0) \le{} &V_0^{\bar{\pi}}(s_0) + (1 + H_{\mathrm{eff}} + \beta\sqrt{H_{\mathrm{eff}}})\varepsilon + \left[\gamma H_{\mathrm{eff}} + \beta(1+\gamma)\sqrt{H_{\mathrm{eff}}}\right] W \\
&+ \left[\zeta + P\left(F^C \mid E\right)\right] H.
\end{aligned}
$$

By setting $\bar{H}$ to

$$\bar{H} = H_{\mathrm{eff}}[1 + (1+\gamma)W + \varepsilon] + \beta[(1+\gamma)W + \varepsilon]\sqrt{2H_{\mathrm{eff}} \log \frac{1}{\zeta}} \ll H,$$

sub-Gaussian tail bounds on $T_N$ implies that $P\left(F^C \mid E\right) \le \zeta$. Finally, by Lemma B.2,

$$V_0^{\tilde{\mathcal{M}}_{\mathrm{hl}}, \tilde{\pi}}(s_0) = V_0^{\mathcal{M}_{\mathrm{hl}}, \pi}(s_0) = V_0^{\mathcal{M}_{\mathrm{Tg}}, \pi},$$

where the last equality follows by the construction of $\mathcal{M}_{\mathrm{hl}}$. We thus obtain the desired suboptimality bound. $\qquad\square$

### B.3.3 REGRET ANALYSIS

As earlier suggested, we now make use of $\tilde{\pi}$ as a comparator policy in order to prove a regret bound on a learner making use of the procedure outlined in Section B.2.

**Theorem B.1.** *Assume that* EULER *generates policies* $\pi_1, \ldots, \pi_N$ *on* $\mathcal{M}_{\mathrm{hl}}$, *as constructed in Section B.2. Then, we have the following regret bound:*

$$\sum_{k=1}^{N} V_0^*(s_0) - V_0^{\pi_k}(s_0) \lesssim \sqrt{H^2 \bar{H} L M N} + N \varepsilon_{\mathrm{subopt}},$$

*where*

$$\varepsilon_{\mathrm{subopt}} := (1 + H_{\mathrm{eff}} + \beta \sqrt{H_{\mathrm{eff}}})\varepsilon + \left[ \gamma H_{\mathrm{eff}} + \beta(1 + \gamma)\sqrt{H_{\mathrm{eff}}} \right] W + \zeta H.$$

*Proof.* Throughout the proof, we consider applying EULER to $\mathcal{M}_{\mathrm{hl}}$ where the rewards are scaled by $1/H$ to ensure that rewards are bounded in $[0, 1]$. As a result, we can bound $\mathcal{G} \leq 1$ in the EULER regret bound in Zanette & Brunskill (2019), since the sum of rewards in $\mathcal{M}_{\mathrm{hl}}$ is also the sum of rewards in $\mathcal{M}$, and scaling by $1/H$ gives the desired bound on $\mathcal{G}$. Therefore,

$$\sum_{k=1}^{N} V_0^{*, \mathcal{M}_{\mathrm{hl}}}(s_0) - V_0^{\pi_k}(s_0) \lesssim H \sqrt{\frac{1}{H_{\mathrm{eff}}} \bar{H} L M H_{\mathrm{eff}} N} = \sqrt{H^2 \bar{H} L M N}.$$

Furthermore,

$$V_0^*(s_0) - V_0^{*, \mathcal{M}_{\mathrm{hl}}}(s_0) \leq V_0^*(s_0) - V_0^{\pi}(s_0) + V_0^{\pi, \mathcal{M}_{\mathrm{hl}}}(s_0) - V_0^{*, \mathcal{M}_{\mathrm{hl}}}(s_0) \leq \varepsilon_{\mathrm{subopt}}.$$

We thus obtain the desired result. $\qquad\square$

### B.4 AN EXPONENTIAL REGRET SEPARATION FOR A HIERARCHY-OBLIVIOUS LEARNER

In this section, we provide proof of the exponential regret separation between a hierarchical learner and a learner oblivious to the hierarchy. The overall idea behind our proof is the reduction of solving the family of minimax instances described in Domingues et al. (2021) to a particular family of task distributions.

### B.4.1 THE HARD TASK DISTRIBUTION FAMILY

In this section, we describe the family of task distributions that forces any meta-training-oblivious learner to incur exponential regret. For any string $s$, we write $|s|$ for its length.

We now define the family of binary tree room MDPs $\mathbb{M}_W$ of depth $W$. We index a member of this family by a tuple $(\ell^*, a^*, e^*)$, where $\ell^*$ is a binary string of length $W - 1$, and $a^*, e^* \in \{0, 1\}$. The MDP $\mathcal{M}_{(\ell^*, a^*, e^*)} = (\mathcal{S}, \mathcal{A}, \mathbb{P}_{(\ell^*, a^*, e^*)}, r, H)$ corresponding to this tuple is constructed as follows:

**State Space $\mathcal{S}$.** We create a root state $s_{\mathrm{root}}$, $2^W - 1$ states indexed by binary strings of length at most $W - 1$ collected into a set $T = \{s_0, s_1, s_{00}, s_{01}, \ldots\}$, a gate state $s_{\mathrm{gate}}$, and terminal states $\ominus_{\mathrm{trap}}, \ominus_S, \ominus_F$.

**Action Space $\mathcal{A}$.** The set of available actions at every state is the set $\{0, 1\}$.

**Transition Dynamics $\mathbb{P}_{(\ell^*, a^*, e^*)}$.** We define the dynamics as follows:

$$\mathbb{P}_{(\ell^*, a^*, e^*)}(\cdot \mid s, a) = \begin{cases} \delta(s_a) & s = s_{\mathrm{root}} \\ \delta(s_{ta}) & s = s_t \in T, |t| < W - 1 \\ b\delta(s_{\mathrm{gate}}) + (1 - b)\delta(\ominus_{\mathrm{trap}}) & s = s_t \in T, |t| = W - 1, \\ & s \neq s_{\ell^*}, b \sim \mathrm{Ber}\,(1/2) \\ b\delta(s_{\mathrm{gate}}) + (1 - b)\delta(\ominus_{\mathrm{trap}}) & s = s_{\ell^*}, b \sim \mathrm{Ber}\,(1/2 + \varepsilon \mathbb{1}\,[a = a^*]) \\ \delta(\ominus_S) & s = s_{\mathrm{gate}}, a = e^* \\ \delta(\ominus_F) & s = s_{\mathrm{gate}}, a \neq e^*. \end{cases}$$

**Reward Function $r$.**    The reward function is $r(s, a) = \mathbb{1}\left[s = \ominus_S\right] + \mathbb{1}\left[s = s_{\text{gate}}, a = a^*\right]$.

Having described all the components of every member of $\mathbb{M}_W$, all that remains is to construct the family of task distributions $\mathbb{T}_W$. Each member of this family will be indexed by $(\ell^*, a^*)$, where $\ell^*$ and $a^*$ are as described above. Then, the task distribution $\mathcal{T}_{(\ell^*, a^*)} \in \mathbb{T}_W$ chooses uniformly within the set $\left\{\mathcal{M}_{(\ell^*, a^*, 0)}, \mathcal{M}_{(\ell^*, a^*, 1)}\right\}$. Note that this implicitly defines the latent hierarchy so that the clusters are $\{s_{\text{root}}, s_{\text{gate}}, \ominus_{\text{trap}}\} \cup T$, $\{\ominus_S\}$, and $\{\ominus_F\}$. Furthermore, the set of exits for the first cluster is $\{(s_{\text{gate}}, 0), (s_{\text{gate}}, 1)\}$.

### B.4.2    A FAMILY OF HARD INSTANCES

In this section, we describe the family of hard instances which we reduce to solving the task distribution above. Intuitively, if an algorithm incurs low regret throughout $\mathbb{M}_W$, then it must be able to quickly find a policy to reliable reach the gate state $s_{\text{gate}}$ for any MDP in the family.

**Constructing the hard instances.**    Accordingly, we define a new MDP family $\mathbb{N}_W$, which now is only indexed by $(\ell^*, a^*)$, and is constructed similarly as any member of $\mathbb{M}_W$, but ignoring states outside $\{s_{\text{root}}, s_{\text{gate}}, \ominus_{\text{trap}}\} \cup T$. Additionally, we redefine the reward function $r$ for any member to be $r(s, a) := \mathbb{1}\left[s = s_{\text{gate}}\right]$. We note that this is exactly the set of hard tasks used to prove a minimax regret bound in Domingues et al. (2021).

**The lower bound.**    We state the lower bound result from Domingues et al. (2021), in a slightly more restricted form for ease of proof and presentation. In particular, we consider the following more restricted definition of an algorithm:

**Definition B.2.** Let $\mathcal{H}_n$ be the trajectory data generated by playing a policy $\pi_n$ in an MDP $\mathcal{M}$. That is, $\mathcal{H}_n = ((s_0, a_0, r_0, s_1), (s_1, a_1, r_1, s_2), \ldots, (s_{H-1}, a_{H-1}, r_{H-1}, s_H))$, where $s_0$ and $a_0$ are fixed, $r_h = r(s_h, a_h)$, and $s_{h+1} \sim \mathbb{P}_{\mathcal{M}}(\cdot \mid s_h, a_h)$. Additionally, we set $\mathcal{H}_0 = \varnothing$.

Then, a *valid algorithm* $\mathcal{A}$ for our purposes is one which, for the $n^{\text{th}}$ episode, outputs a deterministic, non-stationary policy $\pi$ that is solely a function of the current state and action and $\bigcup_{i=1}^{n-1} \mathcal{H}_i$. That is, $\mathcal{A}$ does not output policies that adapt to the current running episode.                    ◇

We again emphasize that this restriction is not necessary but that many algorithms nevertheless satisfy this condition (including UCBVI and EULER). We then have the following hardness result:

**Theorem B.2** (Domingues et al. (2021), Theorem 9, restated)**.** *Assume that $W \geq 2$ and $H \geq 3W$. Then, for every algorithm $\mathcal{A}$, there exists an MDP $\mathcal{M} \in \mathbb{N}_W$ such that*

$$\mathbb{E}_{\mathcal{M}, \mathcal{A}}\left[\sum_{n=1}^{N} V_0^*(s_{\text{root}}) - V_0^{\pi_n}(s_{\text{root}})\right] \gtrsim 2^{W/2}\sqrt{H^2 N}.$$

---

**Algorithm 8** The reduction of learning $\mathbb{N}_W$ to learning $\mathbb{M}_W$ in Section B.4.3.

---

1: **procedure** $\mathcal{P}_{\mathcal{A}}(\mathcal{M} \in \mathbb{N}_W)$
2:    Initialize $\mathcal{H}_0 = \varnothing$
3:    **for all** $n \in [N]$ **do**
4:       Obtain $\pi_n = \mathcal{A}(\mathcal{H}_0, \ldots, \mathcal{H}_{n-1})$.
5:       Play $\pi_n$ in $\mathcal{M}$, get history $\mathcal{G}_n = ((s_0, a_0, r_0, s_1), \ldots, (s_{H-1}, a_{H-1}, r_{H-1}, s_H))$.
6:       **if** $s_{W+1} = s_{\text{gate}}$ **then**
7:          $s'_{W+1} \leftarrow s_{W+1}$
8:          **for all** $h = W + 1, \ldots, H - 1$ **do**
9:             **if** $h = W$ **then**
10:                $s'_{h+1} \leftarrow \ominus_S$ if $a_h = 1$ else $\ominus_F$.
11:                $r'_h \leftarrow \mathbb{1}\left[a_h = 1\right]$
12:             **else**
13:                $s'_{h+1} \leftarrow s'_h, r'_h \leftarrow r_h$
14:             Replace $(s_h, a_h, r_h, s_{h+1})$ with $(s'_h, a_h, r'_h, s'_{h+1})$ in $\mathcal{G}_n$
15:       $\mathcal{H}_n \leftarrow \mathcal{G}_n$.

---

### B.4.3 Proving the Hardness Result

We now use the hardness result in the previous section to demonstrate that no algorithm can incur sub-exponential regret in $W$ on all tasks in $\mathbb{M}_W$. We do so by proving that an algorithm solving all tasks in $\mathbb{M}_W$ can be used to construct an algorithm for solving all tasks in $\mathbb{N}_W$.

Formally, let $\mathcal{A}$ be any algorithm for learning any MDP in $\mathbb{M}_W$. We then construct an algorithm $\mathcal{P}_\mathcal{A}$ for learning any MDP in $\mathbb{N}_W$ as in Algorithm 8.

Given this reduction, we aim to prove the following result:

**Proposition B.2.** *For any* $\mathcal{M}_{(\ell^*, a^*)} \in \mathcal{N}_W$, *we have that*

$$\mathbb{E}_{\mathcal{M}_{(\ell^*, a^*)}, \mathcal{P}_\mathcal{A}} \left[ \sum_{n=1}^{N} V_0^*(s_{\mathrm{root}}) - V_0^{\pi_n}(s_{\mathrm{root}}) \right] \leq \mathbb{E}_{\mathcal{M}_{(\ell^*, a^*), 1}, \mathcal{A}} \left[ \sum_{n=1}^{N} V_0^*(s_{\mathrm{root}}) - V_0^{\pi_n}(s_{\mathrm{root}}) \right].$$

To prove this result, we first prove that $\mathcal{P}_\mathcal{A}$ can simulate $\mathcal{M}_{(\ell^*, a^*), 1}$:

**Lemma B.5.** *For any* $n$, *the distribution over* $(\mathcal{H}_0, \ldots, \mathcal{H}_n)$ *induced by running Algorithm 8 over* $\mathcal{M}_{(\ell^*, a^*)} \in \mathbb{N}_W$ *is equal to that induced by running* $\mathcal{A}$ *over* $\mathcal{M}_{(\ell^*, a^*), 1} \in \mathbb{M}_W$.

*Proof.* We proceed by induction. The result holds trivially for $n = 0$.

Now, assume that the result holds for some $n$. We condition on the histories $(\mathcal{H}_0, \ldots, \mathcal{H}_n)$ Then, note that both algorithms play the same policy $\pi_{n+1}$, since $\mathcal{P}_\mathcal{A}$ uses $\mathcal{A}$ to obtain the next policy. As a result, by the construction of $\mathcal{M}_{(\ell^*, a^*)}$ and $\mathcal{M}_{(\ell^*, a^*), 1}$, the distribution over $(s_h, a_h, r_h, s_{h+1})$ are equal for $h \leq W$. Furthermore, Lines $6 - 14$ simulates the dynamics of $\mathcal{M}_{(\ell^*, a^*), 1}$ conditioned on $s_{W+1} = s_{\mathrm{gate}}$, while conditioned on $s_{W+1} = \ominus_{\mathrm{trap}}$, the dynamics of the two MDPs are the same. Therefore, conditioned on any $(\mathcal{H}_0, \ldots, \mathcal{H}_n)$, the distribution over $\mathcal{H}_{n+1}$ induced by the two algorithms are also the same. Thus, the claim holds by induction. $\qquad\square$

Finally, we can prove Proposition B.2.

*Proof of Proposition B.2.* Throughout this proof, we omit the starting state $s_{\mathrm{root}}$ and the timestep $0$ in the value. We prove the result by induction. Clearly, the result holds for $N = 0$.

Assume that the bound holds for some $N$. Then, we have that

$$\mathbb{E}_{\mathcal{M}_{(\ell^*, a^*)}, \mathcal{P}_\mathcal{A}} \left[ \sum_{n=1}^{N+1} V^* - V^{\pi_n} \right]$$

$$= \mathbb{E}_{\mathcal{M}_{(\ell^*, a^*)}, \mathcal{P}_\mathcal{A}} \left[ \sum_{n=1}^{N} V^* - V^{\pi_n} \right] + \mathbb{E}_{\mathcal{M}_{(\ell^*, a^*)}, \mathcal{P}_\mathcal{A}} \left[ V^* - V^{\pi_{N+1}} \right]$$

$$\leq \mathbb{E}_{\mathcal{M}_{(\ell^*, a^*), 1}, \mathcal{A}} \left[ \sum_{n=1}^{N} V^* - V^{\pi_n} \right] + \mathbb{E}_{\mathcal{M}_{(\ell^*, a^*)}, \mathcal{P}_\mathcal{A}} \left[ \mathbb{E} \left[ V^* - V^{\pi_{N+1}} \mid (\mathcal{H}_0, \ldots, \mathcal{H}_N) \right] \right],$$

where the final inequality uses the inductive hypothesis and the tower property of expectation. Now, recall from Lemma B.5 that

$$\mathbb{E}_{\mathcal{M}_{(\ell^*, a^*)}, \mathcal{P}_\mathcal{A}} \left[ \mathbb{E} \left[ V^* - V^{\pi_{N+1}} \mid (\mathcal{H}_0, \ldots, \mathcal{H}_N) \right] \right]$$
$$= \mathbb{E}_{\mathcal{M}_{(\ell^*, a^*), 1}, \mathcal{A}} \left[ \mathbb{E} \left[ V^* - V^{\pi_{N+1}} \mid (\mathcal{H}_0, \ldots, \mathcal{H}_N) \right] \right].$$

We emphasize that the value functions are still with respect to $\mathcal{M}_{(\ell^*, a^*)}$. However, for any policy $\pi$ output by $\mathcal{A}$,

$$V^* - V^\pi = \mathbb{E}_{\mathcal{M}(\ell^*, a^*), \pi} \left[ (H - W - 1) \mathbb{1} \left[ s_{W+1} \neq s_{\mathrm{gate}} \right] \right]$$
$$\leq \mathbb{E}_{\mathcal{M}(\ell^*, a^*), \pi} \left[ (H - W - 1) \mathbb{1} \left[ s_{W+1} \neq s_{\mathrm{gate}} \text{ or } a_{W+1} \neq 1 \right] \right]$$
$$\leq \mathbb{E}_{\mathcal{M}(\ell^*, a^*), 1), \pi} \left[ (H - W - 1) \mathbb{1} \left[ s_{W+1} \neq s_{\mathrm{gate}} \text{ or } a_{W+1} \neq 1 \right] \right].$$

Note that the right-hand side is the regret in $\mathcal{M}(\ell^*, a^*, 1)$ for playing $\pi$. Therefore, since both algorithms play the same policy $\pi_{N+1}$, we thus obtain the desired result by induction. $\qquad\square$

With Proposition B.2, we can now formally state and prove the separation result:

**Theorem B.3.** *There exists a task distribution $\mathcal{T}_{(\ell^*,a^*)} \in \mathcal{T}_W$ such that an algorithm $\mathcal{A}$, without access to the meta-training tasks (and thus without access to the hierarchy), incurs expected regret lower bounded as*

$$\mathbb{E}_{\mathcal{M}\sim\mathcal{T}_{(\ell^*,a^*)}}\left[\mathrm{Regret}_N(\mathcal{M},\mathcal{A})\right] \gtrsim 2^{W/2}\sqrt{H^2N}.$$

*On the other hand, for any task distribution in the family, the hierarchy-based learner $\mathcal{P}$ in Section B.2, with access to a 0-suboptimal hierarchy oracle, achieves regret bounded by $\sqrt{H^2N}$ with high probability on any sampled task.*

*Proof.* Fix any algorithm $\mathcal{A}$. Using Theorem B.2, there exists $\mathcal{M}_{(\ell^*,a^*)}$ such that

$$\mathbb{E}_{\mathcal{M}_{(\ell^*,a^*)},\mathcal{P}_\mathcal{A}}\left[\sum_{n=1}^{N} V_0^*(s_{\mathrm{root}}) - V_0^{\pi_n}(s_{\mathrm{root}})\right] \gtrsim 2^{W/2}\sqrt{H^2N}$$

Thus, by Proposition B.2,

$$\mathbb{E}_{\mathcal{M}_{(\ell^*,a^*,1)},\mathcal{A}}\left[\sum_{n=1}^{N} V_0^*(s_{\mathrm{root}}) - V_0^{\pi_n}(s_{\mathrm{root}})\right] \gtrsim 2^{W/2}\sqrt{H^2N}.$$

Note that the proof in Proposition B.2 can be extended for $\mathcal{M}_{(\ell^*,a^*,0)}$ with appropriate modifications to $\mathcal{P}_\mathcal{A}$, and thus the same inequality holds. Consequently,

$$\mathbb{E}_{\mathcal{M}\sim\mathcal{T}_{(\ell^*,a^*)}}\left[\mathrm{Regret}_N(\mathcal{A})\right] \gtrsim 2^{W/2}\sqrt{H^2N}.$$

On the other hand, with access to the 0-suboptimal hierarchy oracle, observe that the learner only has to plan at timesteps 0 and $W+1$, allowing us to obtain tighter bounds (as $\mathcal{S}_{\mathrm{hl}}$ is smaller than the construction in Section B.2). Furthermore, the suboptimality of planning with the hierarchy oracle is 0 for any task distribution in the family. We thus obtain the desired bound. □

### B.5 A DISCUSSION OF DEFINITION 5.2

In this section, we discuss why the values defined in Definition 5.2 control the suboptimality of the hierarchical learner. In particular, we provide examples of MDPs that satisfy Assumption 5.1, and are thus in a sense tasks that are "compatible with the hierarchy", but nevertheless force a hierarchy-based learner to incur $O(H)$ suboptimality.

#### B.5.1 $(\alpha,\beta)$-UNRELIABILITY

Consider the MDP in Figure 9 with horizon $H+2$ and two actions $a^*$ and $a_1$. The optimal policy chooses $a^*$ at every step, achieving a value of $H - O(1)$, since

$$V_{H+1}^*(s_0) = \frac{1}{2}H + \frac{1}{2}V_H^*(s_1) = \frac{1}{2}H + \frac{1}{4}(H-1) + \frac{1}{4}V_{H-1}^*(s_2)$$

$$= H\sum_{h=1}^{H}\frac{1}{2^h} - \frac{1}{2}\sum_{h=1}^{H}\frac{h}{2^h} = H - O(1).$$

Now, assume that the MDP has a latent hierarchy so that the set of exits are given by $(t_i, a)$ for any $i \in [H]$ and $a \in \mathcal{A}$. Clearly, the optimal hierarchy-based learner would always choose $(t_0, a^*)$ or $(t_0, a_1)$ as its high-level action. However, if the agent fails to transition to $t_0$ at the first timestep due to stochasticity, it will go to the end of the chain, back to $s_0$ and try $a^*$ once more. This is because it already has set a meta-action, and *does not replan until an exit is performed*. Thus, the optimal agent on the meta-MDP achieves a value of $H/2$, and is therefore $O(H)$-suboptimal, even with a 0-suboptimal hierarchy oracle.

Intuitively, hierarchy-based learners as formulated in Section B.2 fail on the MDP in Figure 9 because such learners commit to a skill until completion. Thus, when such skills exhibit high variance in completion times, hierarchy-based learners fare worse than other learners which are able to replan based on the current state (e.g., in this case, choose another exit if $a^*$ fails to take the agent to the current subgoal). Thus, $(\alpha,\beta)$-reliability serves to eliminate such MDPs, ensuring that the skills corresponding to reaching exits are reliable.

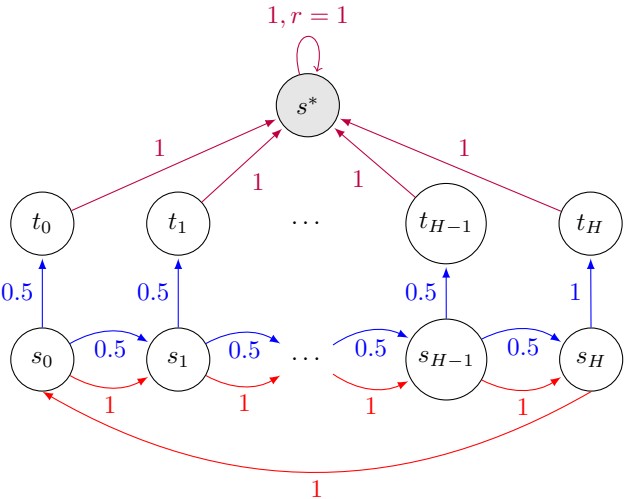

Figure 9: An MDP that does not satisfy low $(\alpha, \beta)$-unreliability, where $a^*$ is in blue, and $a_1$ is in red (and purple for both actions). State shading represents state clusters, and rewards are 0 unless indicated otherwise.

### B.5.2 $\gamma$-GOAL-REACHING SUBOPTIMALITY

In this section, we show that even when a hierarchy-based learner has access to highly reliable skills as in the previous section, the learner may still incur high hierarchical sub-optimality. Consider the MDP in Figure 10, where we focus on a single room for simplicity. Furthermore, assume that there are two exits, one from $l_{H/2}$ and one from $r_{H/2}$. Note that a 0-suboptimal hierarchy oracle has highly reliable goal-reaching policies for reaching both of these exit states, requiring exactly $H/2$ timesteps with no stochasticity.

However, given the values assigned to $l_{H/2}$ and $r_{H/2}$, the optimal policy would opt to take the state $t$, which transitions to either state with probability at least $1/2$ in only two environment steps. Therefore, the optimal policy achieves an optimal value of $H - O(1)$. However, the optimal policy, in having to commit to exactly one of the exits, will achieve a value of $H/2$, and thus be $O(H)$-suboptimal despite having a perfect hierarchy oracle.

Hierarchy-based learners fail on the MDP in Figure 10 because an optimal policy for goal-reaching does not necessarily reach a goal as quickly as possible. Thus, $\gamma$-goal-reaching suboptimality is a regularity condition that ensures that this is indeed the case.

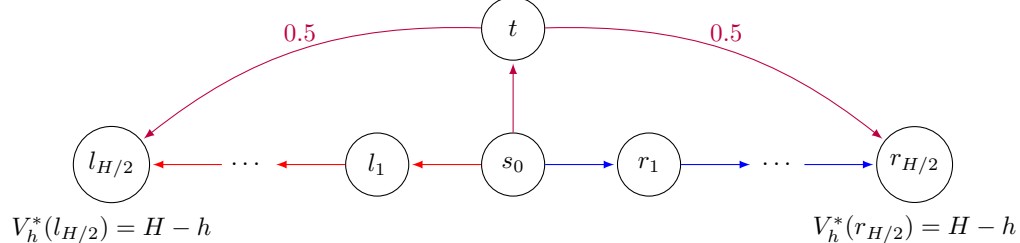

Figure 10: An MDP that does not satisfy low $\gamma$-goal-reaching suboptimality, with three actions indicated by red, blue, and purple, and exits $l_h$ and $r_h$. The MDP satisfies $(\infty, 0)$-unreliability, yet nevertheless exhibits high hierarchical suboptimality.

