# OpenReview forum: "Provable Hierarchy-Based Meta-Reinforcement Learning"
_ICLR.cc/2022/Conference — ICLR 2022 Submitted_

### Official Review · Reviewer_AT7G · 2021-10-28

**Correctness:** 4
**Technical Novelty And Significance:** 3
**Empirical Novelty And Significance:** Not applicable
**Recommendation:** 5
**Confidence:** 3

**Main Review:**

However, the current version does not convince me to recommend acceptance due to the following concerns:

Q1. The authors assume that the agent has access to transition models (including transition probabilities and reward functions), but transition models are difficult to obtain especially for complex real-life environments. I, therefore, suspect whether the theory is valid for MDPs with high-dimensional state spaces and complex transition models?

Q2. The authors assume that the latent hierarchy partitions MDPs into clusters such that the non-exit dynamics is unchanged. Does that latent hierarchy exist in general meta-learning RL problems? Additionally, in many meta-learning settings, meta-training tasks do not necessarily share a common state space. I suspect whether the proposed theory and algorithms can be used in general meta-learning RL problems.

Q3. The authors make many assumptions, but never explain whether they are reasonable for realistic RL problems.

Q4. The main text does not introduce how to partition the state space into clusters, and how to optimize the high-level and low-level hierarchical policies.


**Summary Of The Paper:**

The authors develop a theoretical framework to discover the latent hierarchical structures shared across meta-training RL tasks, and propose a tractable hierarchy-learning algorithm with provable guarantees. The paper seems to be theoretically solid, and is well organized. I appreciate the proposed notions, such as latent hierarchy, $\beta$-dynamics separation, $\alpha$-importance, and so on. I believe they can provide researchers with a novel perspective for studying hierarchical RL.

**Summary Of The Review:**

Although the paper seems to be theoretically solid and presents many interesting notions, the current version does not convince me to recommend acceptance.

---

> ### Author Response · Authors · 2021-11-18
> **Response to Reviewer AT7G**
>
> We thank the reviewer for their time as well as their comments on our work. We address the concerns raised below:
>
> **Q1.** Assuming access to the reward function is a standard assumption in theoretical analyses of RL - there exists standard reductions demonstrating that this assumption has no bearing on sample complexity. On the other hand, we emphasize that we do not assume access to ground truth transition models, and instead estimate them over the course of the algorithm.
>
> **Q2.** Our discussion in the main text includes the recently proposed Alchemy environment, which captures our notion of latent hierarchy (under certain restrictions on the task distribution). Furthermore, there are other reasonable examples, including navigating several buildings with similar rooms connected in different ways, or video games with procedurally-generated areas.
>
> While we agree with the reviewer that analyzing meta-learning when the tasks do not share a state space would be interesting, many existing empirical works [1, 2, 3] restrict their attention to task distributions where tasks share state and action spaces. In light of these works and  given the relative lack of theoretical work on meta-RL, we believe that this restriction is of sufficient importance and could be a useful step to analyses of future generalizations of this setting.
>
> **Q3.** Please see our general comments to all reviewers. We note that the original submission already includes extensive discussions for the actual assumptions we make; nevertheless, we would be happy to clarify any specific assumptions that the reviewer has in mind.
>
> **Q4.** The algorithm does not need to partition the state space into clusters to be able to solve the downstream task. The learner only needs to be able to (1) determine, for any entrance, the set of exits in the same cluster, and (2) obtain a policy to reach any such exit in that set. The hierarchy oracle in Definition 5.2, in disconnecting all the clusters, can solve (1) since the only reachable exits are the exits in the same cluster (and thus the oracle would only predict non-zero values for reaching these exits). Solving this reachability problem, phrased as goal-reaching, also provides the desired policy in (2), thus providing the low-level policies for the downstream task. We have clarified this point in our revision in Section 4.3.
>
> Obtaining the high-level hierarchical policy follows from the simple observation that the low-level policies induce a “reduced MDP” where the states are entrances and actions are exits. This simpler MDP can then be solved by any tabular method (in our case, EULER). We have noted this in our revision.
>
> We hope that all the above addresses the reviewer’s concerns, and that they consider increasing their score if our revisions are satisfactory. We would be happy to engage further with the reviewer to address any residual concerns during the remainder of the rebuttal period!
>
> [1] Nagabandi, A., Clavera, I., Liu, S., Fearing, R. S., Abbeel, P., Levine, S., & Finn, C. (2018, September). Learning to Adapt in Dynamic, Real-World Environments through Meta-Reinforcement Learning. In International Conference on Learning Representations.
> [2] Mendonca, R., Gupta, A., Kralev, R., Abbeel, P., Levine, S., & Finn, C. (2019). Guided meta-policy search. arXiv preprint arXiv:1904.00956.
> [3] Liu, E. Z., Raghunathan, A., Liang, P., & Finn, C. (2021, July). Decoupling exploration and exploitation for meta-reinforcement learning without sacrifices. In International Conference on Machine Learning (pp. 6925-6935). PMLR.

---

> > ### Comment · Reviewer_AT7G · 2021-11-27
> > **Response**
> >
> > Thanks for the response that addresses part of my concerns. As I mentioned before, I appreciate the notions proposed by the authors. However, I still concern about whether the theoretical analysis can be applied to general meta-RL problems. The algorithm seems to be specifically designed based on some assumptions. Thus, I keep my score unchanged.

---

### Official Review · Reviewer_wF2a · 2021-10-29

**Correctness:** 4
**Technical Novelty And Significance:** 3
**Empirical Novelty And Significance:** Not applicable
**Recommendation:** 6
**Confidence:** 2

**Main Review:**

Strong points
- paper provides several interesting theoretical results

Weak points
- the study is limited to a tabular setting
- discussion on the relation to the existing methods can be improved

While theoretical analysis is limited to the tabular setting, the paper presents interesting theoretical results for HRL in a meta-RL setting.
Although I'm not the expert in the field of the theoretical analysis of HRL, the regret -guarantee for the meta-test setting seems novel.
Similarly, it is interesting to see that it is provable that the hierarchical oracle is implementable using the proposed algorithm.

On the other hand, I would like to see clearer connection to the existing methods. The methods for discovering the bottleneck states have been investigated for decades, and there are many ways to define the bottleneck states. I would like to encourage authors to discuss the connection between the exits found by the proposed algorithm and the bottleneck states found by the existing methods. For example, I would like to see the connection between the proto-value function and the exits found by the proposed algorithm or $\alpha$-importance used in this study.

Minor comments
- Regarding the work by Wei et al. (2020), I think that it is not very appropriate to state like "they focus on the reduction in regret when the learner already knows the decomposition." In my understanding, their work revealed the sample efficiency of HRL based on some quantities that characterize HRL and suggested what kind of structure should be used in HRL. Although this is just my opinion, please consider better way to phrase the position of the work.

**Summary Of The Paper:**

This paper presents a theoretical analysis of hierarchical reinforcement learning in a meta-RL setting. This work is focused on a tabular  case. To quantify the importance of the state-action pair, $\alpha$-importance is introduced. Subsequently, for exit coverage, optimistic imagination is introduced. The proposed algorithm discovers exits that characterize the hierarchical structure by leveraging these concepts. It is theoretically proved that the hierarchy oracle that reduces the complexity of exploration is implementable using the proposed algorithm. In addition, the regret bound of a hierarchical oracle is theoretically proved.

**Summary Of The Review:**

While theoretical analysis is limited to the tabular setting, the paper presents interesting theoretical results for HRL in a meta-RL setting. To further clarify the contribution of the work, I would like to encourage the authors to deepen the discussion on the relation to the previous studies.

---

> ### Author Response · Authors · 2021-11-18
> **Response to Reviewer wF2a**
>
> We thank the reviewer for their time as well as their suggestions for improving our work. We have rephrased the contributions of Wen et al. (2020) to more faithfully position their paper in our revision.
>
> **On Prior Bottleneck-based Hierarchies.** We have included additional discussion on the connections to prior work on defining hierarchies by finding special bottleneck states in the related work section. We note that the connection between these methods (and proto-value functions) to our definition of bottlenecks do not align in many cases. For example, one of the cluster interiors may have a bottleneck (in the sense of these prior works) whose dynamics are fixed between tasks, and thus our method would not consider such a state-action pair to be an exit. However, this is desirable behavior, since such bottlenecks are not relevant to meta-learning.
>
> We also note that although proto-value functions capture large-scale geometry of the MDP state space, we know of no prior work that has explored its use in bottleneck detection.
>
> We hope the above addresses the reviewer’s concerns, and that they consider increasing their score if our revisions are satisfactory. We would be happy to engage with the reviewer further if any other questions arise during the rebuttal period!

---

### Official Review · Reviewer_WLF1 · 2021-10-30

**Correctness:** 3
**Technical Novelty And Significance:** 3
**Empirical Novelty And Significance:** Not applicable
**Recommendation:** 6
**Confidence:** 4

**Main Review:**

The formulation of hierarchical structures (Definition 4.1) is quite interesting and covers many real situations in HRL. From the discussion on the related work, I understand that this formulation is novel enough, and the understanding of HRL is still limited. I believe that this paper can serve as a good start to the theoretical understanding of HRL.

However, I feel disappointed that there are so many assumptions in both the main text and the appendix. Since most of these assumptions are proposed for the first time and not commonly used in the previous literature, I believe the authors should discuss more on the necessity of each assumptions. Also, there should be more discussion on whether combining these assumptions may exclude common situations in HRL. I state several problems that I am mostly concerned:
 - Assumption 5.2 seems to be made to ensure that optimistic imagination works for the latent hierarchy formulation. However, it looks artificial and I doubt whether this assumption is commonly satisfied in the HRL tasks.
- In Assumption 6.1 you assume that only one cluster has the positive reward. Why this assumption is necessary? This is not the common case in HRL tasks, and can we remove this assumption?
- There are many parameters in these assumptions such as $\alpha$, $\zeta$,$\rho$, $\tilde{H}$, $T^*_{\tilde{H}}$, $T^{min}$ ... If possible, I wonder the magnitude of these parameters in common HRL tasks, e.g. the gated four-room environment used in this paper as an example.

In section 5, the authors propose a novel algorithm to learn the transition dynamics and detect exits. Compared with the query complexity of the brute-force approach, is the complexity in Theorem 5.1 strictly better in all parameter regimes? Can you briefly explain the main difference between your algorithm and the brute-force approach?

Another minor issue is about Theorem 6.2. The regret bound in Theorem 6.2 ignores the sample complexity to learn the hierarchy oracle. If I understand correctly, the total complexity to learn both the hierarchy and the sampled task can still be exponential in $W$ in the hard instance construction. Therefore, there is actually no exponential separation between HRL methods and brute-force methods.

The paper can be further polished by fixing the typos and clarify several statements:

- The definition of Regret in Section 3: $V^*, V^{\pi}$ -> $V_1^*, V_1^{\pi}$
- Section 5.2.3, line 2: What do you mean by optimistically choosing dynamics estimates?
- Section 5.2.3, line 4: $M_t$ in Assumption 5.2, not Assumption 5.1?
- Definition 5.2: What is successful and failed termination? Are they two newly-constructed states in the definition of the new MDP? What is the definition of the distribution $\delta$? Is the input of the oracle the four-tuple $(x,f,r,\tilde{H})$? If so, how can we know $P_t$ used in the definition of $P_f$?
- Theorem 5.1: By saying query complexity, do you mean the total samples needed in all three phases (sample complexity)?


**Summary Of The Paper:**

This paper proposes a novel formulation  to analyze the provable benefits of hierarchical RL algorithms. Based on this new formulation of hierarchical structures, they propose new algorithms to learn the latent hierarchy and apply the extracted hierarchy on the downstream tasks. Under several assumptions, they prove that their algorithms enjoy better regret or sample complexity bounds than brute-force methods.

**Summary Of The Review:**

Overall, I believe that the formulation and algorithms are quite novel, and the hierarchy formulation covers many real situations in HRL. The paper can be further improved by adding more explanation about assumptions and main theorems.

---

> ### Author Response · Authors · 2021-11-18
> **Response to Reviewer WLF1**
>
> We thank the reviewer for their time as well as their suggestions for improving our work. We have incorporated fixes to the typos found by the reviewer in our revised submission. We address the reviewer’s concerns below:
>
> **On the number of assumptions.** Please see our general comments to all reviewers, Section I.
>
> **On optimistic imagination.** Please see our general comments to all reviewers, Section II.
>
> **On hierarchical compatibility.** Please see our general comments to all reviewers, Section III. We also note that our assumption allows for non-zero reward on the exits.
>
> Assuming zero rewards in cluster interiors other than that of $Z^\ast$ is likely necessary, since the low-level policies in the hierarchy seek to exit clusters as quickly as possible (and thus any hierarchical policy can incur arbitrarily bad suboptimality under some adversarially-designed reward function).
>
> **On parameter magnitudes.** $\alpha$ and $\zeta$ represent value differences and thus lie in $(0, H]$. $\rho$ is simply the maximal probability of reaching an entrance, and is thus in $(0, 1]$. The other parameters that the reviewer referenced are reaching times and can vary between different environments, depending on cluster sizes, start and end states, etc. These reaching times are not parameters in the assumptions, but instead are objects of interest that vary between different MDPs.
>
> **Clarifying exponential separation.** Indeed, learning the hierarchy during meta-training incurs exponential complexity. However, Theorem 6.2 highlights that Theorem 6.1 is a few-shot guarantee. In particular, extracting the hierarchical structure during meta-training time has allowed for exponentially faster learning on (compatible) future tasks. This is analogous to supervised meta-learning theory results, where one obtains faster statistical rates on the meta-test task compared to single-task minimax rates. Such results are of particular interest in cases where access to the meta-training tasks is cheap compared to the meta-test task (e.g., sim-to-real).
>
> **Other clarifications:**
>
> > “What do you mean by optimistically choosing dynamics estimates?”
>
> Let $\hat{\mathbb{P}}_1, \dots, \hat{\mathbb{P}}_T$ be the Phase I dynamics models and $\hat{\mathbb{P}}_0$ the reward-free estimate. Then, the proposed Bellman backup is given by
>
> $$Q_h(s, a) \gets r(s, a) + \max_{t \in \{0, \dots, T\}}\mathbb{E}_{s' \sim \hat{\mathbb{P}}_t}\left[V _ {h + 1}(s')\right]$$
>
> assuming $(s, a)$ is not known to be an exit (we do not borrow if $(s, a)$ is an exit as we already know that its dynamics varies between tasks). This backup formalizes the idea of optimistic imagination, in that the backup uses the dynamics estimate that results in the best Q-values.
>
> > Definition 5.2: What is successful and failed termination? Are they two newly-constructed states in the definition of the new MDP? What is the definition of the distribution $\delta$? Is the input of the oracle the four-tuple $(x, f, r, \tilde{H})$? If so, how can we know $P_t$ used in the definition of $P_f$?
>
> Indeed, successful and failed termination are newly-constructed states. $\delta(x)$ is the Dirac delta measure on $x$ (which we note was defined in the “Notation” section). The input of the oracle is indeed the four-tuple. Since the dynamics on the exits is determined by $f$, $P_t$ is only used for non-exits (and thus the $t$ does not matter). Recall that Phase II fully learns the dynamics of a single task to the point that any reward function can be optimized well. Therefore, the reward-free dynamics estimate from Phase II is sufficient for estimating this part of the dynamics.
>
> > “By saying query complexity, do you mean the total samples needed in all three phases (sample complexity)?”
>
> Yes, we do mean sample complexity. We had chosen the terminology to remain consistent with the “Query Model” section at the end of Section 4.
>
> We hope that all of the above addresses the reviewer’s concerns, and that they consider increasing their score if they deem our revisions to be satisfactory. We would be happy to further discuss any lingering concerns that the reviewer may have for the rest of the review period!

---

### Official Review · Reviewer_ht9E · 2021-11-10

**Correctness:** 3
**Technical Novelty And Significance:** 2
**Empirical Novelty And Significance:** Not applicable
**Recommendation:** 3
**Confidence:** 3

**Main Review:**

As the authors themselves point out, while there has been much work in HRL and many methods proposed for how hierarchical structure might be recovered by an agent - there remain little to no formal guarantees (either for the discovery of the hierarchy, or for the performance on novel tasks using the hierarchy). The current paper might therefore constitute an important contribution to an at-present poorly understood area.

I found the paper to be generally well written, albeit perhaps not so well presented. In particular, it seems that in order to get any traction with the problem the authors had to build up a large number of loosely justified assumptions building up to a result which is difficult to reason about given the afore mentioned setup. A clearer presentation might've been to _assume_ the existence of the latent hierarchy (or better yet a comparable structure more closely related to the existing literature) and then prove the regret bounds. And then separately (in a follow on paper or in the appendix) note that the assumed hierarchy is in fact recoverable.

I have some concerns with the specific assumptions / constructions that the paper builds up en-route to it's final result:
- the authors first introduce the notion of a 'latent hierarchy' defined, loosely speaking, in terms of entrance and exit states. While this definition is appealing in the rooms domain I am not convinced about its generality. I did however appreciate the author's attempts to at least somewhat address this question in their consideration of the Alchemy benchmark. Nevertheless this setup feels particularly geared towards domains with obvious bottle-neck states (and there are a number of other papers whose methods would recover the bottle necks at least in this domain). It's worth noting that this sort of condition doesn't lend itself naturally to a description of hierarchy in trivial open domains (like a 1d chain or empty room).
- while some notion of 'coverage' is a typical requirement, I found the proposed 'optimistic imagining' largely unprincipled. The authors first defined the notion of /alpha-importance (ok), then noted that it doesn't work in the simple rooms domain and so propose an alternative definition that does work in the rooms domain... but it's generality remains completely unclear (my intuition is, again, that is probably unique to domains with single bottleneck states).
- the authors propose an algorithm by which the latent hierarchy is recoverable, but it seems that in order to implement it (phase 2) requires that the agent explore in a reward-free way in order to determine the full transition dynamics. That is to say that the proposed algorithm indeed demonstrates that the latent hierarchy is recoverable, but not practically so. Given the paper's earlier assertion that the latent hierarchy they define suggests a natural set of options policies, I wonder whether the whole first part of the paper might be done away with, and the key result demonstrated under the assumption that the option policies are already provided? (in line with my earlier suggestion)
- the key results establish regret bounds for an agent utilizing the latent hierarchy when performing novel tasks... but critically these tasks must themselves satisfy a number of additional conditions - in particular the tasks must be 'hierarchically compatible'. This would seem to exclude the vast majority of tasks (to see this in the rooms domain it seems it would suffice for one to just consider the set of tasks with |S|-choose-k terminal states with equal reward).

And so, in summary, it seems that the key result is established only for a new definition of what constitutes a hierarchy, where the meta-RL tasks satisfy a bespoke definition of coverage, and is then applicable only to a small percentage of tasks (even in the single motivating domain).

With all that said, I think we desperately need more stable theoretical footing in HRL. I would encourage the authors to consider a reframing of their work addressing the separate contributions separately, and hopefully in more generally applicable ways.

**Summary Of The Paper:**

In this paper the authors prove specific regret bounds for an RL agent utilizing a particular hierarchy when solving (some) novel tasks in the domain. The authors further demonstrate that the particular hierarchy considered is recoverable, and present a method for doing so.

**Summary Of The Review:**

An interesting paper proving regret bounds in a meta-learning HRL setting. However, in order to get any traction on the problem, the authors seem to need to make a large number of unprincipled assumptions (and introduce many novel definitions) such that the general interest / applicability of results is in questions.

A reframing of the results in which the authors present the key contributions separately would likely result in a much more impactful contribution.

---

> ### Author Response · Authors · 2021-11-18
> **Response to Reviewer ht9E**
>
> We thank the reviewer for their time as well as their suggestions for improving our submission. We address the reviewer’s comments below:
>
> **Generality of the latent hierarchy definition.** We believe that many domains exhibit our latent hierarchy definition. For example, a learner might perform tasks in several buildings having similar rooms, albeit connected in different ways. Similar structures are also prominent in video games with procedurally-generated areas. While we agree with the reviewer that our setup is geared towards settings with bottleneck states, **bottlenecks have been studied by many works as a basis for hierarchies** [1, 2]. Furthermore, the definition is related to the setup considered in [3]. Despite all this, **we know of no works that pose theoretical hierarchy learning guarantees even in this simple setting**. Thus, we believe that this particular definition, despite not capturing the full complexity of hierarchical RL, is an important first step.
>
> **On the “optimistic imagination” mechanism.** See our general comments to all reviewers, Section II. Furthermore, contrary to the reviewer’s claim, this condition can hold for environments with many bottlenecks. For example, for the four-room environment in the paper, all eight bottlenecks can be detected by including tasks analogous to those illustrated in Figure 4.
>
> **Practicality of hierarchy recovery.** Prior work has demonstrated that reward-free learning can be done efficiently, even in the function approximation setting [4]. In addition, our procedure only needs to explore in a reward-free way on a constant-size subset of the meta-training MDPs.
>
> **On hierarchical compatibility.** Please see our general comments to all reviewers, Section III.
>
> > “it seems that the key result is established only for a new definition of what constitutes a hierarchy, where the meta-RL tasks satisfy a bespoke definition of coverage, and is then applicable only to a small percentage of tasks (even in the single motivating domain).”
>
> Ultimately, as with most theoretical work, assumptions need to be made to derive theoretical results, and it is generally the case that there will be a gap between the analyzed setting and settings in practice. Given the lack of theoretical results on hierarchy recovery as well as minimal results on regret bounds for hierarchy-based learners, we believe that our sufficient characterization of these phenomena is an important first step for future progress. Furthermore, our setting captures the goal-conditioned RL setting prominent in recent empirical work, and thus we believe that the results are not as limited as claimed.
>
> > “I would encourage the authors to consider a reframing of their work addressing the separate contributions separately, and hopefully in more generally applicable ways.”
>
> Our theoretical contribution is the analysis of an end-to-end meta-learning pipeline that can quantify how meta-training performance affects performance on the meta-test task. A significant technical hurdle in this endeavor is ensuring the compatibility of these two phases, which would not be reflected well by addressing these contributions separately. Prior work has only addressed the impact of learning with known options, and thus we believe our submission presents an important and novel theoretical contribution to the literature: **a study of notions of hierarchy that are not only useful, but are also provably learnable**.
>
> We hope that we have addressed the reviewer’s concerns, and that they consider increasing their score if our revisions are satisfactory. We would be happy to engage with the reviewer further if any other questions arise during the rebuttal period!
>
> [1] McGovern, A., & Barto, A. G. (2001, January). Automatic Discovery of Subgoals in Reinforcement Learning using Diverse Density. In ICML.
> [2] Şimşek, Ö., & Barto, A. G. (2004, July). Using relative novelty to identify useful temporal abstractions in reinforcement learning. In Proceedings of the twenty-first international conference on Machine learning (p. 95).
> [3] Wen, Z., Precup, D., Ibrahimi, M., Barreto, A., Van Roy, B., & Singh, S. (2020). On efficiency in hierarchical reinforcement learning. Advances in Neural Information Processing Systems, 33.
> [4] Wang, R., Du, S. S., Yang, L. F., & Salakhutdinov, R. (2020). On reward-free reinforcement learning with linear function approximation. arXiv preprint arXiv:2006.11274.

---

### Author Response · Authors · 2021-11-18
**General Response to all Reviewers**

We thank all of the reviewers for the time and effort they have dedicated to the review process so far, as well as their comments and suggestions for improving our work. We find it encouraging that all reviewers found our results on hierarchical RL to be of interest.

Before addressing concerns, we emphasize that our theoretical contribution is providing the **first end-to-end analysis of hierarchical meta-RL** demonstrating (1) provable learning of a notion of hierarchy which (2) can be leveraged to learn downstream tasks faster, all under appropriate coverage conditions. Given that prior work only considers (2) in isolation, we believe that our work presents an important first step in **studying notions of hierarchy that are not only useful, but also provably learnable**.

Given that the reviewers had similar concerns with regards to certain issues, we address these shared concerns together below:

**I. Number of assumptions.** Several reviewers had concerns regarding the many assumptions we make to derive our theoretical results. However, **many of these assumptions merely quantify certain values rather than actually impose restrictions** on the settings we consider (e.g., Assumptions 5.1, 6.2, A.2, A.3 in the previous version). To clarify this distinction, we changed these assumptions into definitions in the revision (Definitions 4.1, 5.2, A.2, A.3 in the revision) to emphasize that we are merely naming certain values of interest in the analysis.

**II. On the “optimistic imagination” condition.** Optimistic imagination posits that there is a task where changing the bottleneck structure makes the task noticeably easier, using observed bottlenecks from other tasks. This formalizes the intuition of “this happened before while solving a prior task, and thus may be worth exploring in a new task”. Assumption 5.2 can be met simply by having enough meta-training tasks with diverse bottleneck configurations and appropriately chosen reward functions, thus indeed formalizing a notion of coverage. In particular, we emphasize that **the validity of Assumption 5.2 is not a function of the possible family of tasks, but whether there are enough tasks available to the learner**. We have included an additional discussion on this assumption in Appendix A.7.

Furthermore, we note that optimistic imagination and the preliminary coverage condition described in Section 5.1 are complementary to each other. In fact, one can easily check the preliminary coverage condition with our current algorithm, as Phase I dynamics estimates are sufficient for doing so. This can further reduce the required task diversity for hierarchy recovery. We have included this discussion in our revision in Section 4.1.

**III. On “hierarchical compatibility”.** Reviewers raised concerns regarding the “hierarchical compatibility” assumption. However, **Assumption 5.1 captures the goal-conditioned RL setting**, where the learner is tasked with reaching a particular goal state, **a prominent setting in many recent empirical works** [1, 2, 3]. In more theoretical directions, this class of problems is commonly represented as stochastic shortest paths. Ultimately, meta-learning can only succeed if the downstream task is covered by the meta-training tasks, and we believe that the set of covered tasks is of sufficient interest both in theory and in practice.

List of revisions:
1. **Included a discussion in Section A.7 on the naturality of the optimistic imagination condition.**
2. Minor edits to make space for reviewer suggestions.
3. Clarified how the hierarchy oracle allows one to obtain low-level policies.
4. Changed certain assumptions into definitions to clarify that we were defining relevant quantities for the analysis.
5. Moved the related work section to the end to be able to discuss our method in the context of prior work on detecting bottleneck states for option learning. Included such discussion.
6. Included a footnote on how optimistic imagination and the preliminary coverage condition are complementary.
7. Fixed typos pointed out by reviewers, modified pictures to better demonstrate ideas.

[1] Nachum, O., Gu, S. S., Lee, H., & Levine, S. (2018). Data-Efficient Hierarchical Reinforcement Learning. Advances in Neural Information Processing Systems, 31, 3303-3313.
[2] Levy, A., Konidaris, G., Platt, R., & Saenko, K. (2018, September). Learning Multi-Level Hierarchies with Hindsight. In International Conference on Learning Representations.
[3] Nachum, O., Gu, S., Lee, H., & Levine, S. (2018, September). Near-Optimal Representation Learning for Hierarchical Reinforcement Learning. In International Conference on Learning Representations.

---

### Decision · Program_Chairs · 2022-01-20

**Decision:**

Reject

**Comment:**

This paper studies meta-learning in hierarchical RL, where the unknown hierarchy is learned during meta-training and then applied to a test task. The authors propose an optimistic algorithm for solving this problem and analyze it. The main contribution of the paper is in the first end-to-end analysis.

This paper has three borderline reviews and one reject. Despite the differences in the scores, all reviewers share the same opinion. The idea is novel and very interesting. However, the algorithm and its analysis rely on many assumptions, many of which are introduced in this work and not properly discussed. Because of this, the paper needs a major revision and is rejected for now.